**Technical Report**

# Targeted profiling of human extrachromosomal DNA by CRISPR-CATCH

King L. Hung [1], Jens Luebeck [2,3,21], Siavash R. Dehkordi[3,21], Caterina I. Colón[1,4,5], Rui Li[1], Ivy Tsz-Lo Wong[4,5], Ceyda Coruh[6], Prashanthi Dharanipragada[7], Shirley H. Lomeli[7], Natasha E. Weiser [1,5], Gatien Moriceau[7], Xiao Zhang [7], Chris Bailey[8], Kathleen E. Houlahan[9,10,11], Wenting Yang[9,10,11], Rocío Chamorro González [12], Charles Swanton [8,13,14], Christina Curtis[9,10,11], Mariam Jamal-Hanjani [13,14], Anton G. Henssen [12,15,16,17], Julie A. Law [6], William J. Greenleaf [1,10], Roger S. Lo [7,18,19], Paul S. Mischel [4,5], Vineet Bafna [3] and Howard Y. Chang [1,10,20] ✉

Extrachromosomal DNA (ecDNA) is a common mode of oncogene amplification but is challenging to analyze. Here, we adapt CRISPR-CATCH, in vitro CRISPR-Cas9 treatment and pulsed field gel electrophoresis of agarose-entrapped genomic DNA, previously developed for bacterial chromosome segments, to isolate megabase-sized human ecDNAs. We demonstrate strong enrichment of ecDNA molecules containing *EGFR*, *FGFR2* and *MYC* from human cancer cells and *NRAS* ecDNA from human metastatic melanoma with acquired therapeutic resistance. Targeted enrichment of ecDNA versus chromosomal DNA enabled phasing of genetic variants, identified the presence of an *EGFRvIII* mutation exclusively on ecDNAs and supported an excision model of ecDNA genesis in a glioblastoma model. CRISPR-CATCH followed by nanopore sequencing enabled single-molecule ecDNA methylation profiling and revealed hypomethylation of the *EGFR* promoter on ecDNAs. We distinguished heterogeneous ecDNA species within the same sample by size and sequence with base-pair resolution and discovered functionally specialized ecDNAs that amplify select enhancers or oncogene-coding sequences.

Oncogene amplification is a key cancer-driving mechanism and frequently occurs on circular ecDNA. ecDNA oncogene amplifications are present in half of human cancer types and up to one-third of tumor samples and are associated with poor patient outcomes[1–3]. Given the prevalence of ecDNA in cancer, there is an urgent need for better characterization of unique genetic and epigenetic features of ecDNA to understand how it may differ from chromosomal DNA and obtain clues about how it is formed and maintained in tumors. However, isolation and targeted profiling of megabase-sized, clonal ecDNAs is currently challenging due to their large sizes and sequence complexity, in contrast to small kilobase- and subkilobase-sized DNA circles known as

extrachromosomal circular DNA elements (eccDNAs) observed also in non-cancer cells and apoptotic byproducts[4,5].

There are currently three main approaches to analyzing sequences of ecDNAs in cancer cells: (1) DNA fluorescence in situ hybridization (FISH), (2) bulk whole-genome sequencing (WGS) and (3) exonuclease digestion of linear DNA followed by DNA amplification. The first method, DNA FISH, involves arresting cells in metaphase followed by chromosome spreading and hybridization of a DNA probe on a microscope slide. This method provides excellent separation of ecDNA and chromosomal DNA signals and has been used to confirm the presence of oncogenes and drug resistance genes on ecDNA. However, this

**Fig. 1 | Isolation of megabase-sized ecDNA and its native chromosomal locus from the same cancer cell sample by CRISPR-CATCH. a**, Experimental workflow for enrichment of ecDNA and its corresponding chromosomal locus from the same cell sample. **b**, A representative DNA FISH image on a metaphase spread from a GBM39 glioblastoma cell showing extrachromosomal *EGFR* signals and multiple chromosome 7 (chr7) signals (*n* = 65 cells). Quantification of copy numbers is shown in Extended Data Fig. 2a. DAPI, 4,6-diamidino-2-phenylindole. **c**, Design of CRISPR sgRNAs for linearizing ecDNA circles or extracting the native chromosomal locus. **d**, PFGE images showing linearized ecDNA molecules and the chromosomal locus after treatment with indicated guides (Methods; guide

sequences in Supplementary Table 1). Boxed regions indicate parts of the gel that were extracted for DNA isolation. GBM39 ecDNA cutting and fractionation by PFGE were reproduced in three independent experiments. **e**, Normalized short-read sequencing coverage of the expected ecDNA locus in unenriched WGS or after CRISPR-CATCH (guide A). **f**, Fraction of total sequencing reads aligning to the expected ecDNA locus in unenriched WGS or after CRISPR-CATCH (guide A). **g**, Sequencing tracks showing coverages for enriched ecDNA and its chromosomal locus at the zoomed-in locations compared to WGS. Orange arrows indicate locations of sgRNA targets.

method is low throughput (tens of cells) and provides limited, binary sequence information (a probe either binds or does not bind to DNA). The second method, bulk short- or long-read sequencing, provides much higher sequence resolution. However, sequencing signal represents a combination of all DNA material in a sample, including ecDNA and chromosomal DNA. In addition to the ambiguous origin of sequencing reads, rearranged ecDNA sequences are computationally inferred[1,6] but difficult to validate, as sequencing reads are far too short to span the entire length of an ecDNA molecule (typically several megabases). Optical mapping (OM) allows analysis of longer DNA molecules (up to several hundred kilobases) by compromising nucleotide-level information, but each individual OM molecule is typically shorter than an ecDNA circle[7,8]. Sequence segments can be computationally 'stitched' together to form a list of candidate reconstructed paths, though empirically proving the true ecDNA structure, when possible, is very time-consuming and labor-intensive. The third method, exonuclease treatment combined with DNA amplification, is effective for small DNA circles (up to tens of kilobases; Circle-seq[4,9]) and was recently applied to ecDNA in cancer cells[10]. It entails magnetic-bead-based DNA isolation, treatment with an exonuclease to deplete linear DNA, followed by multiple displacement amplification. This method requires intact DNA circles and is therefore highly limited by ecDNA size, as megabase-sized DNA molecules are extremely fragile in solution and prone to breakage. Further, this method requires DNA amplification and, therefore, cannot be used for epigenetic analyses. Phi29, the processive multiple displacement amplification polymerase, produces amplicons that are tens of kilobases and thus amplifies small circles via rolling-circle amplification; however, this is currently challenging for

megabase-sized ecDNA. Finally, analysis of these enriched ecDNAs by short- or long-read sequencing also suffers from the same read length limitations for amplicon reconstruction.

Here, we adapt a previously developed method, termed CRISPR-CATCH[11] (Cas9-assisted targeting of chromosome segments), to specifically enrich for megabase-sized ecDNA from cancer cells and archival patient tumor tissues. DNA amplification is not required; thus, this method allows targeted analyses of both the genetic sequence and epigenomic landscape of isolated ecDNA. We also provide an analytical pipeline for reconstructing amplicon structures de novo with high confidence using sequence information of ecDNA species separated by size.

## Results

### Enrichment and visualization of ecDNA by CRISPR-CATCH

Analysis of tumor samples in The Cancer Genome Atlas (TCGA) showed that most ecDNA sequences predicted were above 200 kb, a larger size range than that obtained from standard high-molecular-weight (HMW) DNA extraction and exonuclease-based circular DNA enrichment (Extended Data Fig. 1a–c)[4,5]. To preserve large intact circular ecDNA, we encapsulated genomic DNA of GBM39 cells (patient-derived glioblastoma neurosphere model containing *EGFR* ecDNA) in agarose plugs (Methods). Fragment size distribution analysis by pulsed field gel electrophoresis (PFGE) showed that virtually all agarose-entrapped genomic DNA containing ecDNA was restricted to either the loading well or the upper compression zone (CZ; region of large DNA molecules; Extended Data Fig. 1a,d). ecDNA was not detectable in the resolution window, indicating that intact circular ecDNA does not migrate freely in

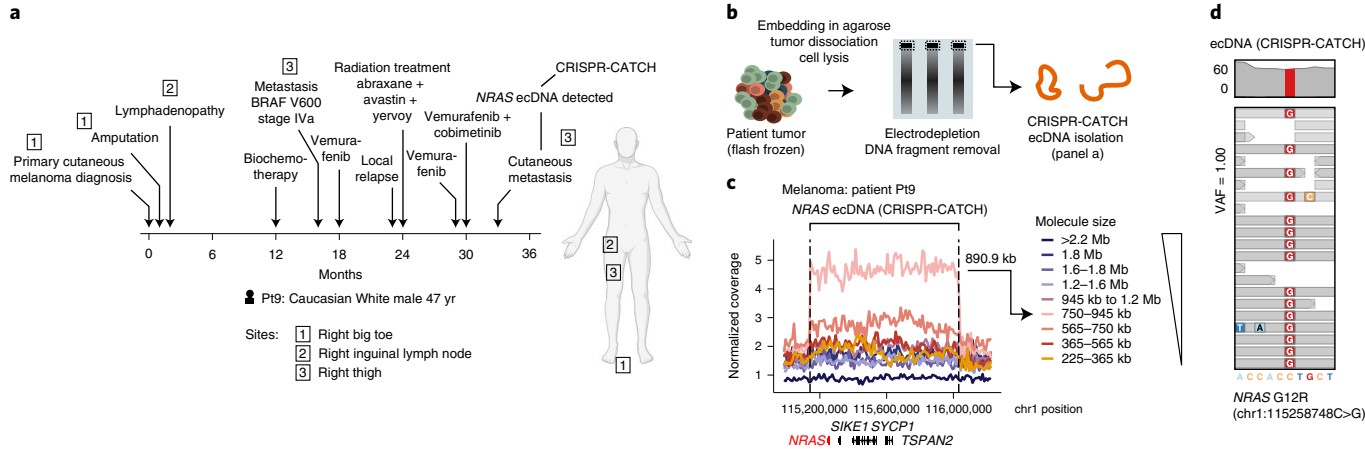

**Fig. 2 | Isolation of ecDNA from a flash-frozen metastatic melanoma tumor. a**, Melanoma patient treatment timeline. *NRAS* ecDNAs were detected in cutaneous metastasis by AmpliconArchitect. Human figure was created with BioRender. com. **b**, A schematic for the tumor processing and electrodepletion protocol for preparing tumor DNA for CRISPR-CATCH. **c**, Normalized short-read sequencing coverage of the expected *NRAS* ecDNA in melanoma patient tumor (Pt9) after CRISPR-CATCH (guide 194; guide sequence in Supplementary Table 1). Amplicon size from sequencing (890.9 kb) was in agreement with molecule size shown by PFGE (750–945 kb). **d**, Sequencing coverage of an *NRAS* G12R mutation identified on ecDNA (top). Sequencing reads supporting single-nucleotide variant (SNV) identification (bottom). VAF, variant allele frequency.

PFGE (Extended Data Fig. 1d). This finding is in agreement with previous Southern blot studies[12–14]. To selectively pull ecDNA into the resolution window of the gel, we preincubated GBM39 genomic DNA in vitro with CRISPR-Cas9 and a single guide RNA (sgRNA) targeting the *EGFR* locus, an amplified sequence on ecDNA. We reasoned that a single cut would linearize ecDNA, resulting in differential migration in PFGE (Fig. 1a). We further reasoned that the same single cut in the corresponding chromosomal locus would result in two much larger chromosomal DNA pieces that migrate much more slowly than ecDNA and therefore would not be coenriched. Cas9 digestion of *EGFR* ecDNA resulted in a prominent band of 1.2–1.37 Mb, concordant with the 1.258-Mb amplicon predicted by bulk WGS and extrachromosomal amplification of the targeted *EGFR* sequence (Fig. 1b–d and Extended Data Fig. 2a)[7,8]. Short-read sequencing of the gel-extracted band confirmed strong enrichment of the expected ecDNA sequence (Fig. 1e,f), demonstrating that a single cut is sufficient to allow enrichment of ecDNA by PFGE. We refer to this method as CRISPR-CATCH (a term previously coined for a two-cut Cas9 treatment followed by gel extraction for isolating and cloning bacterial chromosomal fragments[11,15]). CRISPR-CATCH enabled a 30-fold enrichment of the targeted ecDNA (60% of all sequencing reads versus 2% in WGS), resulting in ultrahigh (~200× normalized) sequencing coverage (Fig. 1e,f and ecDNA in Extended Data Fig. 2b). Simultaneous cleavage of two sgRNA target sites 20 kb away from each other led to loss of the sequence segment between the cut sites, as would be expected given a circular structure and end-to-end junction of the amplified region (Fig. 1g; ecDNA guides A + B). A single cut in the normal diploid chromosomal *EGFR* locus did not result in a DNA band (as shown in Jurkat cells; Fig. 1d), further supporting enrichment of ecDNAs in GBM39 cancer cells. To isolate the chromosomal *EGFR* locus, we performed CRISPR-CATCH using two sgRNAs targeting just outside of the amplified region (upstream and downstream; Fig. 1a,c). This dual-cut strategy resulted in a linear fragment of roughly the same size as the ecDNA molecule and successfully enriched for the chromosomal *EGFR* sequence as demonstrated by increased sequencing coverage around the chromosome-targeting guides (Fig. 1d,g; chromosomal DNA, Extended Data Fig. 2b). Chromosomal gel bands appeared much fainter than ecDNA bands (Fig. 1d), consistent with the fact that ecDNAs exist in higher copy numbers than the chromosomal locus in GBM39 cells. Sequencing coverage analysis further validated enrichment of ecDNA versus chromosomal DNA alleles (Extended Data Fig. 2c,d).

Together, these results showed that CRISPR-CATCH can be used to isolate megabase-sized ecDNA molecules and corresponding chromosomal locus from the same cancer cell sample. Although PFGE was previously used in Southern blot studies to visualize ecDNA sizes[12,13], CRISPR-CATCH provides an empirical pairing of ecDNA amplicon size (by molecular separation) to structure with base-pair resolution (by sequencing).

To expand the capabilities of CRISPR-CATCH, we further optimized a tumor processing protocol for applying CRISPR-CATCH on flash-frozen patient tumor specimens as demonstrated in an instructive case of metastatic melanoma (Fig. 2a and Methods). As tumor specimens can have large amounts of fragmented DNA interfering with CRISPR-CATCH, we introduce electrodepletion, a sequential electrophoretic strategy to remove fragmented DNA from patient tumor samples (Fig. 2b and Methods). This strategy effectively removes DNA fragments and traps intact genomic DNA as well as intact circular ecDNA, as evidenced by removal of DNA size markers as well as successful fractionation of known *FGFR2* ecDNAs from stomach cancer SNU16 cells by CRISPR-CATCH after applying electrodepletion (Extended Data Fig. 3a,b). For our clinical tumor sample, DNA bands were not visible after PFGE due to low amounts of DNA; nonetheless, CRISPR-CATCH still successfully enriched for ecDNAs and confirmed the amplicon size, as shown by strong agreement between the molecular size on the gel and the length of the enriched amplified region in sequencing (Fig. 2c and Extended Data Fig. 3c,d). This clinical tumor sample was obtained from a patient with *BRAF* V600-mutated melanoma who was treated with BRAF and MEK inhibitors and developed a metastatic lesion with acquired resistance coincident with the acquisition of ecDNA (Fig. 2a). CRISPR-CATCH and AmpliconArchitect confirmed the amplification of an 890-kb ecDNA encompassing *NRAS*, a gene known to confer acquired resistance to BRAF inhibition[16] as well as combined BRAF and MEK inhibition when amplified[17] (Fig. 2c and Extended Data Fig. 3c). The *NRAS* amplicon breakpoints coincided with boundaries of topologically associating domains in a melanoma cell line (Extended Data Fig. 3e); the 3′ portion of the amplicon region encompasses a topologically associating domain containing multiple peaks of histone H3 lysine 27 acetylation (H3K27ac) in at least one of seven human cell types (Extended Data Fig. 3e,f), pointing to potential enhancers that may be rewired to the 5′ located *NRAS* gene via ecDNA circularization. An *NRAS* G12R missense mutation, which locks NRAS in the GTP-bound active

conformation and previously linked to melanoma[18], was identified on ecDNAs with an allele frequency of 100%, suggesting strong selection for the mutated allele on ecDNAs (Fig. 2d). Notably, this metastatic tumor sample was 10 years old at the time of ecDNA isolation (biopsy in October 2012), showing that CRISPR-CATCH is fully feasible on archival human tumor specimens. These data further validate an ecDNA mechanism for acquired resistance to MAP kinase pathway inhibitors in authentic human cancer.

### Phasing of oncogenic variants on ecDNA and identification of the chromosomal origin of ecDNA

Next, we performed targeted analysis of the genetic sequences of ecDNA and chromosomal DNA containing the *EGFR* locus in GBM39 cells (Fig. 3a). From ecDNA and chromosomal DNA molecules containing the *EGFR* locus isolated using CRISPR-CATCH, we first identified structural variants (SVs) in short-read sequencing data. GBM39 cells were previously shown to harbor the *EGFRvIII* deletion, an activating *EGFR* mutation[7,8,19]. Importantly, sequencing coverage combined with breakpoint analysis of CRISPR-CATCH data revealed that the *EGFRvIII* mutation is predominantly found on ecDNA, while the chromosomal locus mainly contains full-length *EGFR* (Fig. 3b). Wild-type *EGFR* appeared at ~75% in the chromosomal fraction, consistent with the level of chromosomal DNA enrichment and suggesting that the remaining ~25% *EGFRvIII* comes from carryover ecDNAs (Fig. 3b, Extended Data Fig. 2d). This observation suggests selection and amplification of the *EGFRvIII* mutation and supports previous studies suggesting that ecDNA may help cancer cells adapt to selective pressure and harbor unique genetic alterations[6,20,21].

We then assessed the frequencies of SNVs found on enriched ecDNA and chromosomal DNA. Notably, we observed strong divergence of SNVs on ecDNA compared to those on chromosomal DNA, suggesting that they were haplotype-specific germline variants originating from different parental alleles (Extended Data Fig. 4a,b). Similar to the *EGFRvIII* analysis, unique SNVs located in the chromosomal fraction exhibited allele frequencies of 70–75%, consistent with the level of chromosomal DNA enrichment (Extended Data Fig. 4a,c). CRISPR-CATCH also identified low-frequency subclonal mutations on ecDNA and chromosomal DNA (Extended Data Fig. 4c,d). Importantly, these subclonal mutations on ecDNA are indistinguishable from chromosomal SNVs in bulk WGS data based on variant allele frequencies (VAFs) alone but can be clearly phased using CRISPR-CATCH (Fig. 3c and Extended Data Fig. 4b–d). The divergent ecDNA and chromosomal haplotypes strongly suggest that ecDNA arose from a single chromosomal allele (allele 1), whereas the second allele (allele 2) containing wild-type *EGFR* is still present on chromosomal DNA. Based on this finding, we asked whether the chromosomal allele from which ecDNA originated (allele 1) can still be detected. Although there are six copies of chromosome 7 (native location of *EGFR*) in GBM39 cells (Fig. 1b and Extended Data Fig. 2a), quantification of VAFs in the chromosomal arm upstream and downstream of the *EGFR*-amplified region showed that one haplotype corresponded to one copy of chromosome 7, whereas a second haplotype corresponded to five copies (Fig. 3d). We further identified an SV resulting from deletion of the amplified region corresponding to one copy of chromosome 7, suggesting that it was an excision scar left behind during the formation of ecDNA (Fig. 3d). Together, this analysis shows the sequence of genomic events that preceded the formation of ecDNA and provides strong evidence for an excision model of ecDNA genesis (Fig. 3e). From the two original parental alleles, there was a DNA rearrangement event on allele 1 that led to the excision and circularization of the *EGFR* ecDNA. The gain of the *EGFRvIII* mutation and ecDNA amplification led to the major ecDNA allele we observed. In addition, there was a gain of four additional copies of allele 2 of chromosome 7. These data suggest that the allele that served as the original template for the ecDNA no longer contains the sequence harboring *EGFR* and provide strong evidence for the 'episome model', a model of ecDNA formation in which a genomic locus is excised from chromosomal DNA

as an episome and circularized to form an ecDNA (Fig. 3e) rather than duplication of sequences[22–25].

### Single-molecule DNA methylation profile of isolated ecDNA revealed hypomethylation of gene promoters

We then examined the feasibility of analyzing epigenomic profiles of ecDNA using CRISPR-CATCH. After ecDNA isolation as before, we performed nanopore sequencing to obtain single-molecule sequence information and DNA cytosine methylation (5mC) profiles. We analyzed 5mC-CpG methylation of isolated ecDNA as a proof of concept and observed a strong anti-correlation of 5mC with chromatin accessibility based on bulk assay for transposase-accessible chromatin using sequencing (ATAC-seq), validating the identification of regulatory elements (Methods, Fig. 4a,b and Extended Data Fig. 5a). We also isolated the corresponding *EGFR* chromosomal locus in GBM39 cells and analyzed its DNA methylation profile (Fig. 4a,b). We observed reduced DNA methylation at regulatory elements on ecDNA compared to the same elements on chromosomal DNA, suggesting altered gene regulation (top 50 ATAC-seq peaks; Fig. 4c). The four regions that lost 5mC on ecDNA compared to its chromosomal locus in the same cells were all gene promoters, including that of the *EGFR* oncogene (Methods, Fig. 4d,e and Extended Data Fig. 5b–d). The pattern of hypomethylation corresponded to nucleosome positions shown by micrococcal nuclease digestion with deep sequencing (MNase-seq), implying a more active chromatin state on ecDNA (Fig. 4e and Extended Data Fig. 5d)[26,27]. These hypomethylated sites are located outside the *EGFR* deletion on ecDNAs and therefore cannot be explained by the SV. Finally, single-molecule analysis of enriched ecDNA at the *EGFR* promoter showed hypomethylation at the *EGFR* promoter and co-occurrence of methylation spanning hundreds of CpG sites around the region on the same molecules (285 CpG sites; Fig. 4f). Together, these data show that gene promoters on ecDNA may have increased activities compared to the corresponding chromosomal locus on a single-molecule level and demonstrate that CRISPR-CATCH can be used to measure epigenomic features of ecDNA.

### Mapping of ecDNA amplicon structures resolved heterogeneous SVs and an altered enhancer landscape

Many cancer cells contain ecDNAs with more complex, heterogeneous structures, including multiple sequence rearrangements and more than one circle species[6]. We reasoned that CRISPR-CATCH may provide direct evidence of molecule size and amplicon-phased structural information for these complex amplicons and that this information can be used to computationally reconstruct ecDNA with higher confidence. To this end, we developed an analytical pipeline for amplicon reconstruction from CRISPR-CATCH data (Methods and Fig. 5a). We modified and adopted AmpliconArchitect[6] for generating a copy-number-aware breakpoint graph for each isolated amplicon. Next, we implemented a method for extracting ecDNA candidate paths from the graph, called candidate amplicon path enumerator (CAMPER). Candidate ecDNA structures were generated from the breakpoint graph, estimated multiplicity of genomic segments and molecular size based on PFGE using a depth-first search approach (Methods). Finally, quality estimates of resulting structures were produced for filtering out any low-confidence reconstructions in the case of low-quality gel extractions (for example, incompletely separated ecDNA species) or undetectable breakpoints from sequencing, etc. As validation, we reconstructed the 1.258-Mb circular ecDNA circle encoding *EGFR* in GBM39 cells using this workflow, yielding a structure fully consistent with previous reports using WGS and OM[7,8] (Extended Data Fig. 6). To further demonstrate the utility of this tool, we applied this pipeline to a stomach cancer cell line, SNU16, which contains multiple ecDNA species with *MYC*, *FGFR2* and additional sequences connected by complex structural rearrangements (Extended Data Fig. 7a)[28]. CRISPR-CATCH using guides targeting the *MYC* or *FGFR2* amplicon resulted in multiple visible bands in PFGE (Fig. 5b), revealing extensive molecular heterogeneity of ecDNAs. Gel-extracted ecDNAs

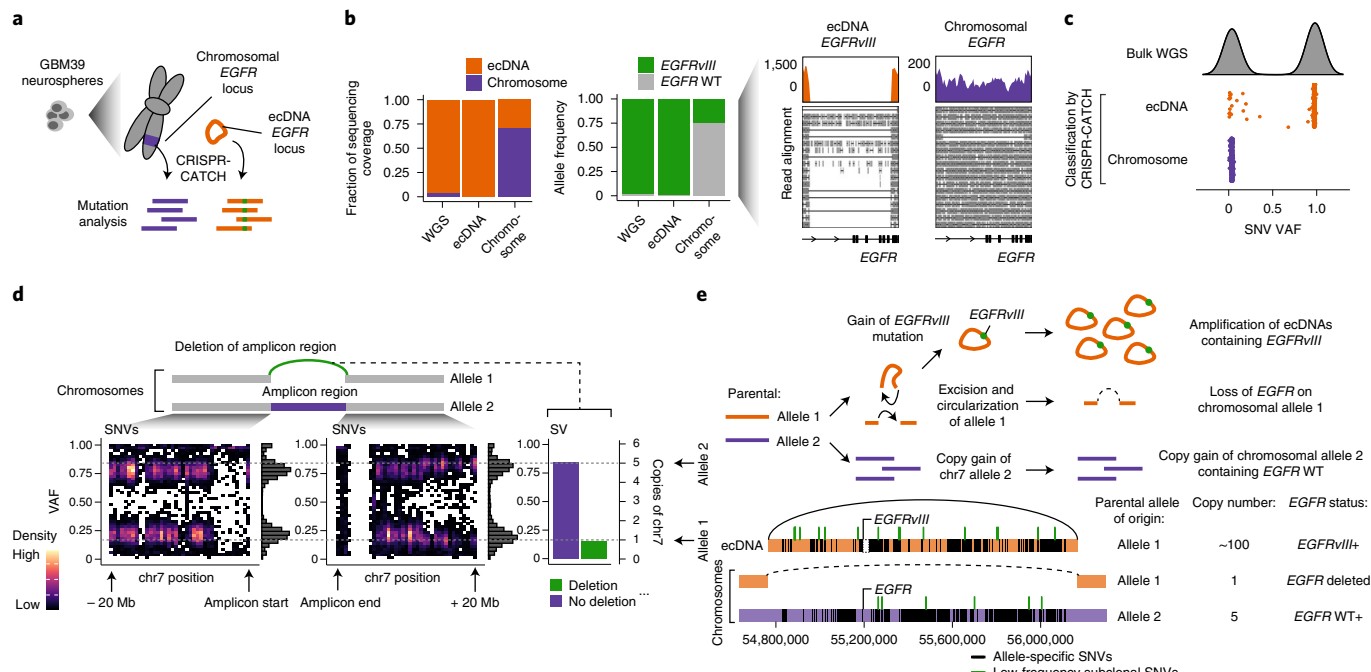

**Fig. 3 | Phasing of SVs and SNVs for ecDNA and its native chromosomal locus identified the chromosomal origin of ecDNA. a**, Isolation of ecDNA and the corresponding chromosomal locus from GBM39 neurospheres by CRISPR-CATCH followed by mutation analysis using short-read sequencing. **b**, Barplot showing relative sequencing coverage of ecDNA (guide A) and chromosomal DNA (guide E + F) (left) and variant allele frequencies (VAFs) of the *EGFRvIII* mutant on ecDNA and chromosomal DNA (middle). Sequencing coverage and junction reads supporting the *EGFRvIII* mutation and wild-type (WT; right). **c**, Bimodal distribution of VAFs of SNVs identified within the ecDNA-amplified region in bulk WGS (top). VAFs of SNVs classified by CRISPR-CATCH as either ecDNA-specific or chromosome-specific (bottom). **d**, Schematic of chromosomes with or without deletion of the ecDNA-amplified region (top). Density plots showing VAFs of non-homozygous SNVs (VAF < 0.99) in WGS 20 Mb upstream or downstream of the region corresponding to the ecDNA amplicon on chromosomes (bottom left). VAF of the SV resulting from deletion of the ecDNA amplicon region and reference sequence without deletion in WGS (bottom right). **e**, Sequence of genomic events leading to ecDNA amplification and chromosome 7 copy gain in GBM39 cells (top). Visualization of all allele-specific genetic variants on ecDNA and chromosomal DNA and their parental alleles of origin identified by CRISPR-CATCH (bottom).

were multiplexed for sequencing. Breakpoint graphs of ecDNA species were greatly simplified by CRISPR-CATCH because each amplicon could be separately reconstructed and was not intermixed with all other amplicons (Extended Data Fig. 7b). In 4 of 23 libraries (bands d,i,m,p; Fig. 5b), short-read sequencing of the CRISPR-CATCH-isolated band was sufficient to enable end-to-end, megabase-scale reconstruction of the ecDNA sequence. Five libraries corresponded to the CZ and showed very low levels of ecDNA enrichment, suggesting that the true ecDNA sizes are smaller than 2.2 Mb (bands a,e,h,o,r; Fig. 5b and Extended Data Fig. 8a). In the remaining cases, large amplicon sequences were enriched, but one or more missing edges prevented unambiguous amplicon resolution (Fig. 5b, c, Extended Data Fig. 8a). From these data, we reconstructed three unique ecDNAs containing *MYC* or *FGFR2*: a 1.604-Mb *FGFR2* ecDNA that was reconstructed from two independent CRISPR-CATCH treatments (using sgRNAs with cut sites >300 kb apart), a smaller *FGFR2* ecDNA species that was 278 kb, and a 622-kb *MYC* ecDNA containing sequences originating from chromosomes 8 and 11 (Fig. 5d–f and Extended Data Fig. 8b). All reconstructions from CRISPR-CATCH data passing quality filters were supported by contigs assembled from OM data (N50 50 Mb) provided to AmpliconReconstructor[7], further validating their structures (Methods, Fig. 5d–f and Extended Data Fig. 8b).

Gene expression is regulated by chromatin interactions between gene promoters and non-coding regulatory elements such as enhancers. Recent studies showed that functional enhancers interacting with oncogenes in *cis* (on the same ecDNA molecule) and in *trans* (between different ecDNA molecules within an ecDNA hub, or between ecDNA and chromosomal loci) shape ecDNA amplicon structure and oncogene expression[28–31]. To identify ecDNA structures containing these enhancers, we performed CRISPR-CATCH using sgRNAs targeting various

enhancers on SNU16 ecDNAs marked by active H3K27ac, BRD4 binding and chromatin accessibility by ATAC-seq and previously identified to modulate *MYC* or *FGFR2* expression via CRISPR interference[28] (Fig. 5c and Extended Data Fig. 8c,d). CRISPR-CATCH enrichment analysis revealed additional ecDNA species showing focal enhancer amplification as well as amplicons containing rearranged enhancers in association with *MYC* and *FGFR2* (Fig. 5c,g,h and Extended Data Fig. 9a). We independently verified instances of *FGFR2* enhancer amplicons lacking the *FGFR2* oncogene-coding sequence using DNA FISH, further supporting our CRISPR-CATCH results (Extended Data Fig. 9b). These findings suggest that extrachromosomal amplification and rearrangement events may be shaped by both enhancer proximity to oncogenes on an ecDNA molecule as well as overall abundance of enhancer sequences in a pool of ecDNA molecules. These focal enhancer amplification events (Fig. 5c,g and Extended Data Fig. 9b), as well as the small ecDNA species containing the *FGFR2* coding sequence but missing its 5′ cognate enhancers (Fig. 5c,f and Extended Data Fig. 9b), suggest ecDNA specialization (Fig. 5h). As ecDNAs can interact with one another in *trans* within a hub[28], amplification of enhancer sequences in a pool of ecDNAs may facilitate intermolecular enhancer-promoter interactions and further increase oncogene expression.

To validate our ecDNA mapping, we compared connected ecDNA segments identified by CRISPR-CATCH with unnormalized background signals in chromatin conformation capture (H3K27ac HiChIP, a protein-directed chromatin conformation capture assay; covalently connected DNA segments have higher frequencies of background interactions than unconnected segments; Methods) and observed a high degree of concordance (Extended Data Fig. 9c,d). In contrast, bulk WGS poorly predicts these ecDNA structures, as shown by low concordance

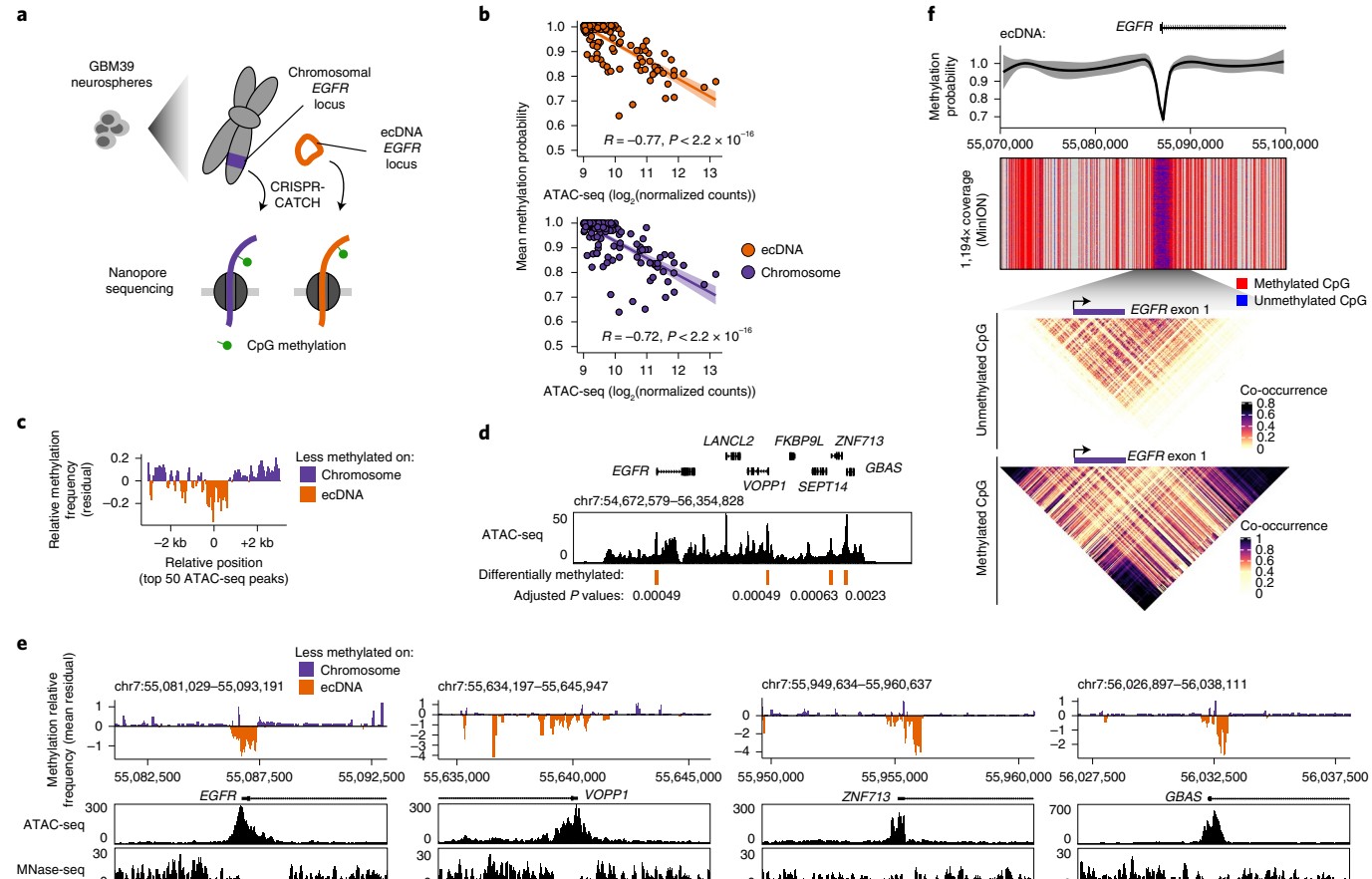

**Fig. 4 | Comparison of CpG methylation statuses of ecDNA and its native chromosomal locus showed hypomethylation of gene promoters on ecDNA. a,** Isolation of ecDNA (guide A) and the corresponding chromosomal locus (guides E + F) from GBM39 neurospheres by CRISPR-CATCH followed by detection of 5mC-CpG methylation by nanopore sequencing. **b,** Negative correlation between mean methylation probabilities of ATAC-seq peaks and their ATAC-seq signals (Pearson's R, two-sided test; error bands represent 95% confidence intervals). **c,** Aggregated levels of relative CpG methylation of ecDNA compared to the chromosomal locus at top 50 ATAC-seq peaks in the ecDNA-amplified region. Mean methylation frequencies were calculated in 100-bp windows sliding every 10 bp. Relative frequencies were quantified from standardized residuals for a linear regression model for mean frequencies on ecDNA vs. chromosomal DNA (Methods). **d,** Bulk ATAC-seq track with differentially methylated regions annotated (Methods; two-sided z-test,

*P* values were Benjamini-Hochberg adjusted; regions with *P* < 0.005 were considered significant). **e,** Relative CpG methylation of ecDNA compared to the chromosomal locus in differential regions and concordance with accessibility by ATAC-seq and nucleosome positioning by MNase-seq. Mean methylation frequencies were calculated in 100-bp windows sliding every 10 bp. Relative frequencies were quantified from standardized residuals for a linear regression model for mean frequencies on ecDNA vs. chromosomal DNA (Methods). **f,** From top to bottom: Loess-smoothed methylation probability around the *EGFR* promoter (error band represents 95% confidence intervals); nanopore sequencing reads showing CpG methylation calls (gray denotes regions with no CpG sites); heatmap showing co-occurrence probabilities of unmethylated CpG sites on the same molecules; heatmap showing co-occurrence probabilities of methylated CpG sites on the same molecules (Methods). Reads were collected using a MinION sequencer (Oxford Nanopore Technologies).

with chromatin conformation capture background signals, demonstrating that WGS provides a collapsed and limited picture of the true diversity of ecDNA structures (Extended Data Fig. 9c,d). Finally, to orthogonally validate ecDNA maps generated by CRISPR-CATCH, we performed dual-color DNA FISH targeting pairs of loci originating from chromosomes 8 and 10 segments on metaphase spreads and confirmed that colocalization of the targeted loci strongly correlated with connected ecDNA segments identified by CRISPR-CATCH (Extended Data Fig. 9c,f). Together, these data demonstrate the utility of CRISPR-CATCH as a method for disambiguating ecDNA structures, particularly when a diverse mixture of ecDNAs is present. This method aids in accurate amplicon mapping and reconstruction orthogonal to contig assembly from bulk DNA and provides insights into the ecDNA structural and regulatory landscape.

## Discussion

By exploiting the distinctive PFGE migration pattern of large circular ecDNA, we show that ecDNA can be isolated from human cancer cells,

including archival patient tumor specimens, and separated by size using CRISPR-CATCH. This method enables targeted analyses of ecDNA sequences and epigenomic features that were previously challenging (Extended Data Fig. 10). CRISPR-CATCH also makes it possible to directly compare ecDNA and the corresponding chromosomal locus in the same cell sample by physically separating them. It is now possible to obtain allele-specific information of ecDNA versus chromosomal DNA without solely relying on SNVs. Furthermore, although ecDNA sequences represent copy-number-amplified genomic regions, we show that VAFs in bulk WGS alone do not accurately reflect the locations of SNVs (for example, a low-frequency SNV can be located on either non-amplified chromosomal DNA or a small subset of ecDNA molecules). In contrast, the ability to phase SNVs by CRISPR-CATCH enables accurate identification of sequencing signal originating from ecDNA to obtain allele-specific information (for example, in bulk ATAC-seq, RNA-sequencing or ChIP-seq data[8,32]). In addition, allele phasing using CRISPR-CATCH led to our discovery of the chromosomal allelic origin of ecDNAs and direct evidence of an excision site. Future systematic

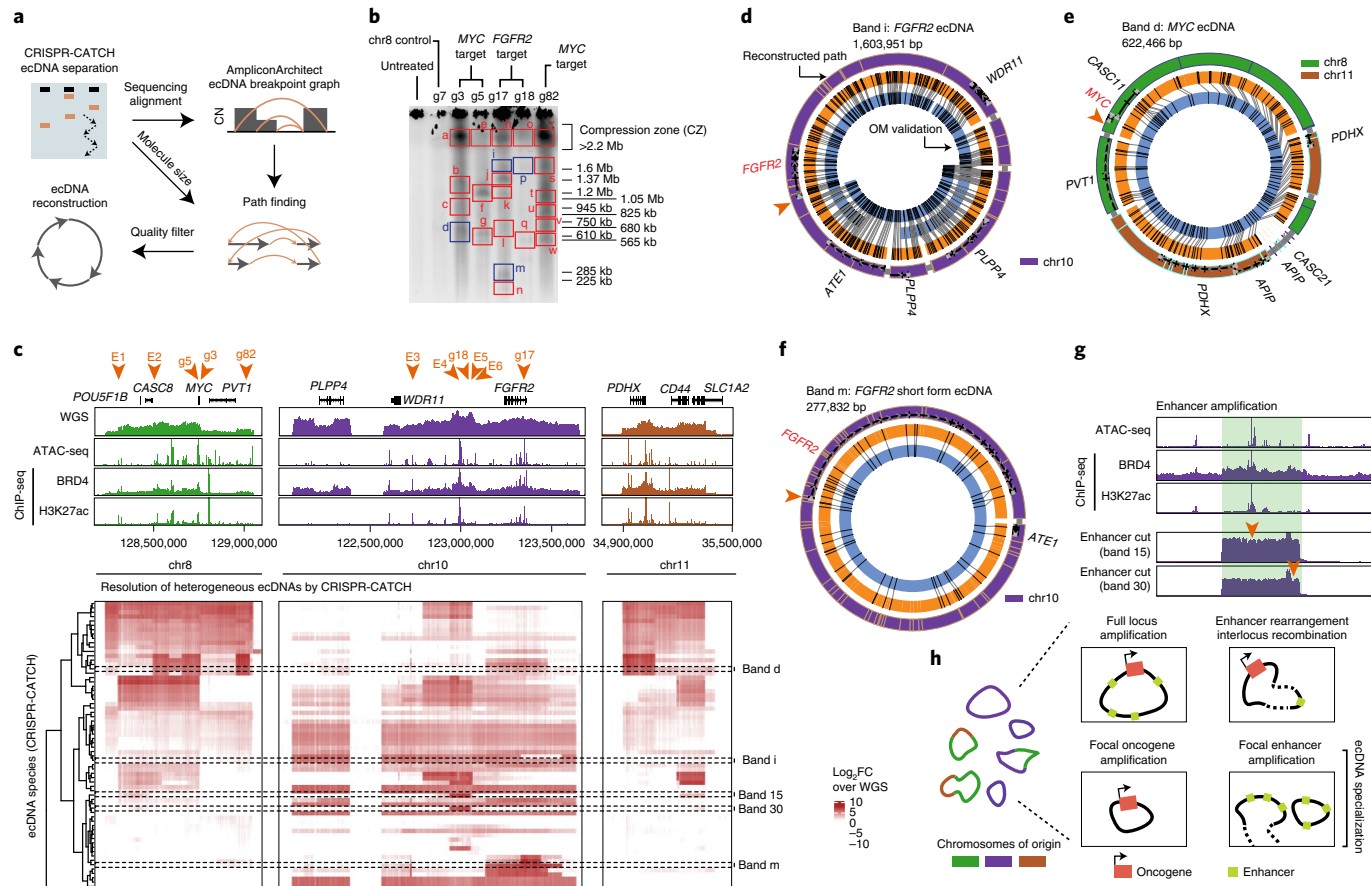

**Fig. 5 | Identification of diverse ecDNA species revealed heterogeneous structural rearrangements and an altered enhancer landscape. a**, Analysis of ecDNA structure using CRISPR-CATCH. ecDNA species are separated by size in PFGE and sequenced. AmpliconArchitect generates CN-aware breakpoint graphs, which are used in combination with molecule sizes from PFGE to find paths and identify candidate ecDNA structures. **b**, PFGE image for SNU16 after treatment with independent sgRNAs targeting either the *FGFR2* or *MYC* locus (guide sequences in Supplementary Table 1). PFGE result is representative of two independent experiments. Bands passing all quality filters for reconstruction are shown in blue. **c**, From top to bottom: WGS, ATAC-seq, BRD4 and H3K27ac chromatin immunoprecipitation with sequencing (ChIP-seq); heatmap showing enrichment of multiple structurally distinct ecDNA species by CRISPR-CATCH. ecDNA species were isolated from bands shown in PFGE gels in panel b and Extended Data Fig. 8c. Orange arrows on the top mark all sgRNA target sites.

**d**–**f**, ecDNA reconstructions using CRISPR-CATCH data (outer rings indicate coordinates along reference genome and gene bodies, and thin gray bands mark connections between sequence segments). OM patterns (orange rings) and assembled contigs (blue rings, contig IDs indicated) validated CRISPR-CATCH reconstructions. Orange arrows mark sgRNA target sites. An *FGFR2* ecDNA structure containing the full amplicon locus was reconstructed from band 'i' as shown in panel d. A *MYC* ecDNA reconstructed from band 'd' containing sequences from chromosomes 8 and 11 is shown in panel e. A short-form *FGFR2* ecDNA reconstructed from band 'm' is shown in panel f. **g**, Sequencing coverage of ecDNAs (bands 15 and 30 correspond to bands extracted from gel in Extended Data Fig. 8c). Region highlighted in green denotes enhancer amplification. ATAC-seq, BRD4 and H3K27ac ChIP-seq show locations of enhancers. Orange arrows mark sgRNA target sites. **h,** Schematic showing diverse ecDNA structures and an altered enhancer landscape revealed by CRISPR-CATCH. FC, fold change.

examination of allelic origins of ecDNAs across different cancers may provide clues about the mechanism of ecDNA genesis.

The scope and challenge of ecDNA isoforms were not fully appreciated in the past. As bulk WGS represents the aggregation of sequencing reads originating from multiple ecDNA species as well as chromosomal DNA, it provides a collapsed and limited picture of the true diversity of ecDNA structures. On the other hand, CRISPR-CATCH enables separation of ecDNAs from the rest of the genome and accurate reconstruction of diverse amplicon structures. Thus, CRISPR-CATCH may be applied to future studies on cancer cells during early formation of ecDNA, to cells evolving under chemotherapeutic or other selective pressures and in other settings where changes in genetic and chromatin features of ecDNA are hypothesized to contribute to cancer cell evolution. As ecDNA often exhibits tremendous structural heterogeneity, CRISPR-CATCH opens up a new window into deciphering intratumoral genetic heterogeneity in cancer. The ability to separate ecDNAs by size may provide increased structural resolution to other types of analysis,

such as single-cell sequencing, in which heterogeneous mixes of ecDNA structures are computationally inferred but difficult to resolve confidently. These future applications of CRISPR-CATCH may also address how ecDNA and chromosomal DNA diverge as they evolve separately and under different kinetics. We note that tandem duplications on chromosomal DNA (for example, homogeneously staining regions) can also be isolated by CRISPR-CATCH with a single guide. In addition, CRISPR-CATCH requires prior knowledge of the ecDNA-amplified genomic locus and therefore should be used to complement additional methods like WGS to identify the amplified genomic locus and/or metaphase FISH to verify the source of isolated DNA. Delivery of multiple sgRNAs targeting various loci may allow multiplexing in the future in cases in which there are multiple distinct ecDNA-amplified loci and/or sample materials are limited.

We demonstrate that CpG methylation can be measured from enriched ecDNA molecules. Past studies have shown that cells containing ecDNA express amplified genes at higher levels than cells

**Technical Report**

containing linear amplifications, and that the ecDNA oncogene locus is more accessible than other loci on linear DNA by bulk ATAC-seq[1,8]. Our comparison of ecDNA versus chromosomal DNA encoding the same gene loci from the same cells showed that gene promoters on circular ecDNA are less methylated than the same promoters on linear chromosomal DNA, suggesting that ecDNA enables more active transcription. In principle, CRISPR-CATCH may be coupled to several genomic assays to understand key chromatin-templated processes on ecDNA such as transcription, DNA replication, and repair[33–35].

Together, we show that ecDNA profiling using CRISPR-CATCH can provide insights into ecDNA structure, diversity, origin and epigenomic landscape. As such, CRISPR-CATCH presents an opportunity for a multitude of molecular studies that will help elucidate how ecDNA oncogene amplifications are regulated in cancer cells.

## Online content

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

¹Center for Personal Dynamic Regulomes, Stanford University, Stanford, CA, USA. ²Bioinformatics and Systems Biology Graduate Program, University of California, San Diego, La Jolla, CA, USA. ³Department of Computer Science and Engineering, University of California, San Diego, La Jolla, CA, USA. ⁴Sarafan ChEM-H, Stanford University, Stanford, CA, USA. ⁵Department of Pathology, Stanford University, Stanford, CA, USA. ⁶Plant Molecular and Cellular Biology Laboratory, Salk Institute for Biological Studies, La Jolla, CA, USA. ⁷Division of Dermatology, Department of Medicine, David Geffen School of Medicine, University of California, Los Angeles, CA, USA. ⁸Cancer Evolution and Genome Instability Laboratory, The Francis Crick Institute, London, UK. ⁹Department of Medicine, Division of Oncology, Stanford University School of Medicine, Stanford, CA, USA. ¹⁰Department of Genetics, Stanford University, Stanford, CA, USA. ¹¹Stanford Cancer Institute, Stanford University School of Medicine, Stanford, CA, USA. ¹²Department of Pediatric Oncology/Hematology, Charité—Universitätsmedizin Berlin, Berlin, Germany. ¹³Cancer Research UK Lung Cancer Centre of Excellence, University College London Cancer Institute, University College London, London, UK. ¹⁴University College London Hospitals NHS Trust, London, UK. ¹⁵Experimental and Clinical Research Center (ECRC), Max Delbrück Center for Molecular Medicine and Charité—Universitätsmedizin Berlin, Berlin, Germany. ¹⁶German Cancer Consortium (DKTK), partner site Berlin, and German Cancer Research Center DKFZ, Heidelberg, Germany. ¹⁷Berlin Institute of Health, Berlin, Germany. ¹⁸Department of Molecular and Medical Pharmacology, David Geffen School of Medicine, University of California, Los Angeles, Los Angeles, California, USA. ¹⁹Jonsson Comprehensive Cancer Center, David Geffen School of Medicine, University of California, Los Angeles, Los Angeles, California, USA. ²⁰Howard Hughes Medical Institute, Stanford University School of Medicine, Stanford, CA, USA. ²¹These authors contributed equally: Jens Luebeck, Siavash R. Dehkordi ✉e-mail: howchang@stanford.edu

## Methods

### Tissue sample collection
Patient tissue sample used in this study was obtained with informed consent and approval by the institutional review boards at the University of California, Los Angeles.

### Cell culture
GBM39 neurospheres were derived from patient tissue as previously described[8] and were authenticated using metaphase DNA FISH with probes hybridizing to *EGFR* as well as a chromosome 7 centromeric probe to confirm ecDNA amplification status. SNU16 cells were obtained from ATCC (CRL-5974). GBM39 cells were maintained in DMEM/Nutrient Mixture F-12 (DMEM/F12 1:1; Gibco, 11320-082), B-27 Supplement (Gibco, 17504044), 1% penicillin-streptomycin (Thermo Fisher, 15140-122), human epidermal growth factor (20 ng ml$^{-1}$; Sigma-Aldrich, E9644), human fibroblast growth factor (20 ng ml$^{-1}$; Peprotech) and heparin (5 µg ml$^{-1}$; Sigma-Aldrich, H3149-500KU). SNU16 cells were maintained in DMEM/F12 supplemented with 10% FBS and 1% penicillin-streptomycin. All cells were cultured at 37 °C with 5% $CO_2$. All cell lines tested negative for mycoplasma contamination.

### WGS
WGS data from bulk GBM39 cells were previously published[8] and raw fastq reads obtained from the National Center for Biotechnology Information (NCBI) Sequence Read Archive under BioProject accession PRJNA506071. Reads were trimmed of adapter content with Trimmomatic[36] (version 0.39), aligned to the hg19 genome using BWA MEM[37] (0.7.17-r1188), and PCR duplicates were removed using Picard's MarkDuplicates (version 2.25.3). WGS data from bulk SNU16 cells were previously generated (SRR530826, Genome Research Foundation).

### Analysis of TCGA ecDNA amplicon sizes
To obtain ecDNA intervals for TCGA tumors, we ran AmpliconClassifier (version 0.4.6; https://github.com/jluebeck/AmpliconClassifier) on AmpliconArchitect outputs published previously using WGS data[2]. ecDNA amplicon sizes were estimated by summing ecDNA amplicon interval sizes for each tumor.

### ecDNA isolation by CRISPR-CATCH
Genomic DNA was embedded in agarose plugs using a modified protocol based on guidelines from the manufacturer of the CHEF Mapper XA System (Bio-Rad Laboratories) as previously described[38]. Briefly, molten 1% certified low-melt agarose (Bio-Rad, 1613112) in PBS was equilibrated to 45 °C. One million cells were pelleted per condition, washed twice with cold 1× PBS, resuspended in 30 µl PBS and briefly heated to 37 °C. Then, 30 µl agarose solution was added to cells, mixed, transferred to a plug mold (Bio-Rad Laboratories, 1703713) and incubated on ice for 10 min. Solid agarose plugs containing cells were ejected into 1.5-ml Eppendorf tubes, suspended in buffer SDE (1% SDS, 25 mM EDTA at pH 8.0) and placed on shaker for 10 min. The buffer was removed and buffer ES (1% *N*-laurolsarcosine sodium salt solution, 25 mM EDTA at pH 8.0, 50 µg ml$^{-1}$ proteinase K) was added. Agarose plugs were incubated in buffer ES at 50 °C overnight. On the following day, proteinase K was inactivated with 25 mM EDTA with 1 mM PMSF for 1 h at room temperature with shaking. Plugs were then treated with RNase A (1 mg ml$^{-1}$) in 25 mM EDTA for 30 min at 37 °C, and washed with 25 mM EDTA with a 5-min incubation. Plugs not directly used for ecDNA enrichment were stored in 25 mM EDTA at 4 °C.

To perform in vitro Cas9 digestion, agarose plugs containing DNA were washed three times with 1× NEBuffer 3.1 (New England BioLabs) with 5-min incubations. Next, DNA was digested in a reaction with 30 nM sgRNA (Synthego) and 30 nM spCas9 (New England BioLabs, M0386S) after pre-incubation of the reaction mix at room temperature for 10 min. To make two cuts on the native chromosomal locus, 15 nM of each sgRNA was added to the reaction. Cas9 digestion was

performed at 37 °C for 4 h, followed by overnight digestion with 3 µl proteinase K (20 mg ml$^{-1}$) in a 200 µl reaction. On the following day, proteinase K was inactivated with 1 mM PMSF for 1 h with shaking. Plugs were then washed with 0.5× TAE buffer three times with 5-min incubations. Plugs were loaded into a 1% certified low-melt agarose gel (Bio-Rad, 1613112) in 0.5× TAE buffer with ladders (CHEF DNA Size Marker, 0.2–2.2 Mb, *Saccharomyces cerevisiae* Ladder: Bio-Rad, 1703605; CHEF DNA Size Marker, 1–3.1 Mb, *Hansenula wingei* Ladder: Bio-Rad, 1703667) and PFGE was performed using the CHEF Mapper XA System (Bio-Rad) according to the manufacturer's instructions and using the following settings: 0.5× TAE running buffer, 14 °C, two-state mode, run time duration of 16 h 39 min, initial switch time of 20.16 s, final switch time of 2 min 55.12 s, gradient of 6 V cm$^{-1}$, included angle of 120° and linear ramping. Gel was stained with 3× Gelred (Biotium) with 0.1 M NaCl on a rocker for 30 min covered from light and imaged. Bands were then extracted and DNA was isolated from agarose blocks using beta-Agarase I (New England BioLabs, M0392L) following the manufacturer's instructions.

To perform CRISPR-CATCH on flash-frozen patient tumor tissues, we removed frozen tissues from −80 °C and incubated them at −20 °C overnight. The tissues were thawed on ice, rinsed with MEM, Hanks' Balanced Salts (Gibco, 11575032) and cut into approximately 5 mm × 5 mm pieces using microdissection scissors. Molten 0.5% certified low-melt agarose (Bio-Rad, 1613112) in 1× PBS was equilibrated to 45 °C, and 50 µl was added to each plug mold (Bio-Rad, 1703713). Each piece of tissue was then suspended into the molten agarose in the plug mold and minced using microdissection scissors. The agarose plug molds were allowed to solidify on ice for 10 min. To dissociate the tissues, agarose-embedded tumors were treated with a mix of 0.1826–1.826 U collagenase (Sigma-Aldrich, C9891), 49.92–124.8 U hyaluronidase (Sigma-Aldrich, H3506) and 1 U dispase (Stem Cell, 07913) in 1 ml MEM at 37 °C for 1 h. Agarose plugs containing tumors were treated with buffer SDE for 10 min as above and buffer ES for 48 h at 50 °C. Plugs were treated PMSF and RNase A and washed with 25 mM EDTA as above. To remove fragmented DNA background in tumor samples via electrodepletion, plugs were loaded into a 1% certified low-melt agarose gel in 0.5× TAE buffer and run in the CHEF Mapper XA System at 14 °C using the following settings: multi-state mode, block 1 with 3 h of constant voltage of 5.2 V cm$^{-1}$ (3 h initial and final switch times, linear ramping, state 1) and included angle of 0°, block 2 with 2 min of constant voltage of 5.2 V cm$^{-1}$ (2 min initial and final switch times, linear ramping, state 1) and included angle of 180°. The gel was removed from the chamber, and agarose plugs trapping intact DNA were carefully removed from the loading wells to avoid breakage. The resulting agarose plugs were then subjected to CRISPR-Cas9 in vitro digestion, PFGE and DNA extraction as described above. All guide sequences are provided in Supplementary Table 1. Unprocessed PFGE images are provided as Source Data.

### In-solution HMW DNA isolation and exonuclease treatment
For comparison between agarose-embedded DNA and in-solution HMW DNA, we performed HMW DNA extraction using the Qiagen MagAttract HMW DNA Kit (67563) following the manufacturer's protocol. To digest linear DNA, we used Plasmid-Safe ATP-Dependent DNase (Biosearch Technologies, E3110K) and performed the reaction according to the manufacturer's protocol over 5 days at 37 °C (1 µl Plasmid-Safe ATP-Dependent DNase, 2 µl 25 mM ATP, 800 ng HMW DNA and 5 µl Plasmid-Safe 10× Reaction Buffer with nuclease-free water to bring up the total reaction volume to 50 µl). After every 24 h, additional enzyme and ATP was added (1 µl Plasmid-Safe ATP-Dependent DNase, 2 µl 25 mM ATP and 0.3 µl Plasmid-Safe 10× Reaction Buffer). After 5 days, DNase was inactivated by a 30-min incubation at 70 °C. To visualize DNA by PFGE, samples were mixed with 1% certified low-melt agarose (Bio-Rad, 1613112) in 0.5× TAE buffer, mixed, transferred to a plug mold (Bio-Rad, 1703713) and incubated on ice for 10 min. Solid agarose plugs

were loaded into a 1% certified low-melt agarose gel (Bio-Rad, 1613112) in 0.5× TAE buffer with ladders (CHEF DNA Size Marker, 0.2–2.2 Mb, *S. cerevisiae* Ladder: Bio-Rad, 1703605; CHEF DNA Size Marker, 1–3.1 Mb, *H. wingei* Ladder: Bio-Rad, 1703667), and PFGE was performed using the CHEF Mapper XA System (Bio-Rad) using the same settings as those used in CRISPR-CATCH experiments described above.

## Hi-C visualization

Hi-C data from the SK-MEL-5 melanoma cell line were obtained from ENCODE (generated by Dekker laboratory) and visualized using the 3D Genome Browser (3dgenome.fsm.northwestern.edu; hg19, raw-rep1)[39,40].

## Metaphase DNA FISH

Cells were arrested at mitosis with 30 ng ml$^{-1}$ KaryoMAX Colcemid Solution in PBS (Gibco) for 18 h. Cells were washed once with PBS and resuspended in 0.075 M KCl at 37 °C for 15–20 min and then fixed in an equal volume of freshly prepared Carnoy's fixative (3:1 methanol/glacial acetic acid, v/v) at room temperature. The cells were washed another three times with fixative, resuspended and dropped onto humidified glass slides. Air-dried samples were washed briefly in 2× SSC buffer (Promega) and then dehydrated in ascending ethanol series (70%, 85% and 100%) each for 2 min. For GBM39 cells, Cytocell *EGFR* amplification Probe (OGT) targeting both *EGFR* and D7Z1 (centromeric probe as a control for chromosome 7) was added to the slide and a coverslip was applied. For SNU16 cells, probes targeting *MYC* (chromosome 8 segment; Empire Genomics, MYC-20-RE), *FGFR2* (Empire Genomics, FGFR2-20-GR), and various chromosome 10 segments from Empire Genomics were used (enhancer region in Extended Data Fig. 9b: WI2-2170K5; probes in Extended Data Fig. 9e targeting region 1: RP11-257O17; region 2: RP11-95I16; region 3: RP11-57H2; region 4: RP11-1024G22). The probes were mixed with the provided hybridization buffer in 1:10 ratio and applied onto the sample. The sample was denatured at 75 °C in a slide moat for 3 min and hybridized overnight at 37 °C in a humidified chamber. The sample was washed in 0.4× SSC for 2 min, followed by another 2-min wash with 2× SSC with 0.1% Tween-20. The sample was stained with 4,6-diamidino-2-phenylindole and washed once in ddH$_2$O before mounted onto a glass slide with ProLong Diamond Antifade Mountant (Invitrogen). Images were acquired on a Leica DMi8 widefield microscope with a ×63 objective.

## Metaphase DNA FISH image analysis

Colocalization analysis for two-color metaphase FISH data for ecDNAs in SNU16 cells described in Extended Data Fig. 9f was performed using Fiji (version 2.1.0/1.53c)[41]. Images were split into the two FISH colors + 4,6-diamidino-2-phenylindole channels, and signal threshold set manually to remove background fluorescence. Overlapping FISH signals were segmented using watershed segmentation. Colocalization was quantified using the ImageJ-Colocalization Threshold program and individual and colocalized FISH signals were counted using particle analysis.

## Short-read sequencing of DNA isolated by CRISPR-CATCH

To perform short-read sequencing on DNA isolated by CRISPR-CATCH, we first transposed it with Tn5 transposase produced as previously described[42] in a 50-µl reaction with TD buffer[43], 10 ng DNA and 1 µl transposase. The reaction was performed at 37 °C for 5 min, and transposed DNA was purified using MinElute PCR Purification Kit (Qiagen, 28006). Libraries were generated by seven to nine rounds of PCR amplification using NEBNext High-Fidelity 2× PCR Master Mix (NEB, M0541L), purified using SPRIselect reagent kit (Beckman Coulter, B23317) with double size selection (0.8× right, 1.2× left) and sequenced on the Illumina Miseq, the Illumina Nextseq 550 or the Illumina NovaSeq 6000 platform. For GBM39 enrichment and mutation analyses in Figs. 1 and 2, a 1.2× left-side selection was performed using SPRIselect. Sequencing data were processed as described above for WGS.

## Genetic variant analyses

SVs from short-read sequencing were identified with DELLY[44] (version 0.8.7; using Boost version 1.74.0 and HTSlib version 1.12) using the delly call command. BCF files were converted to VCF using bcftools view in Samtools[45]. VAFs were calculated using both imprecise and precise variants. Read alignment was visualized using Gviz in R.

SNVs were identified using GATK (version 4.2.0.0)[46] from short-read sequencing data as follows. First, base quality score recalibration was performed on bam files (generated as described above) using gatk BaseRecalibrator followed by gatk ApplyBQSR. Covariates were analyzed using gatk AnalyzeCovariates. SNVs were called using gatk Mutect2 from the recalibrated bam files, and SNVs were filtered using gatk FilterMutectCalls. Finally, VCF files were converted to table format using gatk VariantsToTable with the following parameters: '-F CHROM -F POS -F REF -F ALT -F QUAL -F TYPE -GF AD -GF GQ -GF PL -GF GT'. Mutation VAFs were calculated by dividing alternate allele occurrences by the sum of reference and alternate allele occurrences. SNVs that had coverage depth of 5 or less or were not detected in WGS were filtered out. Read alignment was visualized using Gviz in R. To classify ecDNA-specific SNVs in GBM39 cells, we identified all SNVs with VAFs higher than 0.03 in ecDNAs isolated by CRISPR-CATCH using guide A, B or A + B (given chromosome contamination levels of 0.01–0.02; Extended Data Fig. 2d) and with VAFs in WGS lower than 0.997 (non-homozygous variants). Chromosome-specific SNVs were defined as non-ecDNA SNVs with VAFs in WGS lower than 0.1. Homozygous SNVs were defined as non-ecDNA-specific and non-chromosome-specific SNVs with VAFs in WGS above 0.99.

## Nanopore sequencing and 5mC methylation calling

DNA isolated by CRISPR-CATCH was directly used without amplification for nanopore sequencing. Sequencing libraries were prepared using the Rapid Sequencing Kit (Oxford Nanopore Technologies, SQK-RAD004) according to the manufacturer's instructions. Sequencing was performed on a MinION (Oxford Nanopore Technologies).

Bases were called from fast5 files using guppy (Oxford Nanopore Technologies, version 5.0.16) within Megalodon (version 2.3.3) and DNA methylation status was determined using Rerio basecalling models with the configuration file 'res_dna_r941_min_modbases-all-context_v001.cfg' and the following parameters: '–outputs basecalls mod_basecalls mappings mod_mappings mods per_read_mods –mod-motif Z CG 0 –write-mods-text –mod-output-formats bedmethyl wiggle –mod-map-emulate-bisulfite –mod-map-base-conv C T –mod-map-base-conv Z C'. Methylation calls on single molecules were visualized using Integrative Genome Viewer (IGV, version 2.11.1) in bisulfite mode.

To quantify 5mC-CpG methylation levels across an entire locus, rolling averages of CpG methylation percentages were calculated using a window of 100 bp sliding every 10 bp (unless otherwise specified). Rolling averages of ecDNA and the native chromosomal locus were linearly regressed using the lm function in R. Standardized residual for the linear regression for each window was calculated using the rstandard function to represent relative methylation frequencies on ecDNA compared to chromosomal DNA. To identify accessible regions which are differentially methylated on ecDNA, we first filtered on ATAC-seq peaks which had log-normalized coverage above 9 (calculated by DESeq2 as described in the ATAC-seq section below; normalized coverage for each peak was divided by peak width after adding 1, scaled to 500 and $\log_2$ transformed). Next, methylation sites with coverage above 5 for both the isolated ecDNA and chromosomal locus, and overlapping filtered ATAC-seq peaks were linearly regressed using the lm function in R. Standardized residual for the linear regression for each CpG site was calculated using the rstandard function. For each ATAC-seq peak, a z score was calculated using the formula $z = (x - m)/s.e.$, where x is the mean CpG residual within the peak, m is the mean residual of all CpG sites and s.e. is the standard error calculated from the standard

deviation of all CpG sites divided by the square root of the number of CpG sites within the peak. z scores were used to compute two-sided $P$ values using the normal distribution function, which were adjusted with p.adjust in R (version 3.6.1) using the Benjamini–Hochberg procedure.

To quantify co-occurrence of methylated or unmethylated CpGs on single molecules, methylation calls on the '+' strand were offset by 1 bp to match the locations of the corresponding CpG sites on the '−' strand. CpG sites where the base probabilities of methylation were above 0.7 were categorized as methylated, and sites where the base probabilities of unmodified CpG were above 0.7 were categorized as unmethylated. For each pair of CpG sites, co-occurrence was calculated by number of co-occurrences of methylated or unmethylated CpGs on the same nanopore sequencing reads divided by total number of occurrences in which the two CpG sites can be successfully categorized as either methylated or unmethylated.

### ATAC-seq

ATAC-seq data for GBM39 were previously published[8] and raw fastq reads obtained from the NCBI Sequence Read Archive, under BioProject accession PRJNA506071. ATAC-seq data for SNU16 were previously published under Gene Expression Omnibus accession GSE159986 (ref. [28]). Adapter-trimmed reads were aligned to the hg19 genome using Bowtie2 (2.1.0). Aligned reads were filtered for quality using samtools (version 1.9)[45], duplicate fragments were removed using Picard's MarkDuplicates (version 2.25.3) and peaks were called using MACS2 (version 2.2.7.1)[47] with a q-value cut-off of 0.01 and a no-shift model. Peaks from replicates were merged, and read counts were obtained using bedtools (version 2.30.0)[48] and normalized using DESeq2 (using the 'counts' function in DESeq2 with normalized = TRUE; version 1.26.0)[49].

### MNase-seq

MNase-seq data for GBM39 were previously published[8] and raw fastq reads obtained from the NCBI Sequence Read Archive under BioProject accession PRJNA506071. Reads were trimmed of adapter content with Trimmomatic[36] (version 0.39), aligned to the hg19 genome using BWA MEM[37] (0.7.17-r1188), and PCR duplicates removed using Picard's MarkDuplicates (version 2.25.3). Coverage of nucleosome midpoints was obtained using bamCoverage from deepTools (version 3.5.1) with the following parameters: '–MNase –binSize 1'.

### Reporting summary

Further information on research design is available in the Nature Research Reporting Summary linked to this article.

## Data availability

Sequencing data generated in this study are deposited in the Sequence Read Archive under BioProject accession PRJNA777710. WGS data from bulk GBM39 cells were obtained from the NCBI Sequence Read Archive under BioProject accession PRJNA506071. WGS data from bulk SNU16 cells were previously generated (SRR530826, Genome Research Foundation). ATAC-seq and MNase-seq data for GBM39 were obtained from the NCBI Sequence Read Archive under BioProject accession PRJNA506071. ChIP-seq data for SNU16 were previously published under Gene Expression Omnibus accession GSE15998628. Sequencing reads were mapped to the hg19 human reference genome. Source data are provided with this paper.

## Code availability

Custom code to perform reconstructions of candidate ecDNA structures from CRISPR-CATCH data is available at https://github.com/siavashre/CRISPRCATCH.

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

## Acknowledgements

We thank members of the Chang and Bafna laboratories for discussions. This work was supported by National Institutes of Health (NIH) grants R35-CA209919 (H.Y.C.) and RM1-HG007735 (H.Y.C., W.J.G. and C.C.); Cancer Grand Challenges grant CGCSDF-2021\100007, with support from Cancer Research UK and the National Cancer Institute (H.Y.C., V.B., P.S.M., M.J.-H. and A.G.H.); NIH grants U24CA264379, R01GM114362, OT2CA278635 and Cancer Grand Challenges grant CGCATF-2021/100025 with support from Cancer Research UK and the National Cancer Institute (V.B.); NIH grants 1R01CA176111A1 and 1P01CA168585 (R.S.L.); the Melanoma Research Alliance (R.S.L. and G.M.); and a Jonsson Comprehensive Cancer Center postdoctoral fellowship (P.D.). K.L.H. was supported by a Stanford Graduate Fellowship and an NCI Predoctoral to Postdoctoral Fellow Transition Award (NIH F99CA274692). H.Y.C. is an Investigator of the Howard Hughes Medical Institute. A.G.H. is supported by Deutsche Forschungsgemeinschaft (German Research Foundation) grant 398299703 and the European Research Council under the European Union's Horizon 2020 Research and Innovation Programme (grant agreement 949172). K.E.H. was supported by a Canadian Institutes of Health Research Banting postdoctoral fellowship.

## Author contributions

K.L.H. and H.Y.C. conceived the project. K.L.H. performed experiments for CRISPR-CATCH method development for ecDNA enrichment and analyses of genetic variants and epigenomic features from short-read sequencing and nanopore sequencing data. J.L. and S.R.D. analyzed short-read sequencing and OM data for amplicon reconstruction. K.L.H., C.I.C. and R.L. performed experiments for CRISPR-CATCH optimization for human tumor processing. I.T.L.W. performed DNA

FISH validation experiments. P.D., S.H.L., N.E.W., G.M., X.Z., C.B., K.E.H., W.Y., R.C., C.S., C.C., M.J.-H., A.G.H. and R.S.L. provided human tumor specimens and patient-derived samples for CRISPR-CATCH optimization. P.D. and R.S.L. analyzed ecDNA amplicons in human tumor specimens. C.C. and J.A.L. generated OM data and provided de novo assembly and rare variant analysis results. W.J.G. advised on single-molecule sequencing. P.M., V.B. and H.Y.C. guided data analysis and provided feedback on experimental design. K.L.H. and H.Y.C. wrote the manuscript with input from all authors.

## Competing interests

H.Y.C. is a co-founder of Accent Therapeutics, Boundless Bio, Cartography Biosciences and Orbital Therapeutics, and an advisor of 10x Genomics, Arsenal Biosciences and Spring Discovery. V.B. is a co-founder, paid consultant and science advisory board member and has equity interest in Boundless Bio and Abterra. The terms of this arrangement have been reviewed and approved by the University of California, San Diego in accordance with its conflict-of-interest policies. P.M. is a co-founder and advisor of Boundless Bio. R.S.L. reports research and clinical trial support from Merck, Pfizer, BMS and OncoSec. J.L. receives compensation as a consultant for Boundless Bio. The remaining authors declare no competing interests.

## Additional information

**Extended data** is available for this paper at https://doi.org/10.1038/s41588-022-01190-0.

**Correspondence and requests for materials** should be addressed to Howard Y. Chang.

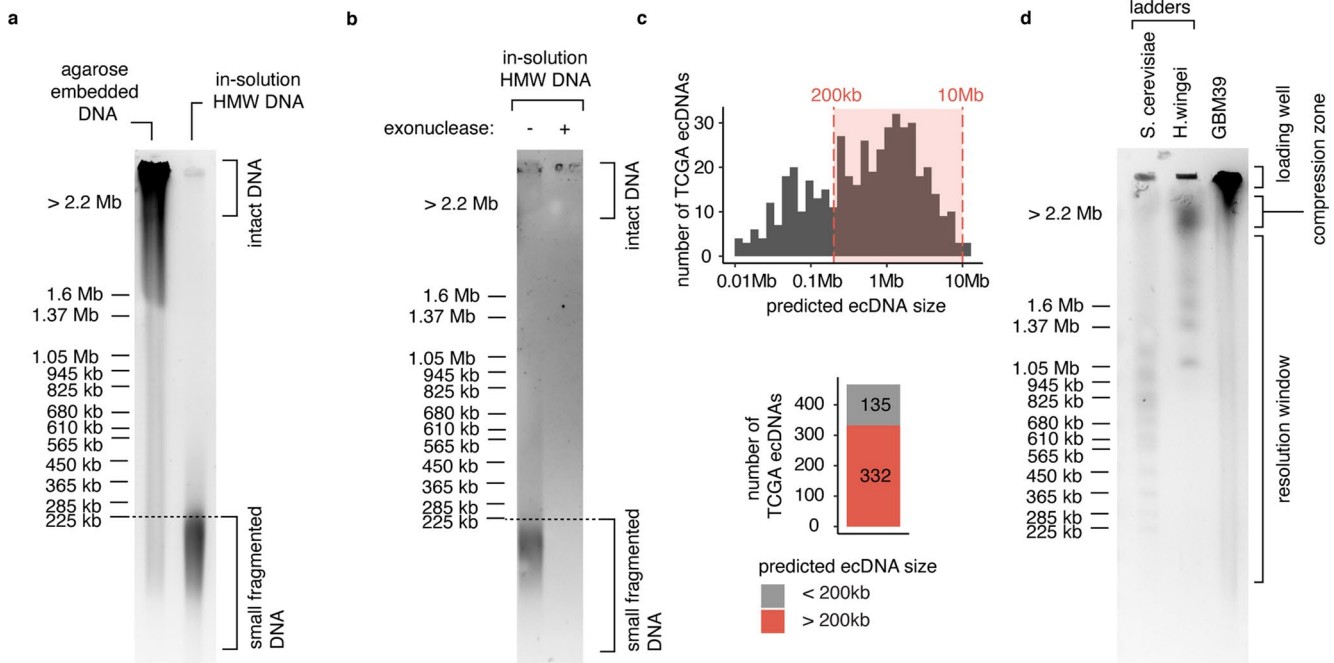

**Extended Data Fig. 1 | Agarose entrapment of genomic DNA preserves intact ecDNA. (a)** A PFGE image showing DNA fragmentation after in-solution HMW DNA isolation as compared with intact agarose-embedded DNA trapped in the loading well. DNA fragmentation was reproduced in two independent experiments. **(b)** A PFGE image showing complete digestion of fragmented in-solution HMW DNA after a 5-day exonuclease treatment. One independent experiment was performed. **(c)** Analysis of ecDNA amplicon sizes predicted by AmpliconArchitect in TCGA tumor samples. **(d)** A PFGE image showing size ladders and GBM39 ultrahigh-molecular weight (UHMW) genomic DNA without *in vitro* CRISPR-Cas9 linearization (representative of three independent experiments). UHMW DNA was trapped in the loading well and the upper compression zone.

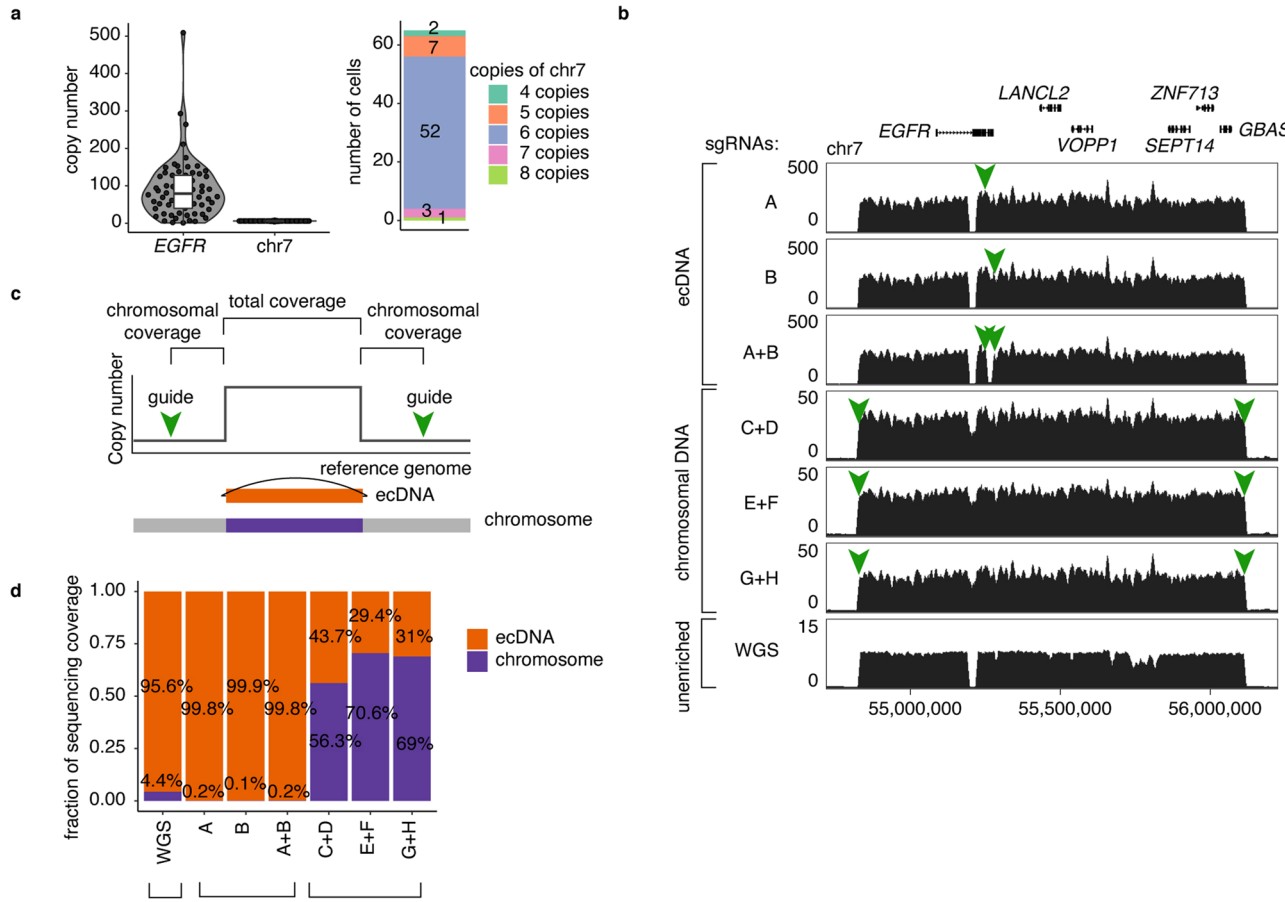

**Extended Data Fig. 2 | Enrichment of circular ecDNA by CRISPR-CATCH. (a)**
Left: quantification of *EGFR* and chromosome 7 copy numbers in GBM39 cells
using DNA FISH on metaphase spreads (n = 65 cells; box center line, median; box
limits, upper and lower quartiles; box whiskers, 1.5× interquartile range). Right:
number of GBM39 cells with 4, 5, 6, 7, or 8 copies of chromosome 7. An example
FISH image is shown in Fig. 1b. **(b)** Full sequencing tracks showing coverage
for isolated ecDNA and its chromosomal locus at the *EGFR* amplified region
compared to WGS. Zoomed-in tracks are shown in Fig. 1f. Orange arrows indicate
locations of sgRNA targets. **(c)** Chromosomal overhangs from chromosome-
targeting guides (guides C-H) outside of the ecDNA-amplified region were
used for calculating sequencing coverage of the chromosomal allele. The mean
coverage of the 5' and 3' chromosomal overhangs was calculated. The coverage
of ecDNA alleles was calculated by subtracting chromosomal coverage from total
coverage in the ecDNA-amplified region. **(d)** Relative sequencing coverage of
chromosomal DNA and ecDNA alleles in WGS or CRISPR-CATCH samples.

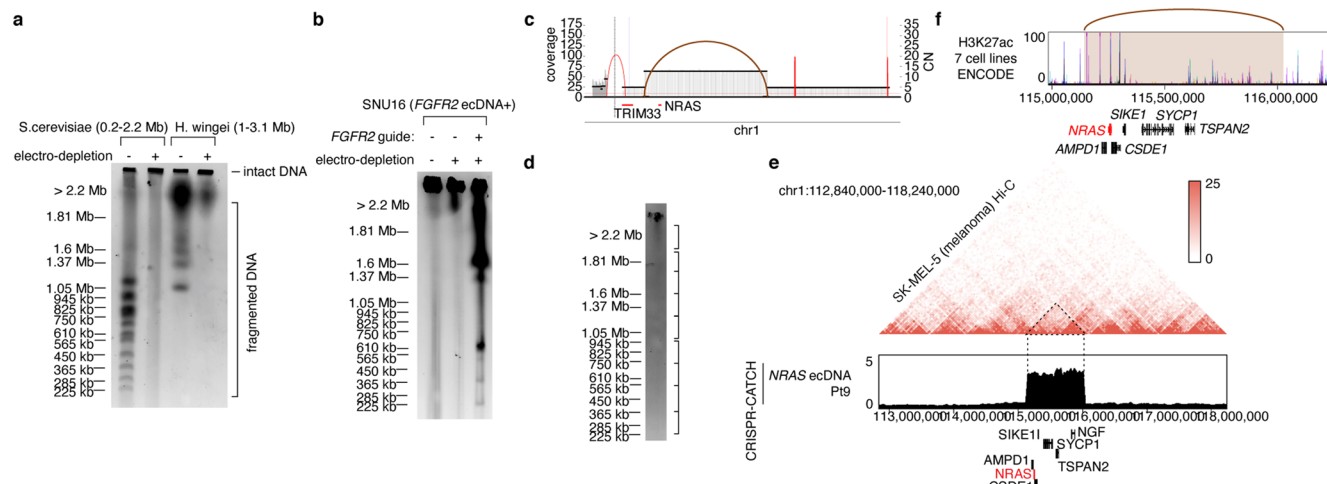

**Extended Data Fig. 3 | Tumor processing and ecDNA enrichment from patient tumor samples using CRISPR-CATCH. (a)** A PFGE image showing presence of DNA bands from S. cerevisiae and H. wingei DNA size markers with or without electrodepletion. One independent experiment was performed. **(b)** A PFGE image showing linearized ecDNA molecules from SNU16 cells containing *FGFR2* ecDNAs after electrodepletion and treatment with an *FGFR2* guide (guide 17; guide sequence in Supplementary Table 1). One independent experiment was performed. **(c)** AmpliconArchitect breakpoint graph from bulk WGS of melanoma patient tumor Pt9 showing amplification of *NRAS*. **(d)** A PFGE image from melanoma patient sample Pt9 after electrodepletion and CRISPR-CATCH

using *NRAS*-targeting guide 194 (guide sequence in Supplementary Table 1). Brackets on the right correspond to gel-extracted regions shown in Fig. 2c. One independent experiment was performed. **(e)** Top: raw Hi-C contact heatmap for the SK-MEL-5 melanoma cell line (40-kb resolution). Bottom: sequencing track showing CRISPR-CATCH-enrichment of the *NRAS* ecDNA from melanoma patient tumor Pt9. **(f)** Layered H3K27ac ChIP-seq tracks from 7 cell lines (GM12878, H1-hESC, HSMM, HUVEC, K562, NHEK, NHLF) in ENCODE using the UCSC Genome Browser. Brown arc marks ecDNA breakpoints. Shaded brown region marks the *NRAS* ecDNA amplicon detected in patient sample Pt9.

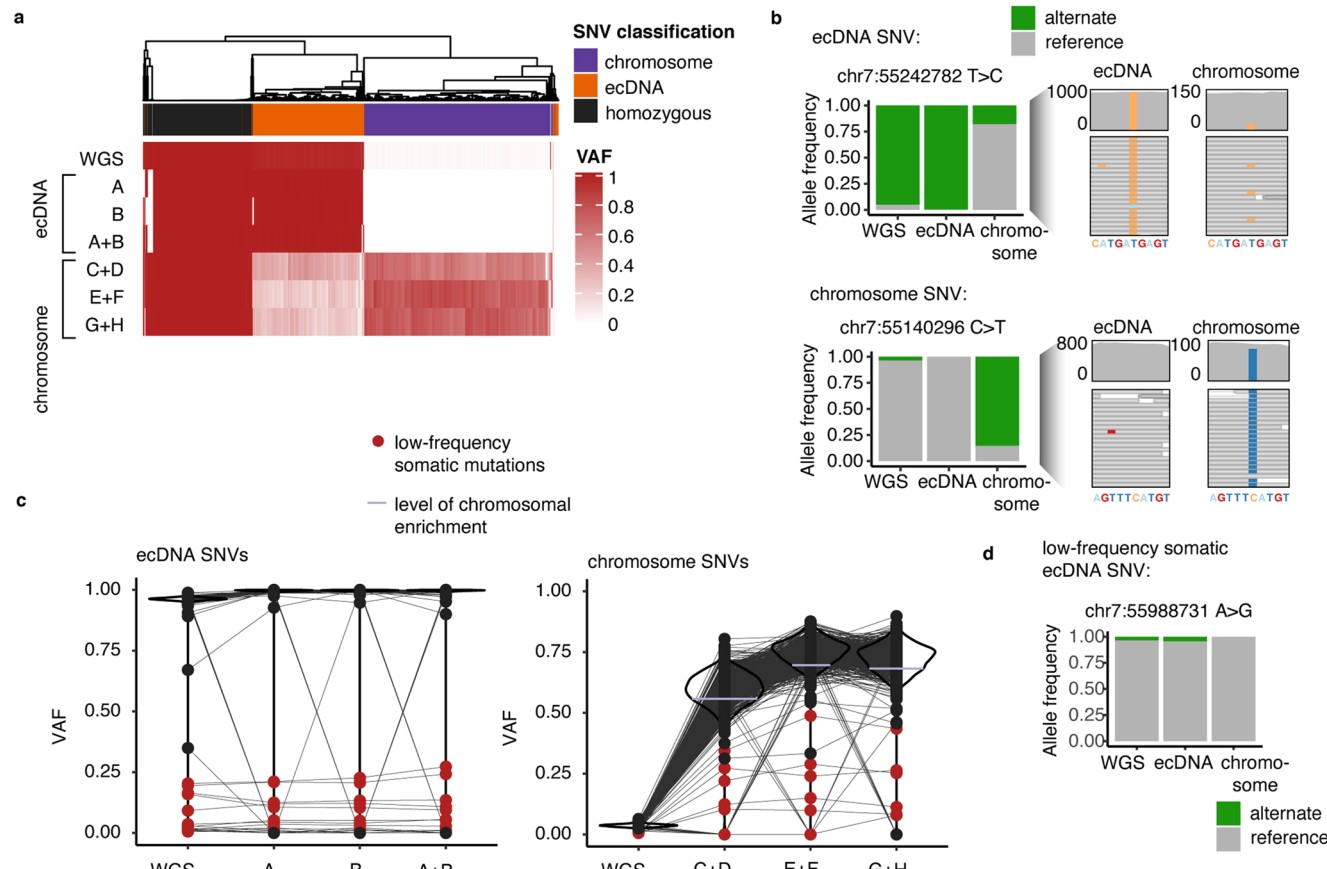

**Extended Data Fig. 4 | Phasing of SNVs for ecDNA and its native chromosomal locus. (a)** VAFs of SNVs identified in the ecDNA-amplified region and its native chromosomal locus in various CRISPR-CATCH treatments. Letters denote sgRNAs used (A-H). **(b)** Left: VAFs of two ecDNA- and chromosome-specific SNVs in WGS, isolated ecDNA or chromosomal molecules using CRISPR-CATCH. Right: sequencing reads supporting SNV identification. **(c)** VAFs of SNVs classified as ecDNA- or chromosome-specific SNVs in various CRISPR-CATCH treatments.

Black lines connect identical SNVs detected in WGS and indicated CRISPR-CATCH treatments. Low-frequency allele-specific SNVs are defined as SNVs with VAFs < 0.5 and are marked in red. Horizontal lines in lilac in the chromosome SNV plot represent levels of chromosomal enrichment corresponding to Extended Data Fig. 2d. **(d)** VAFs of a low-frequency somatic ecDNA SNV in WGS, isolated ecDNA or chromosomal molecules using CRISPR-CATCH.

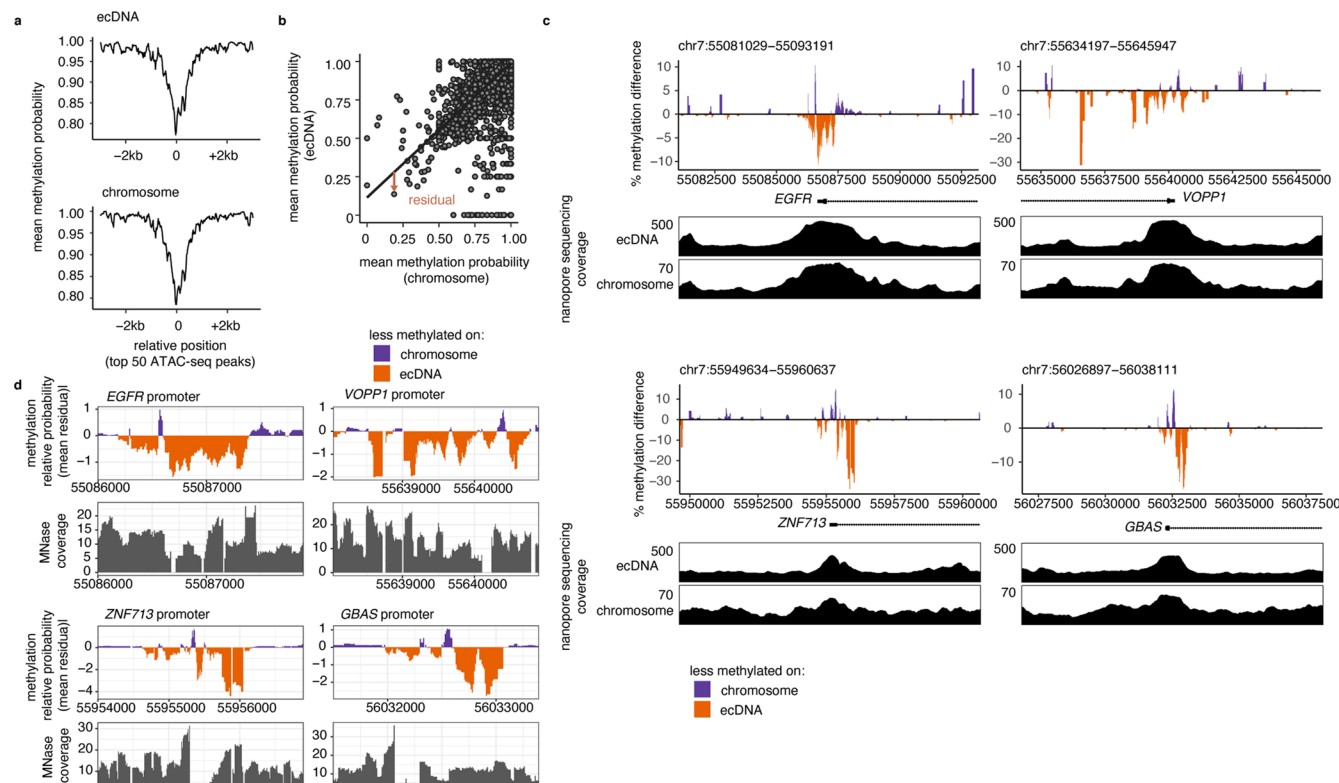

**Extended Data Fig. 5 | Quantification of 5mC-CpG methylation probability of ecDNA and the native chromosomal locus. (a)** Aggregated CpG methylation probability of ecDNA and chromosomal DNA at the top 50 ATAC-seq peaks with highest coverage in the amplified region. Mean methylation frequencies were calculated in 100-bp windows sliding every 10 bp. **(b)** Linear regression model of mean methylation probabilities of ecDNA vs chromosomal DNA. Mean methylation probabilities were calculated in 100-bp windows sliding every 10 bp in the ecDNA-amplified region. Each point represents a window mean. Brown arrow demonstrates the standardized residual of a data point from the regression line. **(c)** Relative CpG methylation of ecDNA compared to the chromosomal locus in differential regions shown as absolute differences in methylation frequencies.

Regions shown correspond to differentially methylated regions in Fig. 4d,e. Mean methylation frequencies were calculated in 100-bp windows sliding every 10 bp. Normalized sequencing coverage tracks are shown on the bottom of each plot. **(d)** Relative CpG methylation of ecDNA compared to the chromosomal locus and nucleosome positioning by MNase-seq, zooming into indicated gene promoters. Regions shown correspond to differentially methylated regions in Fig. 4d,e. Mean methylation frequencies and MNase-seq coverage were calculated in 100-bp windows sliding every 10 bp. Relative frequencies were quantified from standardized residuals for a linear regression model for mean frequencies on ecDNA vs chromosomal DNA (Methods).

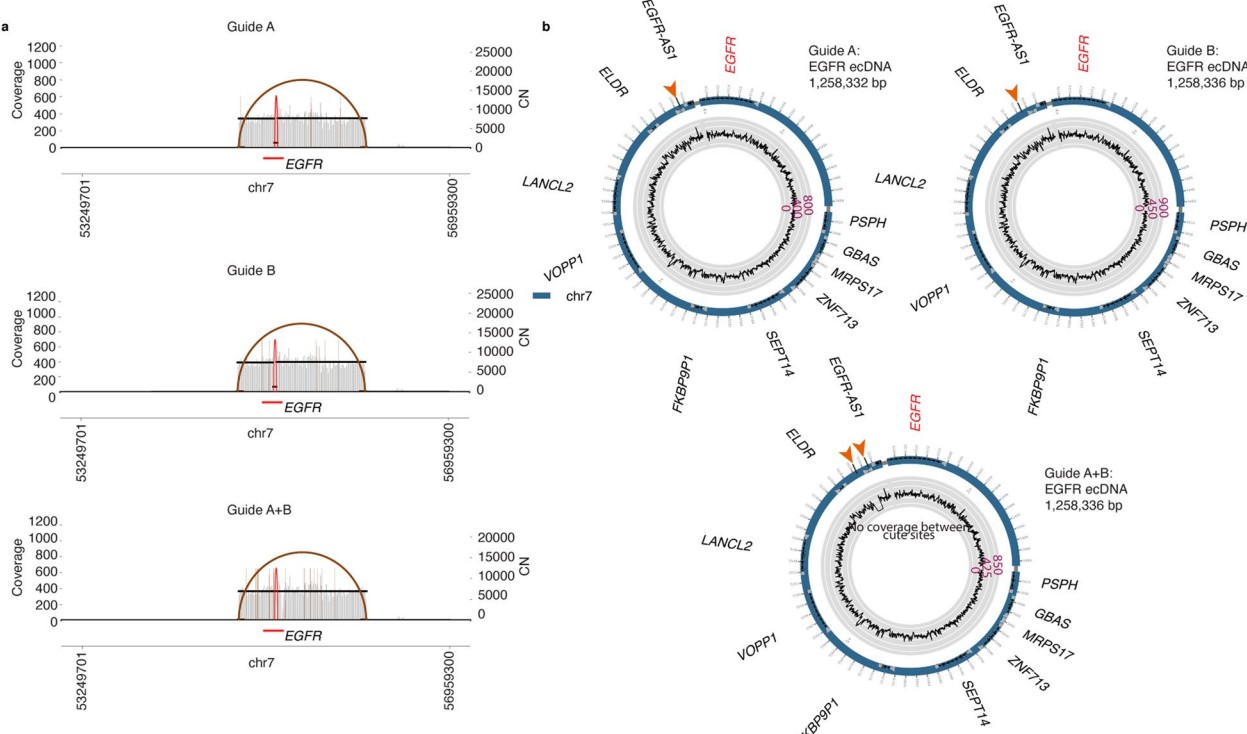

**Extended Data Fig. 6 | Reconstruction of a 1.258 Mb ecDNA from GBM39 neurospheres. (a)** AmpliconArchitect breakpoint graphs for CRISPR-CATCH-isolated ecDNAs using guides A and/or B as in Fig. 1 (guide sequences in Supplementary Table 1). **(b)** Reconstructed ecDNA circles from CRISPR-CATCH data using independent sgRNAs showing equivalent ecDNA structures (outer rings; thin gray bands mark connections between sequence segments). Sequencing coverage is shown along the reconstructed circle (inner rings). Orange arrows mark sgRNA target sites. Coordinate tick marks are printed in 10-kb units. AmpliconArchitect segment IDs and orientations are annotated.

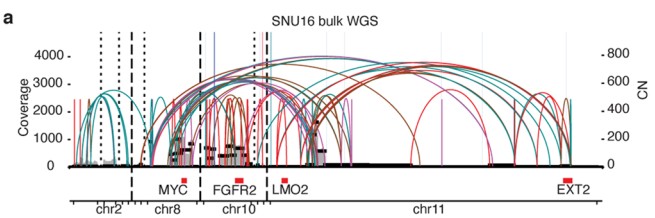

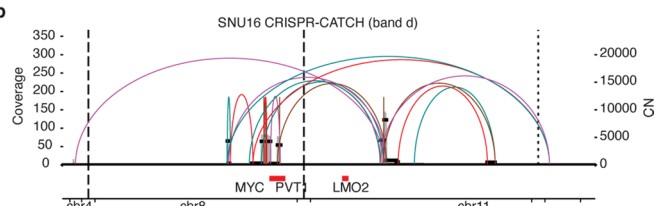

**Extended Data Fig. 7 | CRISPR-CATCH enables disambiguation of heterogeneous structural rearrangements on individual ecDNA species. (a)** AmpliconArchitect breakpoint graph from bulk WGS of stomach cancer SNU16 cells showing significantly amplified sequences from chromosomes 8, 10, and 11. **(b**) An example of an AmpliconArchitect breakpoint graph for a CRISPR-CATCH-separated ecDNA species (band 'd') from SNU16 cells showing greatly simplified breakpoints connecting only sequences from chromosomes 8 and 11. Gray vertical lines represent genomic coverage from WGS data and black horizontal lines indicate the estimated copy number of the region. Colored arcs represent breakpoint junctions, and the orientation of those junctions is specified by the color. Red and brown arcs preserve the orientation of the genome, with red reflecting breakpoints supported by reads in the proper orientation and brown reflecting breakpoints supported by reads in the everted orientation. Teal and magenta arcs indicate breakpoints leading to a change in genome orientation before and after the breakpoint where teal breakpoints are supported by both paired-end reads mapping to the forward strand and magenta breakpoints are supported by both paired-end reads mapping to the reverse strand.

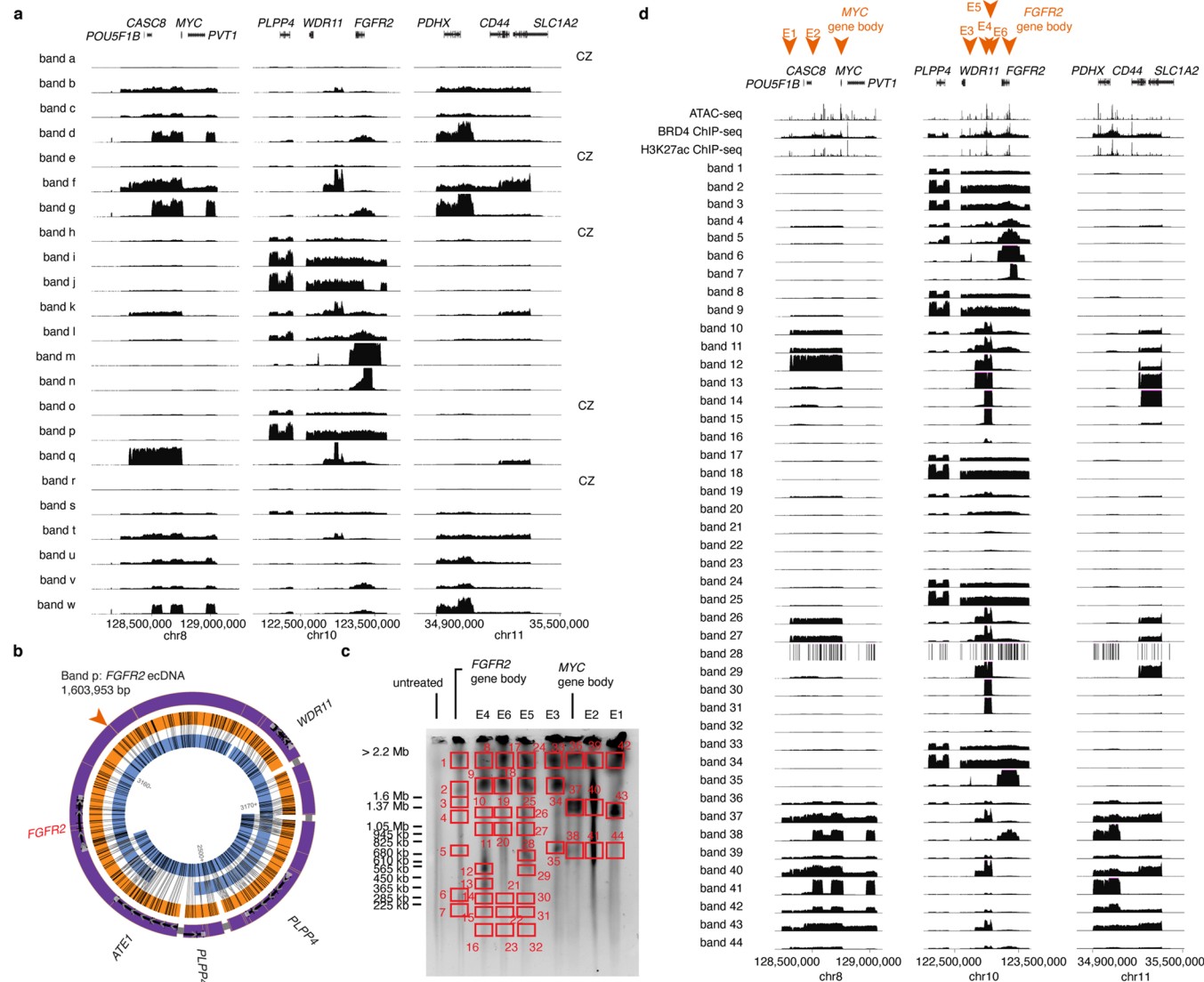

**Extended Data Fig. 8 | Enrichment of multiple ecDNA species from the SNU16 stomach cancer cell line. (a)** Sequencing coverage of multiple ecDNA species from SNU16 cells after CRISPR-CATCH isolation at the *FGFR2*, *MYC* and *CD44* loci. Bands a-w correspond to extracted bands shown in Fig. 5b. Bands corresponding to unresolved DNA content in the compression zone are labeled CZ. **(b)** ecDNA reconstruction using CRISPR-CATCH data (outer rings; thin gray bands mark connections between sequence segments). Optical mapping patterns (orange rings) and assembled contigs (blue rings, contig IDs indicated) validated CRISPR-CATCH reconstructions. Orange arrow marks sgRNA target site. Shown is an *FGFR2* ecDNA structure reconstructed from band 'p', equivalent to that reconstructed from band 'i' (Fig. 5d) using an independent sgRNA. **(c)** PFGE image for SNU16 after treatment with independent sgRNAs targeting either the *FGFR2* or *MYC* gene bodies or enhancers (*FGFR2* gene body: guide 17; *MYC* gene body: guide 5; guide sequences in Supplementary Table 1). One independent experiment was performed. **(d)** Short-read sequencing coverage tracks of multiple ecDNA species from SNU16 cells after CRISPR-CATCH isolation at the *FGFR2*, *MYC* and *CD44* loci. Bands 1–44 correspond to extracted bands shown in **c**. Orange arrows mark sgRNA target sites.

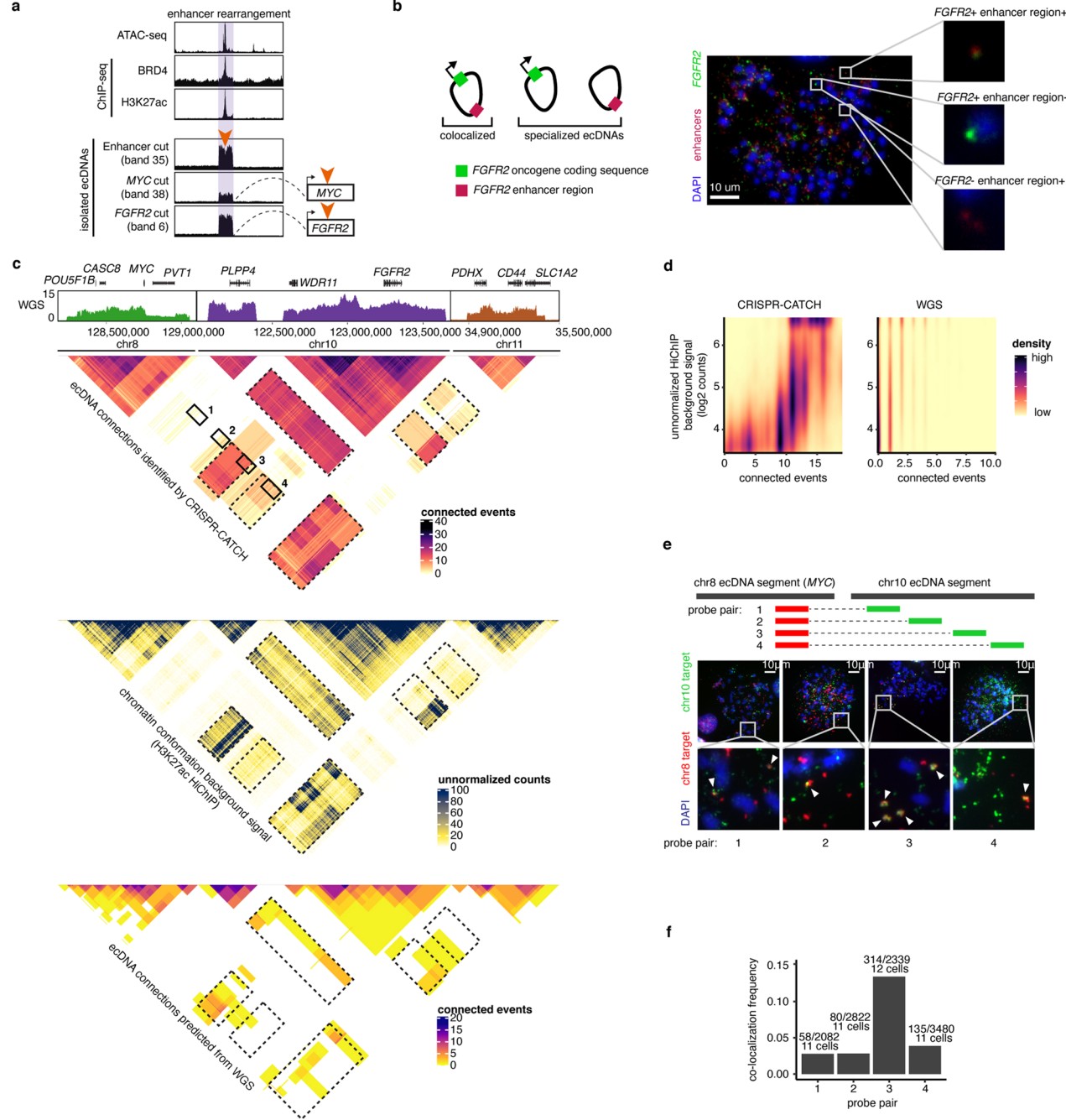

**Extended Data Fig. 9 | Validation of ecDNA species in SNU16 cells mapped by CRISPR-CATCH. (a)** Sequencing coverage of ecDNAs isolated from SNU16 cells (bands extracted from gel in Extended Data Figure 8c). Region highlighted in purple is connected to *MYC* or *FGFR2*. ATAC-seq, BRD4 and H3K27ac ChIP-seq show that an enhancer is located in the rearranged region. Orange arrows mark sgRNA target sites. **(b)** Left: ecDNA species targeted by dual-color FISH. Right: representative two-color DNA FISH image on a metaphase spread showing instances of specialized ecDNAs containing either *FGFR2* (green) or the enhancer region (red, identified in Fig. 5g, chr10:122988480–123026871), as well as ecDNA species with colocalized oncogene and enhancers (n = 69 cells). **(c)** From top to bottom: WGS coverage of ecDNA-amplified regions; connected DNA segments on ecDNAs identified by CRISPR-CATCH (boxes 1–4 mark coordinates targeted by pairs of FISH probes in panel **e** and **f**); unnormalized background signals from chromatin conformation capture using H3K27ac HiChIP; connected DNA segments predicted from WGS data using AmpliconArchitect. **(d)** Levels

of unnormalized HiChIP interactions between inter-chromosomal DNA segments and their co-occurrence on ecDNA as identified by CRISPR-CATCH compared to WGS. Connected ecDNA segments identified by CRISPR-CATCH were strongly supported by HiChIP signals. **(e)** Top: FISH probes targeting either the chromosome 8 or 10 segment located on ecDNAs in SNU16 cells. Bottom: representative two-color DNA FISH images on metaphase spreads for quantifying colocalization of the chromosome 8 and 10 ecDNA segments marked in the CRISPR-CATCH heatmap in **c** (regions 1–4). Red DNA FISH probe targets *MYC*. Green DNA FISH probes target the following: region 1, chr10:122309127–122477445 (n = 11 cells); region 2, chr10:122635712–122782544 (n = 11 cells); region 3, chr10:122973293–123129601 (n = 12 cells); region 4, chr10:123300005–123474433 (n = 11 cells). **(f)** Frequencies of red-green colocalized FISH signals (probe pairs 1–4 correspond to regions targeted in **e**). The number of colocalized over total signals and the number of cells assessed are shown above each bar.

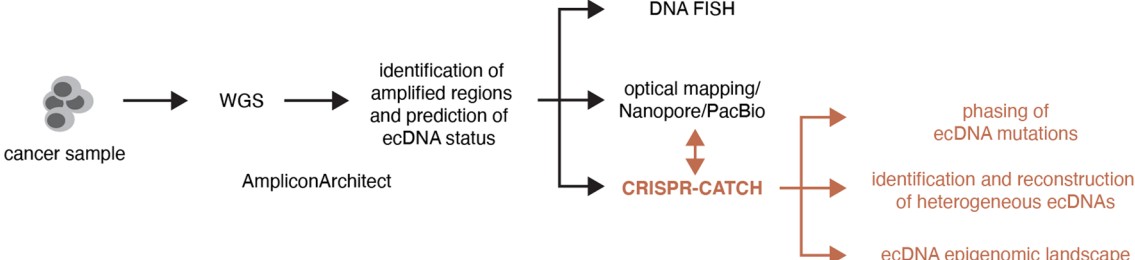

**Extended Data Fig. 10 | Recommended usage of CRISPR-CATCH.** A recommended workflow for using CRISPR-CATCH in complement to WGS, DNA FISH and optical mapping for analysis of ecDNAs in cancer samples.

# Reporting Summary

## Statistics

For all statistical analyses, confirm that the following items are present in the figure legend, table legend, main text, or Methods section.

| n/a | Confirmed | |
|---|---|---|
| ☐ | ☒ | The exact sample size (n) for each experimental group/condition, given as a discrete number and unit of measurement |
| ☐ | ☒ | A statement on whether measurements were taken from distinct samples or whether the same sample was measured repeatedly |
| ☐ | ☒ | The statistical test(s) used AND whether they are one- or two-sided *Only common tests should be described solely by name; describe more complex techniques in the Methods section.* |
| ☒ | ☐ | A description of all covariates tested |
| ☐ | ☒ | A description of any assumptions or corrections, such as tests of normality and adjustment for multiple comparisons |
| ☐ | ☒ | A full description of the statistical parameters including central tendency (e.g. means) or other basic estimates (e.g. regression coefficient) AND variation (e.g. standard deviation) or associated estimates of uncertainty (e.g. confidence intervals) |
| ☐ | ☒ | For null hypothesis testing, the test statistic (e.g. F, t, r) with confidence intervals, effect sizes, degrees of freedom and P value noted *Give P values as exact values whenever suitable.* |
| ☒ | ☐ | For Bayesian analysis, information on the choice of priors and Markov chain Monte Carlo settings |
| ☒ | ☐ | For hierarchical and complex designs, identification of the appropriate level for tests and full reporting of outcomes |
| ☐ | ☒ | Estimates of effect sizes (e.g. Cohen's d, Pearson's r), indicating how they were calculated |

*Our web collection on statistics for biologists contains articles on many of the points above.*

## Software and code

Policy information about availability of computer code

| Data collection | Short-read sequencing of DNA isolated by CRISPR-CATCH<br>DNA libraries were sequenced on the Illumina Miseq, the Illumina Nextseq 550 or the Illumina NovaSeq 6000 platform.<br><br>Nanopore Sequencing and 5mC methylation calling<br>DNA isolated by CRISPR-CATCH was directly used without amplification for nanopore sequencing. Sequencing libraries were prepared using the Rapid Sequencing Kit (Oxford Nanopore Technologies, SQK-RAD004) according to the manufacturer's instructions. Sequencing was performed on a MinION (Oxford Nanopore Technologies).<br><br>Optical Mapping<br>The fluorescently labeled DNA molecules were loaded onto the Saphyr Chip G2.3 (Bionano Genomics, #20366, 30142) and were imaged sequentially across nanochannels on a Saphyr instrument (Bionano Genomics, #90023).<br><br>De novo assembly of SNU16 was performed with Bionano's de novo assembly pipeline (Bionano Solve v3.6, #90023) using standard haplotype aware arguments |
|---|---|
| Data analysis | Whole-Genome Sequencing<br>Reads were trimmed of adapter content with Trimmomatic24 (version 0.39), aligned to the hg19 genome using BWA MEM25 (0.7.17-r1188), and PCR duplicates removed using Picard's MarkDuplicates (version 2.25.3).<br><br>Analysis of TCGA ecDNA amplicon sizes<br>To obtain ecDNA intervals for TCGA tumors, we ran AmpliconClassifier (version 0.4.6; https://github.com/jluebeck/AmpliconClassifier) on AmpliconArchitect outputs published previously using WGS data2. ecDNA amplicon sizes were estimated by summing ecDNA amplicon interval sizes for each tumor. |

Metaphase DNA FISH image analysis

Colocalization analysis for two-color metaphase FISH data for ecDNAs in SNU16 cells described in Extended Data Figure 9f was performed using Fiji (version 2.1.0/1.53c)41. Images were split into the two FISH colors + DAPI channels, and signal threshold set manually to remove background fluorescence. Overlapping FISH signals were segmented using watershed segmentation. Colocalization was quantified using the ImageJ-Colocalization Threshold program and individual and colocalized FISH signals were counted using particle analysis.

Genetic variant analyses

SVs from short-read sequencing were identified with DELLY44 (version 0.8.7; using Boost version 1.74.0 and HTSlib version 1.12) using the delly call command. BCF files were converted to VCF using bcftools view in Samtools45. VAFs were calculated using both imprecise and precise variants. Read alignment was visualized using Gviz in R.

SNVs were identified using GATK (version 4.2.0.0)46 from short-read sequencing data as follows. First, base quality score recalibration was performed on bam files (generated as described above) using gatk BaseRecalibrator followed by gatk ApplyBQSR. Covariates were analyzed using gatk AnalyzeCovariates. SNVs were called using gatk Mutect2 from the recalibrated bam files, and SNVs were filtered using gatk FilterMutectCalls. Finally, vcf files were converted to table format using gatk VariantsToTable with the following parameters: "-F CHROM -F POS -F REF -F ALT -F QUAL -F TYPE -GF AD -GF GQ -GF PL -GF GT". Mutation variant allele frequencies (VAFs) were calculated by dividing alternate allele occurrences by the sum of reference and alternate allele occurrences. SNVs which had coverage depth of 5 or less or were not detected in WGS were filtered out. Read alignment was visualized using Gviz in R. To classify ecDNA-specific SNVs in GBM39 cells, we identified all SNVs with VAFs higher than 0.03 in ecDNAs isolated by CRISPR-CATCH using guide A, B, or A+B (given chromosome contamination levels of 0.01-0.02; Extended Data Figure 2d) and with VAFs in WGS lower than 0.997 (non-homozygous variants). Chromosome-specific SNVs were defined as non-ecDNA SNVs with VAFs in WGS lower than 0.1. Homozygous SNVs were defined as non-ecDNA-specific and non-chromosome-specific SNVs with VAFs in WGS above 0.99.

Nanopore Sequencing and 5mC methylation calling

DNA isolated by CRISPR-CATCH was directly used without amplification for nanopore sequencing. Sequencing libraries were prepared using the Rapid Sequencing Kit (Oxford Nanopore Technologies, SQK-RAD004) according to the manufacturer's instructions. Sequencing was performed on a MinION (Oxford Nanopore Technologies).

Bases were called from fast5 files using guppy (Oxford Nanopore Technologies, version 5.0.16) within Megalodon (version 2.3.3) and DNA methylation status was determined using Rerio basecalling models with the configuration file "res_dna_r941_min_modbases-all-context_v001.cfg" and the following parameters: "--outputs basecalls mod_basecalls mappings mod_mappings mods per_read_mods --mod-motif Z CG 0 --write-mods-text --mod-output-formats bedmethyl wiggle --mod-map-emulate-bisulfite --mod-map-base-conv C T --mod-map-base-conv Z C". Methylation calls on single molecules were visualized using Integrative Genome Viewer (IGV, version 2.11.1) in bisulfite mode.

ATAC-seq

Adapter-trimmed reads were aligned to the hg19 genome using Bowtie2 (2.1.0). Aligned reads were filtered for quality using samtools (version 1.9)31, duplicate fragments were removed using Picard's MarkDuplicates (version 2.25.3), and peaks were called using MACS2 (version 2.2.7.1)32 with a q-value cut-off of 0.01 and with a no-shift model. Peaks from replicates were merged, read counts were obtained using bedtools (version 2.30.0)33 and normalized using DESeq2 (using the "counts" function in DESeq2 with normalized = TRUE; version 1.26.0)34.

MNase-seq

Reads were trimmed of adapter content with Trimmomatic24 (version 0.39), aligned to the hg19 genome using BWA MEM25 (0.7.17-r1188), and PCR duplicates removed using Picard's MarkDuplicates (version 2.25.3). Coverage of nucleosome midpoints was obtained using bamCoverage from deepTools (version 3.5.1) with the following parameters: "--MNase --binSize 1".

Amplicon reconstruction from CRISPR-CATCH sequencing

Using short-read sequencing data from CRISPR-CATCH with double size selection as described above, we implemented new strategies and modified existing methods6 to resolve ecDNA structures. Broadly, the methods involved seven steps. The last six steps are available in a CRISPR-CATCH reconstruction pipeline, available at https://github.com/siavashre/CRISPRCATCH.

1. To identify the regions of interest, we ran PrepareAA (https://github.com/jluebeck/PrepareAA) (version 0.931.4) and AmpliconArchitect (version 1.2_r2, available from https://github.com/jluebeck/AmpliconArchitect) on two public bulk SNU16 WGS datasets (SRX546661250; SRR530826, Genome Research Foundation) and found comparable graphs in both. We used PrepareAA with BWA-MEM37 (version 0.7.12-r1039) to align reads to hg19 and CNVKit51 (version 0.9.7) to generate seed regions having copy number (CN) > 5. These regions were provided to AA, which constructed a CN-aware breakpoint graph. The genome regions AA included in the graph were converted to bed format and used as the seed regions in the analysis of each PFGE band, so that the regions studied were always consistent between bands.

2. Using WGS reads generated from CRISPR-CATCH-isolated DNA, for each band we next aligned to the hg19 reference genome using PrepareAA which included BWA MEM and a PCR-duplicate removal step (using samtools45 version 1.3.1), and we also made estimates of insert size distribution using Picard (version 2.25.6) for quality control purposes.

3. The aligned PFGE data and seed regions identified from bulk sequencing were provided to AmpliconArchitect (version 1.2_r2) to construct the CN-aware breakpoint graph, using non-default arguments –downsample -1 –pair_support 2 –no_cstats –insert_sdevs 8.5. The –insert_sdevs parameter allows for larger insert size variation without forming breakpoints from read pairs marked as discordant, as we found high insert size variance occurred frequently in DNA extracted from the gels. Following AA, we ran a script on the resulting CN-aware breakpoint graph to filter non-foldback graph edges joining regions smaller than 1 kb from the graph, representing potential unfiltered artifact edges arising from overdispersion in insert size variance, in order to reduce the complexity of the graph when performing pathfinding. Since the edges removed joined regions not more than 1 kb apart and did not lead to changes in the orientation of the genome, this step had a negligible effect on the resulting paths. This utility for filtering AA graphs is made available as part of PrepareAA (graph_cleaner.py).

4. Central to the method for ecDNA reconstruction is the assumption that a single ecDNA is being analyzed within the graph, and as a result the estimated genomic copy numbers should closely relate to the number of times a segment appears within the ecDNA. We termed the number of times a segment appeared within a single ecDNA as the "multiplicity" of a genomic segment. The path finding method first removes low CN elements from the graph representing the background genome and contamination from incomplete separation of ecDNAs (i.e., remove segments with CN below 20% of the maximum CN of all segments having length > 100 bp, or below 10% of the maximum, if the

maximum CN is >10000). In the remaining segments, we assumed that the majority of segments appeared once within an ecDNA. We assumed that ecDNAs for which the majority of segments are present more than once would reflect cases where two or more ecDNAs were present, instead of one. Thus, to compute the multiplicity of each graph segment, the method computes the 40th percentile of the remaining graph segment copy numbers and assigns that copy number, S1, to multiplicity = 1. For each segment, i, in the graph, we computed its multiplicity, M(i) as.

$$M(i) = \text{"round("} \ (CN(i))/S\_1 \ )$$

5. To find paths in the graph which represented candidate ecDNA structures, we used an exhaustive search constrained by the multiplicities of the segments and (if available) the estimated maximum molecule size suggested by the CRISPR-CATCH data. Candidate ecDNA structures are determined through a constrained depth-first search (DFS) approach, which attempts to identify paths in the graph, and performs the process starting at every segment in the graph assigned a non-zero multiplicity. During the search, the length of the path (in base pairs) must remain less than the maximum allowed length (L). For every segment i, appearing $n_i$ times in the path, $n_i \leq M(i)$. The DFS recursion terminates if either constraint is violated, and the current path is scored as $\sum_i n_i$ . The path is compared against the current best path (initiated as an empty path with score 0) and updated if it scores higher. Both the best-scoring cyclic paths as well as the best-scoring paths regardless of cyclic status are returned after removing all duplicate (identical) paths from the collection of best-scoring paths. This utility is also individually available from PrepareAA (plausible_paths.py).

6. We found a number of features of both the breakpoint graph and the reconstructions to be informative about the quality of the data in the band. We developed quality annotations reported along with each reconstruction to provide users with annotations about the confidence of the reconstruction. We note that CN-aware breakpoint graphs derived from NGS data may contain a number of error sources including missing edges between graph segments and incorrect estimation of copy numbers (leading sometimes to incorrect estimation of multiplicity). The method applies the following filters.

a) In the amplicon region analyzed by AA, the total amount of amplified material (non-zero multiplicity) should not significantly exceed the maximum estimated molecular size of the band (if provided). We used a cutoff such that amplicons with 1.4x the maximum estimated molecular size of the band were flagged for low quality (incomplete separation of ecDNA).

b) Changes in multiplicity must be accompanied by one or more breakpoint junctions, and thus for a breakpoint graph with |e| total edges, amplicons where
$$(|e|)/(\max(M(i))) \ < 1$$
were flagged for low quality (missing graph edges).

c) We defined a root mean square residual for the unexplained copy numbers of M(i). In a given path, for each segment i, having $n_i$ occurrences in the path, the root mean square residual was defined as
$$RMSR = \sqrt{1/N \ \sum_{(i=1)}^N (n_i - M(i))^2 \ }$$
where N is the number of segments having non-zero multiplicity in the graph. We set a default cutoff such that amplicons with RMSR > 0.9 were flagged as low quality (too many amplified graph segments having incompletely used multiplicity).

d) To assess how tightly segment copy numbers could be segregated by segment multiplicity, we computed the Davies-Bouldin index52 (DBI) on the clusters of copy numbers. Each cluster was comprised of all segment copy numbers assigned to a multiplicity (singleton clusters excluded), and the centroid of the cluster was the mean CN for the cluster. Amplicons where the DBI was > 0.3 were flagged as low quality due to noisy copy number estimation.

e) If a minimum molecular size for the band was given, we flagged reconstructions which fell below that 90% of that value as low quality as they reflected incomplete reconstructions.

f) If no segment in the reconstruction overlapped the CRISPR-Cas9 target site, we flagged it as being low quality as it was either an incomplete reconstruction, or the incorrect amplicon was detected.

7. Since the reconstructed paths are reported in the textual AA_cycles.txt format, the method also provides automated circular visualizations of the structures and the WGS coverage tracks which are generated by CycleViz (https://github.com/jluebeck/CycleViz) (version 0.1.0).

Validating candidate structures with optical mapping
To validate candidate ecDNA paths we used long-range optical mapping (OM) data. Previously, we developed a method, AmpliconReconstructor (AR)7, which uses OM data and AA's outputs as inputs.

ChIP-seq
Paired-end reads were aligned to the hg19 genome using Bowtie253 (version 2.3.4.1) with the --very-sensitive option following adapter trimming with Trimmomatic36 (version 0.39). Reads with MAPQ values less than 10 were filtered using samtools (version 1.9) and PCR duplicates removed using Picard's MarkDuplicates (version 2.20.3-SNAPSHOT). ChIP-seq signal was converted to bigwig format for visualization using deepTools bamCoverage54 (version 3.3.1) with the following parameters: --bs 5 --smoothLength 105 --normalizeUsing CPM --scaleFactor 10.

Code availability
Custom code to perform reconstructions of candidate ecDNA structures from CRISPR-CATCH data is available at https://github.com/siavashre/CRISPRCATCH.

For manuscripts utilizing custom algorithms or software that are central to the research but not yet described in published literature, software must be made available to editors and reviewers. We strongly encourage code deposition in a community repository (e.g. GitHub). See the Nature Portfolio guidelines for submitting code & software for further information.

## Data

Policy information about availability of data

All manuscripts must include a data availability statement. This statement should provide the following information, where applicable:
- Accession codes, unique identifiers, or web links for publicly available datasets
- A description of any restrictions on data availability
- For clinical datasets or third party data, please ensure that the statement adheres to our policy

Sequencing data generated in this study are deposited in SRA under BioProject accession PRJNA777710. WGS data from bulk GBM39 cells were obtained from the NCBI Sequence Read Archive, under BioProject accession PRJNA506071. WGS data from bulk SNU16 cells were previously generated (SRR530826, Genome Research Foundation). ATAC-seq and MNase-seq data for GBM39 were obtained from the NCBI Sequence Read Archive, under BioProject accession PRJNA506071. ChIP-seq data for SNU16 were previously published under GEO accession GSE15998628. Sequencing reads were mapped to the hg19 human reference genome. Source data are provided with this paper.

# Field-specific reporting

Please select the one below that is the best fit for your research. If you are not sure, read the appropriate sections before making your selection.

☒ Life sciences ☐ Behavioural & social sciences ☐ Ecological, evolutionary & environmental sciences

For a reference copy of the document with all sections, see nature.com/documents/nr-reporting-summary-flat.pdf

# Life sciences study design

All studies must disclose on these points even when the disclosure is negative.

| | |
|---|---|
| Sample size | No sample size calculation was performed. Sample sizes for DNA sequencing, optical mapping and metaphase DNA FISH are consistent with current standard sample sizes in the published literature. Sample size for the patient sample was based on the available biological material. |
| Data exclusions | No data were excluded from analysis. |
| Replication | Method was performed using three or more independent biological replicates (including independent CRISPR guides and experiments) to capture variability. All replication attempts were successful. |
| Randomization | Randomization is not relevant to this study. Cell culture samples were collected without prior selection or bias and were randomly assigned to treatment or control conditions without prior selection or bias. Appropriate experimental controls are shown in figure panels. |
| Blinding | Blinding is not relevant to this study. All data were collected using instruments without bias. Furthermore, raw data for all experimental conditions were uniformly processed using the same data processing and analysis pipeline for each experiment, ensuring that no human bias is introduced in the data analysis. |

# Reporting for specific materials, systems and methods

We require information from authors about some types of materials, experimental systems and methods used in many studies. Here, indicate whether each material, system or method listed is relevant to your study. If you are not sure if a list item applies to your research, read the appropriate section before selecting a response.

### Materials & experimental systems

| n/a | Involved in the study |
|---|---|
| ☒ | ☐ Antibodies |
| ☐ | ☒ Eukaryotic cell lines |
| ☒ | ☐ Palaeontology and archaeology |
| ☒ | ☐ Animals and other organisms |
| ☒ | ☐ Human research participants |
| ☒ | ☐ Clinical data |
| ☒ | ☐ Dual use research of concern |

### Methods

| n/a | Involved in the study |
|---|---|
| ☒ | ☐ ChIP-seq |
| ☒ | ☐ Flow cytometry |
| ☒ | ☐ MRI-based neuroimaging |

## Eukaryotic cell lines

Policy information about cell lines

| | |
|---|---|
| Cell line source(s) | GBM39 neurospheres were derived from patient tissue as previously described (Wu et al. Nature. 2019). SNU16 cells were obtained from ATCC (CRL-5974). |

| Authentication | SNU16 cells were obtained from ATCC and therefore were not authenticated. GBM39 neurospheres were derived from patient tissue as previously described (Wu et al. Nature. 2019) and were authenticated using metaphase DNA FISH with probes hybridizing to EGFR as well as chromosome 7 centromeric probe to confirm ecDNA amplification status, same as in Wu et al. Nature. 2019. |
| --- | --- |
| Mycoplasma contamination | Cells were tested negative for mycoplasma. |
| Commonly misidentified lines (See ICLAC register) | None of the cell lines used are registered by ICLAC as commonly misidentified. |

