## [Peer review file. · Nature Genetics]

Peer Review Information

Manuscript Title: Targeted profiling of human extrachromosomal DNA by CRISPR-CATCH

Corresponding author name(s): Dr Howard Chang

Reviewer Comments & Decisions:

Decision Letter, initial version:

3rd Dec 2021

Dear Howard,

Your Technical Report entitled "Targeted profiling of human extrachromosomal DNA by CRISPR-CATCH" has now been seen by 3 referees, whose comments are attached. In light of their advice we have decided that we cannot offer to publish your manuscript in Nature Genetics.

Reviewer #1 says that the technique is clever and elegant but has some limitations (e.g. it depends on prior knowledge about tumor-specific ecDNA) and its utility has not been fully demonstrated (i.e. will it work on patient samples?). Also, the biological findings have limited novelty.

Reviewer #2 notes that the method itself is not new (this would need to be made clearer), only the application to ecDNA. Like Reviewer #1, they'd like to see it being used to profile patient samples (frozen and primary).

Reviewer #3 thinks that the technology is interesting but has limitations: prior knowledge is required (same point as Reviewer #1's) and it's only been used here to study a homogeneous cell population (cell line). Also, the new biological findings are limited.

We feel that these reservations are sufficiently important as to preclude publication of this study in Nature Genetics.

Although we regret that we cannot offer to publish your paper in Nature Genetics given these reviews, I have discussed your manuscript and the reviewers' comments with our colleagues at Nature Communications. They would send an appropriately revised manuscript out for further review if you transfer it to Nature Communications. Should you wish to have your revised paper considered by Nature Communications, please use the link to the Springer Nature manuscript transfer service in the footnote once the revision is ready, and include a point-by-point response to the reviewers' concerns.

Please note that Nature Communications are satisfied that the findings represent a sufficient advance for publication and will not require you to perform new experiments in patient tissue samples as

indicated by the referees; they would only request you to carefully discuss this as a limitation of your study, although they would certainly welcome these data if it were within your possibilities to produce them. However, they will require all technical concerns raised by the referees to be addressed in full before sending the revised paper back to the original referees.

Your handling editor at Nature Communications would be Dr. Ilse Valtierra (ilseariadna.valtierragutierrez@nature.com). If there is anything you would like to discuss before transferring the paper and its reviews, please don't hesitate to contact her by e-mail.

Please note that Nature Communications is a fully open access journal. For information about article processing charges, open access funding, and advice and support from Springer Nature, please consult the Nature Communications Open Access page (www.nature.com/ncomms/open_access/index.html).

I am sorry that we cannot be more positive on this occasion but hope that you will find our referees' comments helpful.

With all best wishes,

Tiago

Tiago Faial, PhD
Senior Editor
Nature Genetics
<https://orcid.org/0000-0003-0864-1200>

Referee expertise:

Referee #1: cancer functional genomics and ecDNA

Referee #2: genetics and epigenetics of brain tumors

Referee #3: CRISPR technology

Reviewers' Comments:

Reviewer #1:

Remarks to the Author:

This manuscript presents a clever strategy for isolating and sequencing ecDNA, which has been

2notoriously challenging. The CRISPR-CATCH approach they present is elegant in its simplicity. Overall, the approach is a technical advance and it will have utility, but it is not without limitations. The main limitation (which annoyingly is not acknowledged) is that the approach requires prior knowledge of the ecDNA status and sequence. Furthermore, while method is robust for in vitro cultured cells, there is no evidence that it will work in vivo on patient tumor specimens, which is really the major barrier in the ecDNA field. Another concern is that, although the method is clever and novel, the findings upon application of the method to the various cancer cell lines are mostly confirmatory of previous studies. Other than the EGFR promoter-hypomethylation finding, very little new knowledge is offered in terms of ecDNA structure, diversity, or function. An example of this is in Figure 2. Prior studies have reported that EGFRvIII variant in the GBM39 cell line is almost exclusively on ecDNA (Turner et al., Nat 2017; Nathanson et al., Science 2014.) The result is presented as a novel finding, but it's confirmatory.

Reviewer #2:

Remarks to the Author:

This is a very well written manuscript reporting on a method to physically separate amplified DNA/Double Minute/ecDNA and the same locus on the chromosome using PFGE. The method is CRISPR-CATCH, developed by others (references 14 and 15), but not previously appreciated for its applicability to ecDNA in human tumor cell lines. Here the application is to characterize ecDNA structure, size, and epigenetic status relative to the chromosomal locus.

1. This is a novel application of a previously established method. In the abstract it is presented as a new method, and this must be revised. On the other hand, the application of CRISPR CATCH to study amplified genes in cancer cell lines is new and potentially quite useful to better understand amplified genes in many cancer cell lines.
2. It is not clear if this method would work with tissue rather than cell lines. Application to frozen or fresh tumor tissue would add significantly to its utility and discovery potential.
3. It is not clear how many chromosome copies are present in the cells being studied. This could seriously confound interpretations of the SNP/mutations. For example, GBM usually has 3 copies of chromosome 7 as a foundational genomic event. Cell lines derived from GBM may retain this status or increase or decrease the number of chromosome 7. One of these copies could in fact have the internal deletion VIII, as suggested by the identification of this allele in the chromosomal EGFR in Figure 2B. Without further testing, I do not think we can distinguish whether the VIII alleles in the chromosomal EGFR enrichment are actually chromosomal (counter to the conclusion) or are carry over from the ecDNA (supporting the conclusion). Same question for other amplicons studied here.
4. The authors could make a stronger case for why CRISPR-CATCH is substantially better than standard sequencing, such as WGS. The ratio of ecDNA vs chromosomal for the amplified segments may be 25X or more, and thus standard WGS sequencing provides sequences that are primarily from the ecDNA. WGS requires much less DNA (10 X less?) and gives more precise start and ends of the amplified region.

35. This is not the first empirical proof of amplicon size. Amplicons sizes have been reported by others using PFGE, including for EGFR, they have been estimated by direct visualization and even more precisely by segmentation analysis of WGS. Amplicon sizes have also been estimated by other methods including exome, RNA sequencing, and DNA methylation arrays at lower resolution.

6. Individual cancer cell lines often differ somewhat between labs and over time. They are cancer cells and continue to accumulate changes and variation. Are the authors concerned about using prior WGS from a study that not from the DNA used for CRISPR-CATCH experiments?

7. The term purified is inappropriate. The PFGE method yields ecDNA enrichment not purification.

8. The authors point out that their results with PFGE are discrepant with prior studies using Southern blotting but it is not clear why this would be the case.

9. In figure 2, how do the authors distinguish normal single nucleotide variants from mutations without the corresponding normal DNA?

10. The comments on evolutionary timeline are predicated on these variants being mutations rather than being germline in origin or being related to the 10+ years this cell line has been in culture. Presumably there were either two or three copies (early clonal chromosome 7 gain that is characteristic of GBM) of the EGFR locus prior to amplification as well. Along these lines, it should be possible for the authors to determine if the original template for the ecDNA is just one of the original (maternal/paternal) chromosomal segments, or both parental alleles were amplified.

11. The use of CRISPR CATCH to measure epigenomic features of ecDNA is very interesting, however it is not clear if there is much biological relevance to the differences observed. The uncertainty here is how to translate the y-axis values (relative methylation frequency (residual)" and "methylation relative frequency (mean residual)") into approximate absolute differences. Additionally, a normalized coverage graph depicting coverage at each CpG analyzed would be useful as well.

12. Are the nanopore reads sufficiently long to read out a chromosomal EGFR specific alteration (SNP?) and a ecDNA specific alteration (potentially VIII deletion or SNP?) and the DNA methylation status near the promoter? If so, a control experiment to address the admixtures in the enriched DNA would be analysis of the EGFR locus without the prior enrichments of ecDNA and chromosomal EGFR. Presumably reads could be assigned to ecDNA or chromosomal DNA based on these differences and the methylation difference should still be apparent.

13. Related to the two preceding points, the enrichment for ecDNA is 75% (25% is chromosomal), and the enrichment of chromosomal is also not 100% - how is that information taken into account when making the comparison of DNA methylation in ecDNA vs chromosomal DNA at the EGFR locus?

Reviewer #3:

4Remarks to the Author:

CRISPR-CATCH is a very innovative and interesting technology that separates and enriches megabase-sized ecDNA from chrDNA by performing a targeted cut on the DNA so that the size difference after cutting would allow the separation and enrichment of ecDNA from chrDNA.

What CRISPR-CATCH can do: separate and enrich megabase-sized ecDNA from chrDNA mixtures.

How well CRISPR-CATCH works: In this paper, the author demonstrated the technology by showing that: 1. CRISPR-CATCH could distinguish mutation and SNV on ecDNA comparing to chrDNA; 2. CRISPR-CATCH allows single-molecule DNA methylation profile on the purified ecDNA; 3. CRISPR-CATCH can reconstruct full length by performing de novo reconstruction of ecDNA.

Comments:

Overall, it is a well-written and organized paper; it is easy to understand. Data and methodology are also clear. I think this paper does a great job explaining the technology, validating the technology, but I think new biological findings are lacking in this paper. In addition, I also believe that this technology has a few drawbacks/disadvantages, that are open for discussion:

1. In the introduction, the authors mentioned that "Given the prevalence of ecDNA in cancer, there is an urgent need for better characterization of unique genetic and epigenetic, features of ecDNA in order to understand how it may differ from chromosomal DNA and obtain clues about how it is formed and maintained in tumors." I think the authors should emphasize with 1-2 sentences on why this particular technology is important, i.e. why do we need to sequence long ecDNA? What additional information does it give comparing to the shorter ecDNAs? I think these kinds of information will be helpful to highlight the importance of this technology.

2. The authors explained very well on describing the three current technologies used for studying ecDNA and the disadvantages of each technology. The authors mentioned that "This method (circle-seq) requires intact DNA circles and is therefore highly limited by ecDNA size, as megabase-sized DNA molecules are extremely fragile in solution and prone to breakage." Is there any data or reference supporting this? I would suggest that the authors provide some data on this, for example, perform exonuclease digestion, and then run a gel on the digested product without MDA amplification? Are all megabase-sized ecDNA gone? 100%? Or 90%? Or 50%? If it is not 100%, then why could not we just gel cut those megabase-sized ecDNA and run nanopore on them since nanopore sequencing could work with single molecules so you don't need to have that many molecules anyway.

3. This technology although allows separation of megabase-size ecDNA, but it is targeted, so we would need to know where it covers beforehand so that we could design sgRNA on that particular location. So we would need to have prior knowledge on what we are looking for, which is probably one disadvantage of this technology. Could you please comment on this?

4. When applied to human samples, as we all know, the population is heterogenous. For example, in any given targeted gene, there could be cells with 4 different profiles: 1. WT chrDNA + WT ecDNA; 2. Mut chrDNA + WT ecDNA; 3. WT chrDNA + Mut ecDNA; 4. Mut chrDNA + Mut ecDNA. Any human

5tissue would consist of a mixture of all 4 possibilities, so in the end even if we can get the ecDNA mutation information, we would not be able to tell if they are consistent or inconsistent with the cell chrDNA because the linkage between chrDNA and ecDNA is lost, which is probably another disadvantage of this technology. This technology has only been applied to human cell lines, which is a homogeneous population. Could you please comment on this?

5. Just a suggestion: The authors use the word “purifying”, “purification” a lot. However, the purity after purification is not very good (60% as shown in Figure 1e), and not reported directly in the paper (i.e., when I search for the keyword purity, it is not there). I would rather use the words “enriched”, “separation”, and emphasis on how many times it is enriched (like the authors mentioned: x30 enriched, etc.).

6. Since CRISPR-CATCH could capture megabase-sized DNA, why do you need to do de novo assembly? Why couldn't we just use nanopore sequencing, read the full length and it is done?

7. This method might suffer from multiplexing issue because each ecDNA might consist of DNAs from different chromosomes, so it is possible to lose information (similar to sgRNA A+B case in Figure f) if we design multiple sgRNAs targeting multiple loci. So, this technique should only be used using one sgRNA at a time. Is this correct?

Author Rebuttal to Initial comments

Targeted profiling of human extrachromosomal DNA by CRISPR-CATCH

Overview

We thank the reviewers for their careful reading of the manuscript and enthusiastic assessment, highlighting this work as “clever”, “elegant”, and “novel”. We also thank you for thoughtful critiques, which have significantly improved this work. The main additions in this revision are as follows:

1. **CRISPR-CATCH for patient tumor samples.** We have now optimized methods for applying CRISPR-CATCH for clinical tumor samples. We characterized the optimal tumor processing method for CRISPR-CATCH. We further introduce electro-depletion, a sequential electrophoretic strategy to remove fragmented DNA from patient tumor samples and enable CRISPR-CATCH. (revised Figure 1, Extended)

6Data Figure 3). We applied CRISPR-CATCH to an instructive case of metastatic melanoma, where the patient developed acquired resistance to BRAF and MEK inhibitors concordant with a new ecDNA. We identified and confirmed an 890 kb ecDNA encoding *NRAS*, a gene known to confer resistance to BRAF and MEK inhibitors (Nazarian et al., 2010) (**revised Fig. 1i**). An *NRAS* G12R missense mutation, which locks NRAS in the GTP-bound active conformation, was identified on ecDNAs with 100% allele frequency, suggesting strong selection for the mutated allele on ecDNAs (**Extended Data Figure 3h**). These data show that CRISPR-CATCH is fully feasible on patient tumor specimens, and validate an ecDNA mechanism for acquired resistance to MAP kinase pathway inhibitors in human patient for the first time.

- 2. Complete phasing of ecDNAs, chromosomal alleles, and subclonal mutations.** Using CRISPR-CATCH, we identified SNVs on ecDNA and the corresponding chromosomal DNA locus and determined their parental alleles of origin (**revised Figure 2, Extended Data Figure 4**). We found that the parental allele from which ecDNA originated no longer contains the amplified sequence and instead contains an excision scar. This direct linkage of an original DNA excision to tumor ecDNA formation and strongly supports an episome excision model of ecDNA genesis rather than sequence duplication. We further discover subclonal mutations that exist only in a subset of ecDNA molecules, supporting distinct mutagenic or DNA repair processes on ecDNAs.
- 3. Discovery of functionally specialized ecDNAs which amplify select enhancers or oncogene coding sequence.** We revealed an expansive landscape of heterogeneous and rearranged ecDNA structures. Using full-scale ecDNA amplicon reconstruction from CRISPR-CATCH in gastric cancer cells previously thought to have 2 types of ecDNAs, we discovered more than 10 species of ecDNAs (**revised Figure 4, Extended Data Figure 8,9**). This analysis provided a far more accurate picture of ecDNA structures compared to bulk whole-genome sequencing analysis. ecDNA oncogenes were previously thought to require co-amplification of its enhancers on the same molecule (Morton et al., 2019). We discovered ecDNAs containing rearrangements that led to (i) fused *MYC* and *FGFR2* loci originally from two distinct chromosomes; (ii) a truncated *FGFR2* ecDNA that contain only the protein coding sequence; (iii) ecDNAs that amplify select enhancer elements (**revised Figure 4**). These findings suggest that extrachromosomal amplification and rearrangement events are driven by enhancer proximity to oncogenes on an ecDNA molecule, as well as by intermolecular cooperation of abundant enhancer sequences

in a pool of ecDNA molecules. We further validated these findings by chromatin conformation capture, DNA fluorescence in-situ hybridization on metaphase spreads, and optical mapping data. These analyses demonstrate the utility of CRISPR-CATCH in identifying ecDNA oncogene amplicon structures in cancer cells and how mapping these structures can be used to understand the altered structural and regulatory landscape on ecDNAs.

Please find our detailed responses all reviewer comments in blue below and **highlighted within the main PDF file**. **Page and line numbers of added text are bolded and underlined in blue in the response.**

Referees' comments:

Reviewer #1:

Remarks to the Author:

This manuscript presents **a clever strategy for isolating and sequencing ecDNA, which has been notoriously challenging.** The CRISPR-CATCH approach they present is **elegant in its simplicity.** Overall, the approach is a technical advance and it will have utility, but it is not without limitations. The main limitation (which annoyingly is not acknowledged) is that the approach requires prior knowledge of the ecDNA status and sequence.

We thank the reviewer for the positive assessment of our work. We have acknowledged the limitation of requiring prior knowledge of amplified gene locus in the revised Discussion (**lines 406-409**). We note that whole-genome sequencing (WGS) or exome data are typically acquired for new cancer samples and already exist for many established cell lines and most human tumor types, such in TCGA and PCAWG. Thus the characteristic amplification landscape for most tumors are known. These bulk sequencing data can be analyzed by AmpliconArchitect (Deshpande et al., 2019) to identify amplified regions and predict potential ecDNA amplifications. Therefore, we find that it is typically not difficult to quickly narrow down potential ecDNA loci based on preliminary bulk DNA sequencing data. Metaphase DNA FISH is also routinely performed to validate ecDNA status on chromosomal spreads. However, as the reviewer pointed out, the major hurdle currently is in targeted isolation and profiling of ecDNA. CRISPR-CATCH fills this technical gap by enabling targeted analysis of ecDNA genetic and epigenetic composition with the resolution of short- and long-read sequencing while providing signal specificity to ecDNA. To demonstrate this technical need and how CRISPR-CATCH can meet it, we constructed a decision tree based on

8currently available technologies for ecDNA characterization with recommendations for how CRISPR-CATCH can be used (**Extended Data Figure 10**). Also of note, one of the greatest challenges created by ecDNA, is the intracellular heterogeneity of ecDNAs in terms of size and content. This is difficult to decipher from bulk WGS with amplicon reconstruction, or even from single cell sequencing. CRISPR-CATCH, because of its separation of ecDNA for subsequent analyses, opens a new window revealing the sequence diversity of ecDNAs, that cannot be resolved by direct sequencing approaches.

Furthermore, while method is robust for in vitro cultured cells, there is no evidence that it will work in vivo on patient tumor specimens, which is really the major barrier in the ecDNA field.

We thank the reviewer for this comment. We agree with the reviewer that isolation and targeted profiling of ecDNA/oncogene amplicons in patient tumors is another major barrier in understanding ecDNA structural and epigenetic characteristics. Following the reviewer's suggestion, we have now optimized methods for applying CRISPR-CATCH for clinical tumor samples from embedding tumors directly into agarose, chemical tumor dissociation, lysis, digestion and removal of fragmented tumor DNA to CRISPR-CATCH. One main difference between a tumor sample and a cancer cell line sample is the presence of fragmented DNA and general lower quality and amount of genomic DNA in a tumor sample due to tumor cell death and/or sample preparation and freezing. These DNA fragments can migrate in pulsed field gel electrophoresis and result in random sequencing background. To remove this background, we introduce electro-depletion, a sequential electrophoretic strategy to remove fragmented DNA from patient tumor samples and enable CRISPR-CATCH (**Figure 1h**). Briefly, processed tumor DNA entrapped in agarose plugs is loaded into a first agarose gel and a constant voltage is applied to deplete agarose plugs of DNA fragments. The electric field is then briefly reversed in direction to ensure that the intact DNA remains trapped in agarose plugs. These agarose plugs are then removed from the gel and subjected to CRISPR-CATCH as we described in the manuscript. This strategy effectively removes DNA fragments and traps intact genomic DNA as well as intact circular ecDNA, as evidenced by the presence of ecDNA bands from SNU16 cells after applying this method (**Extended Data Figure 3a,b**). For our clinical tumor sample, DNA bands were not visible after PFGE due to low amounts of DNA. Nonetheless, CRISPR-CATCH still successfully enriched for ecDNAs and confirmed the amplicon size shown by strong agreement between the molecular size on the gel and the length of the enriched amplified region in sequencing (**Fig. 1i, Extended Data Figure 3c-e**).

We applied optimized CRISPR-CATCH to an instructive case of metastatic melanoma. A patient with *BRAF* V600MUT melanoma was treated on BRAF and MEK inhibitors and developed metastatic lesion with acquired resistance coincident with the acquisition of ecDNA (**Extended Data Figure 3c**). CRISPR-CATCH and AmpliconArchitect confirmed the amplification of an 890-kb ecDNA encompassing *NRAS*, a gene known to confer resistance to BRAF inhibition (Nazarian et al., 2010) (**Fig. 1i, Extended Data Figure 3d**). Together, these data show that CRISPR-CATCH is fully feasible on human tumor specimens, including archival samples, and validate an ecDNA mechanism for acquired resistance to MAP kinase pathway inhibitors in authentic human cancer. We agree with the reviewer that the application of CRISPR-CATCH to tumor tissues significantly adds to its utility and thank the reviewer for this suggestion, which greatly improved this work.

The strength of CRISPR-CATCH is the ability to separate ecDNA-specific signal from the rest of the genome, which is particularly useful when the ecDNA landscape is complex and heterogeneous. For example, we show in this manuscript that the SNU16 stomach cancer cell line contains multiple ecDNA species with combinations of sequences originating from chromosomes 8, 10, and 11 (**Figure 4**). This heterogeneous mix of ecDNA species would be otherwise difficult to identify using bulk sequencing data. This is demonstrated by our mapping of heterogeneous ecDNA species using CRISPR-CATCH and accurate predictions of covalently connected amplicons (**Figure 4, Extended Data Figure 9**). In contrast, mapping of ecDNA structures from bulk WGS data are much less accurate when ecDNAs are heterogeneous (**Extended Data Figure 9**) as it is very difficult to determine the origin of sequencing reads or identify megabase-sized amplicon structures based on sequencing reads which are orders of magnitude shorter than the amplicons (only hundreds of bases from Illumina sequencing or several tens of kilobases from nanopore sequencing, as compared to ecDNA amplicons which can be over a megabase in size). As the reviewer pointed out in the beginning, isolating and sequencing ecDNA is very challenging in any context, even for cultured cancer cells. Therefore, we would like to point out that the ability to perform these targeted analyses on cell line models is itself a major technical hurdle overcome by CRISPR-CATCH. ecDNA profiling using CRISPR-CATCH can provide novel insights into ecDNA structure, diversity, origin, and epigenomic landscape as we showed in the revised manuscript and as detailed in the response to the following question. We have added these new analyses to the manuscript.

Another concern is that, although **the method is clever and novel**, the findings upon application of the method to the various cancer cell lines are mostly confirmatory of

10previous studies. Other than the EGFR promoter-hypomethylation finding, very little new knowledge is offered in terms of ecDNA structure, diversity, or function. An example of this is in Figure 2. Prior studies have reported that EGFRvIII variant in the GBM39 cell line is almost exclusively on ecDNA (Turner et al., Nat 2017; Nathanson et al., Science 2014.) The result is presented as a novel finding, but it's confirmatory.

Thank you for the opportunity to elaborate on the novel aspects of our study. As separation of megabase-sized ecDNAs was previously challenging, most analyses of ecDNA genetic and epigenetic composition heavily relied on computational inference from bulk DNA sequencing data e.g. mutational frequencies (Nathanson et al., 2014; Nikolaev et al., 2014; Turner et al., 2017; Wu et al., 2019), amplicon structure and size (Deshpande et al., 2019; Wu et al., 2019). Inference of ecDNA genetic variants from bulk DNA sequencing relies on allele frequencies. In essence, prior investigators assumed that if ecDNAs are present at high copy number (e.g. 100 copies per cell) and chromosomes are at 2 copies per cell, any mutation within the genomic interval that is present at high copy number must be on ecDNA. Any mutation that is present at low copy number must be on the chromosome. However, recent discovery of clustered, APOBEC-mediated mutagenesis of ecDNAs (termed *kyklonas*) challenge these assumptions and raise the importance of mapping ongoing low frequency ecDNA mutations (Bergstrom et al., 2022).

In contrast, we show in this manuscript that there are genetic variants which appear at low frequencies but can be located either on chromosomal DNA (minor allele) or ecDNA (major allele, low-frequency somatic mutations) as phased by CRISPR-CATCH (**Figure 2d**). The example provided by the reviewer about the EGFRvIII variant was computationally inferred to be on ecDNAs based on mutation allele frequencies but was not definitively proven to be exclusively on ecDNAs. Interpretations of ecDNA sequences in bulk sequencing data based solely on overall allele frequencies can be seriously confounded by heterogeneity in mutations and structures commonly observed in cancer cells. This fact is demonstrated in our manuscript in the analysis of both GBM39 and SNU16 cells (**revised Figure 2, Figure 4, Extended Data Figures 4,8,9**). In the case of GBM39 cells, there are close to 100 ecDNAs and 6 copies of chromosome 7 per cell on average (**Figure 1b, Extended Data Figure 2a**), resulting in a ratio of ~16X ecDNA to chromosomal DNA. Single-nucleotide variant (SNV) analysis using bulk WGS of GBM39 showed a bimodal distribution of variant allele frequencies (**Figure 2c**). With the assumption that most sequencing reads originated from ecDNA due to copy number amplification, one may conclude that the high-frequency SNVs are located on ecDNA (major allele), while the low-frequency SNVs are located on

chromosomal DNA (minor allele). However, we showed that this assumption is wrong. Using CRISPR-CATCH, we identified SNVs originating from either ecDNA or chromosomal DNA via molecular separation (**Figure 2c, Extended Data Figure 4**). We showed that there are a dozen of low-frequency somatic SNVs that cluster with the minor allele variants by total allele frequency in bulk WGS data but are in fact located on ecDNAs as identified by CRISPR-CATCH (**Figure 2c, Extended Data Figure 4c,d**). The discovery of subclonal ecDNA mutations supports the concept of ongoing mutagenesis or unique DNA repair mechanism on ecDNAs (Bergstrom et al., 2022).

Furthermore, CRISPR-CATCH allows separation of extrachromosomal signals from chromosomal signals, revealing important differences between the two. Phasing of genetic variants on ecDNA and chromosomal DNA allowed us to identify their parental alleles of origin and the excision scar from which ecDNA originated (**Figure 2c,d,e, Extended Data Figure 4a,b**). Direct comparison of the CpG methylation landscapes of ecDNA and chromosomal DNA further identified a state of hypomethylation at gene promoters on ecDNAs (**Figure 3**), a phenomenon that was previously obscured in bulk, aggregated sequencing data.

As highlighted in Overview point #3, CRISPR-CATCH enabled discovery of functionally specialized ecDNAs harboring pure enhancers or protein coding sequence. We revealed an expansive landscape of heterogeneous and rearranged ecDNA structures. Using full-scale ecDNA amplicon reconstruction from CRISPR-CATCH in gastric cancer cells previously thought to have 2 types of ecDNAs, we discovered more than 10 species of ecDNAs (**revised Figure 4, Extended Data Figure 8,9**). This analysis provided a far more accurate picture of ecDNA structures compared to bulk whole-genome sequencing analysis. ecDNA oncogenes were previously thought to require co-amplification of its enhancers on the same molecule (Morton et al., 2019). We discovered ecDNAs containing rearrangements that led to (i) fused *MYC* and *FGFR2* loci originally from two distinct chromosomes; (ii) a truncated *FGFR2* ecDNA that contain only the protein coding sequence; (iii) ecDNAs that amplify select enhancer elements (**revised Figure 4**). The existence of a truncated *FGFR2* ecDNA that lacks cognate enhancers known to be essential for *FGFR2* expression and the existence of enhancer-only amplicon (**revised Figure 4f-h**) challenges existing models of oncogene selection. Our findings suggest that extrachromosomal amplification and rearrangement events are driven by enhancer proximity to oncogenes on an ecDNA molecule, as well as by intermolecular cooperation of abundant enhancer sequences in a pool of ecDNA molecules. We further validated these findings by chromatin conformation capture, DNA fluorescence in-situ hybridization on metaphase spreads, and optical mapping data.

These analyses demonstrate the utility of CRISPR-CATCH in identifying ecDNA oncogene amplicon structures in cancer cells and how mapping these structures can be used to understand the altered structural and regulatory landscape on ecDNAs.

Per the reviewer's request, we have elaborated on the novel biological findings from the application of CRISPR-CATCH. These findings are summarized in Overview and listed here:

- 1. Complete phasing of ecDNAs, chromosomal alleles, and subclonal mutations.** Using ecDNA- and chromosome-specific SNVs identified by CRISPR-CATCH, we demonstrated that ecDNA arose from a single chromosomal parental allele and left behind an excision scar (**revised Figure 2, Extended Data Figure 4**). These data suggest that the allele that served as the original template for the ecDNA no longer contains the sequence harboring *EGFR* and provide strong evidence for the "episome model", a model of ecDNA formation in which a genomic locus is excised from chromosomal DNA as an episome and circularized to form an ecDNA (**Figure 2e**), rather than duplication of sequences due to replication, translocation or other errors (Bailey et al., 2020; Carroll et al., 1988; Storlazzi et al., 2006, 2010). We further discover subclonal mutations only on ecDNAs but not chromosomal *EGFR* alleles, supporting the *kyklonas* model of ongoing ecDNA mutagenesis (Bergstrom et al., 2022) and providing direct evidence for the concept that mutations continue to occur once ecDNAs have formed (PMID 25471132).
- 2. CRISPR-CATCH enabled epigenetic analysis led to the discovery of promoter hypomethylation on ecDNAs.** To our knowledge, this is the first ecDNA methylome analysis. This is also the first use of CRISPR-CATCH to profile the epigenetic landscape of isolated DNA molecules. Using this method, we discovered a pattern of promoter CpG hypomethylation on ecDNAs compared to chromosomal DNA originating from the same cell sample (**Figure 3**). This study may serve as the ground work for other types of targeted epigenetic analyses of ecDNAs, including the use of in-vitro DNA labeling strategies for assessment of chromatin accessibility and protein binding (Altemose et al., 2022; Shipony et al., 2020).
- 3. Discovery of ecDNA heterogeneity and functional specialization.** Heterogeneous ecDNA species are observed in many cancer samples (Deshpande et al., 2019), but the full scope and functional implications of the heterogeneity were unclear. CRISPR-CATCH enables molecular separation of multiple megabase-sized ecDNA species based on differences in size, providing fine detail of the composition

of DNA segments on each ecDNA species (**revised Figure 4, Extended Data Figures 8,9**). We also implemented a new method for reconstructing ecDNA sequence paths from CRISPR-CATCH sequencing data, called **Candidate AMplicon Path EnumeratoR (CAMPER; Methods)**, to perform full reconstruction of megabase-sized ecDNA circles containing multiple sequence segments. Furthermore, we discovered ecDNA species with oncogenes that are missing their enhancers and ecDNA species with select enhancer amplification, challenging the current dogma of gene expression (**Figure 4c,g,h, Extended Data Figure 9a**). These findings suggest that extrachromosomal amplification and rearrangement events may be driven by enhancer proximity to oncogenes on an ecDNA molecule as well as overall abundance of enhancer sequences in a pool of ecDNA molecules. As ecDNAs can interact with one another in trans within a hub (Hung et al., 2021), amplification of enhancer sequences in a pool of ecDNAs facilitates intermolecular enhancer-promoter interactions and further increase oncogene expression. These analyses demonstrate the utility of CRISPR-CATCH in identifying ecDNA oncogene amplicon structures in cancer cells and how mapping these structures can be used to understand the altered enhancer landscape on ecDNAs.

Reviewer #2:

Remarks to the Author:

This is a very well written manuscript reporting on a method to physically separate amplified DNA/Double Minute/ecDNA and the same locus on the chromosome using PFGE. The method is CRISPR-CATCH, developed by others (references 14 and 15), but not previously appreciated for its applicability to ecDNA in human tumor cell lines. Here the application is to characterize ecDNA structure, size, and epigenetic status relative to the chromosomal locus.

1. This is a novel application of a previously established method. In the abstract it is presented as a new method, and this must be revised. On the other hand, **the application of CRISPR CATCH to study amplified genes in cancer cell lines is new and potentially quite useful to better understand amplified genes in many cancer cell lines.**

We thank the reviewer for the positive assessment of our work. We have revised the abstract to clarify previous work on CRISPR-CATCH (already cited) and the novel application on ecDNAs in cancer cells in this technical report. We pioneered the following methodologic innovation as well: (1) “electrodepletion” to apply CRISPR-

14CATCH to human tumor DNA (**Figure 1**); (2) single molecule ecDNA methylome (**Figure 3**); (3) computational reconstruction of complex ecDNA amplicons, termed CAMPER (**Figure 4**). We agree that the novel application of CRISPR-CATCH to copy number amplifications caused by extrachromosomal DNA in the context of cancer will allow us to better understand and characterize these amplified molecules.

2. It is not clear if this method would work with tissue rather than cell lines. Application to frozen or fresh tumor tissue would add significantly to its utility and discovery potential.

We thank the reviewer for this comment. We agree with the reviewer that isolation and targeted profiling of ecDNA/oncogene amplicons in patient tumors is another major barrier in understanding ecDNA structural and epigenetic characteristics. Following the reviewer's suggestion, we have now optimized methods for applying CRISPR-CATCH for clinical tumor samples from embedding tumors directly into agarose, chemical tumor dissociation, lysis, digestion and removal of fragmented tumor DNA to CRISPR-CATCH. One main difference between a tumor sample and a cancer cell line sample is the presence of fragmented DNA and general lower quality and amount of genomic DNA in a tumor sample due to tumor cell death and/or sample preparation and freezing. These DNA fragments can migrate in pulsed field gel electrophoresis and result in random sequencing background. To remove this background, we introduce electro-depletion, a sequential electrophoretic strategy to remove fragmented DNA from patient tumor samples and enable CRISPR-CATCH (**Figure 1h**). Briefly, processed tumor DNA entrapped in agarose plugs is loaded into a first agarose gel and a constant voltage is applied to deplete agarose plugs of DNA fragments. The electric field is then briefly reversed in direction to ensure that the intact DNA remains trapped in agarose plugs. These agarose plugs are then removed from the gel and subjected to CRISPR-CATCH as we described in the manuscript. This strategy effectively removes DNA fragments and traps intact genomic DNA as well as intact circular ecDNA, as evidenced by the presence of ecDNA bands from SNU16 cells after applying this method (**Extended Data Figure 3a,b**). For our clinical tumor sample, DNA bands were not visible after PFGE due to low amounts of DNA. Nonetheless, CRISPR-CATCH still successfully enriched for ecDNAs and confirmed the amplicon size shown by strong agreement between the molecular size on the gel and the length of the enriched amplified region in sequencing (**Fig. 1i, Extended Data Figure 3c-e**).

We applied optimized CRISPR-CATCH to an instructive case of metastatic melanoma. A patient with *BRAF* V600MUT melanoma was treated on BRAF and MEK inhibitors and developed metastatic lesion with acquired resistance coincident with the

15acquisition of ecDNA (**Extended Data Figure 3c**). CRISPR-CATCH and AmpliconArchitect confirmed the amplification of an 890-kb ecDNA encompassing *NRAS*, a gene known to confer resistance to BRAF inhibition (Nazarian et al., 2010) (**Fig. 1i, Extended Data Figure 3d**). Together, these data show that CRISPR-CATCH is fully feasible on human tumor specimens, and validate an ecDNA mechanism for acquired resistance to MAP kinase pathway inhibitors in authentic human cancer. We agree with the reviewer that the application of CRISPR-CATCH to tumor tissues significantly adds to its utility and thank the reviewer for this suggestion, which greatly improved this work. We would further like to point out that isolating and sequencing ecDNA is very challenging in any context, even for cultured cancer cells. Therefore, we note that the ability to perform these targeted analyses on cell line models is itself a major technical hurdle overcome by CRISPR-CATCH. ecDNA profiling using CRISPR-CATCH can provide novel insights into ecDNA structure, diversity, origin, and epigenomic landscape as we showed in the revised manuscript and as detailed in the Overview. We have added these new analyses to the manuscript.

3. It is not clear how many chromosome copies are present in the cells being studied. This could seriously confound interpretations of the SNP/mutations. For example, GBM usually has 3 copies of chromosome 7 as a foundational genomic event. Cell lines derived from GBM may retain this status or increase or decrease the number of chromosome 7. One of these copies could in fact have the internal deletion VIII, as suggested by the identification of this allele in the chromosomal EGFR in Figure 2B. Without further testing, I do not think we can distinguish whether the VIII alleles in the chromosomal EGFR enrichment are actually chromosomal (counter to the conclusion) or are carry over from the ecDNA (supporting the conclusion). Same question for other amplicons studied here.

We thank the reviewer for raising this point. To quantify chromosome copies, we performed DNA fluorescence in-situ hybridization (FISH) for *EGFR* and chromosome 7 (centromeric probe) on metaphase chromosome spreads and found that the vast majority of GBM39 cells have six copies of chromosome 7, the chromosomal origin of the *EGFR* locus. (**Figure 1b, Extended Data Figure 2a**). Second, we calculated the fraction of molecules in the chromosomal EGFR enrichment corresponding to carryover ecDNA using the ratio of sequencing coverage in the ecDNA-amplified region to coverage in the chromosomal overhangs (enriched chromosomal regions located outside of the ecDNA-amplified region resulting from CRISPR-CATCH guide cleavage; **Extended Data Figure 2c**). This analysis showed 70.6% of the signal originating from chromosomal DNA and 29.4% originating from ecDNA in chromosomal EGFR isolated

using guides E+F (**Extended Data Figure 2d**), which was used in the *EGFRvIII* analysis shown in Figure 2b. This level of chromosomal *EGFR* enrichment corresponds almost exactly to the fraction of wild-type *EGFR* (without vIII deletion) observed (75.4% wild-type *EGFR* and 24.6% *EGFRvIII*, **Figure 2b**). Given that virtually all ecDNAs carry the *EGFRvIII* mutation (99.6% *EGFRvIII* allele frequency in isolated ecDNAs with 99.8% enrichment using guide A; **Figure 2b, Extended Data Figure 2d**), the presence of ecDNA in 29.4% of isolated *EGFR* copies in the enriched chromosomal fraction can fully explain and is the simplest explanation for the observed frequency of *EGFRvIII* (**Figure 2b**). This strong concordance between chromosomal enrichment and fraction of wild-type *EGFR* further supports our conclusion that chromosomal DNA predominantly carries wild-type *EGFR* while ecDNAs predominantly carry the *EGFRvIII* mutant.

Furthermore, we followed the reviewer's excellent suggestion below about using the phased SNVs to identify paternal/maternal alleles on chromosomal DNA and ecDNA (in response to **Question #10**). The majority of SNVs we observed in the GBM39 neurospheres are present on either all alleles (likely homozygous germline SNVs), all ecDNA molecules, or all chromosomal DNA based on correspondence to the known level of DNA enrichment (**Figure 2c, Extended Data Figure 4a,b**). In fact, the vast majority of chromosome SNVs were undetectable on ecDNAs, and the vast majority of ecDNA-specific SNVs were undetectable beyond the level of residual ecDNA in the enriched chromosomal DNA fraction (~30%, **Extended Data Figure 2d,4a**). This observation suggests that these are haplotype-specific SNVs originating from different parental alleles. Thus, using these SNVs, we were able to infer the parental allele from which ecDNA originated (allele 1), a different allele than that whose *EGFR* locus is still present on chromosomal DNA (allele 2). This led us to further identify allele 1 on chromosomal DNA containing an excision scar from which ecDNA originated. These observations support the distinct origins of *EGFR* copies on ecDNA and chromosomal DNA (**Figure 2d**, detailed below in response to **Question #10**). These data suggest that ecDNA arose from a single chromosomal allele and further supports our conclusion that chromosomal *EGFR* and extrachromosomal *EGFR* are from different alleles. Therefore, it is highly unlikely that a very tiny fraction of the chromosomal *EGFR* allele contains the same vIII deletion as the ecDNA allele as it would have had to arise independently. We have incorporated the additional analysis and have summarized the series of genomic events that preceded the formation and amplification of ecDNA as revealed by CRISPR-CATCH (**Figure 2e**, also discussed in response to **Question #10**).

4. The authors could make a stronger case for why CRISPR-CATCH is substantially better than standard sequencing, such as WGS. The ratio of ecDNA vs chromosomal

for the amplified segments may be 25X or more, and thus standard WGS sequencing provides sequences that are primarily from the ecDNA. WGS requires much less DNA (10 X less?) and gives more precise start and ends of the amplified region.

We thank the reviewer for the question and opportunity to clarify the technical advantages of CRISPR-CATCH. While WGS aims to achieve 40x genome coverage, we routinely obtain >200-1000x coverage of ecDNA sequence using CRISPR-CATCH (**Figures 1 and 3**). This greatly increased depth allowed single-molecule analysis of ecDNA methylome (**Figure 3**) and discovery of subclonal mutations on ecDNAs, detailed below.

First, even in cases in which ecDNA copy numbers are high, interpretations of ecDNA sequences in bulk sequencing data based solely on overall allele frequencies can be seriously confounded by heterogeneity in mutations and structures commonly observed in cancer cells. This fact is demonstrated in our manuscript in the analysis of both GBM39 and SNU16 cells. In the case of GBM39 cells, there are close to 100 ecDNAs and 6 copies of chromosome 7 per cell on average (**Figure 1b, Extended Data Figure 2a**), resulting in a ratio of ~16X ecDNA to chromosomal DNA. Single-nucleotide variant (SNV) analysis using bulk WGS of GBM39 showed a bimodal distribution of variant allele frequencies (**Figure 2c**). With the assumption that most sequencing reads originated from ecDNA due to copy number amplification, prior investigators assumed that high-frequency SNVs are located on ecDNA (major allele), and all low-frequency SNVs are located on chromosomal DNA (minor allele). However, we showed that this assumption is wrong. Using CRISPR-CATCH, we identified SNVs originating from either ecDNA or chromosomal DNA via molecular separation (**Figure 2c, Extended Data Figure 4**). We showed that there are a number of low-frequency somatic SNVs that cluster with the minor allele variants by total allele frequency in bulk WGS data but are in fact located on ecDNAs as identified by CRISPR-CATCH (**Figure 2c, Extended Data Figure 4c,d**). This erroneous assumption based on copy numbers alone is particularly consequential when observing somatic genetic changes in ecDNA sequences over time or in response to treatments, as well as when these SNVs are used to infer properties of ecDNA in other sequencing datasets such as RNA-seq, ATAC-seq, and ChIP-seq to assess allele-specific activities (Abramov et al., 2021; Wu et al., 2019). Moreover, the discovery of subclonal mutations on ecDNA supports the concept of ongoing mutagenesis or unique DNA repair mechanism on ecDNAs (Bergstrom et al., 2022).

Second, CRISPR-CATCH allows separation of extrachromosomal signals from chromosomal signals, revealing important differences between the two. Phasing of genetic variants on ecDNA and chromosomal DNA allowed us to identify their alleles of origin and the excision scar from which ecDNA originated (**Figure 2c-e**, **Extended Data Figure 4a,b**, detailed in response to **Question #10** below). Direct comparison of the CpG methylation landscapes of ecDNA and chromosomal DNA further identified a state of hypomethylation at gene promoters on ecDNAs, a phenomenon that was previously obscured in bulk, aggregated sequencing data (**Figure 3**).

Third, CRISPR-CATCH can resolve ecDNA heterogeneity that WGS cannot. In the case of SNU16 cells as well as a significant subset of cancer samples, heterogeneous ecDNA species can be observed in a cell population (**Figure 4c**). As these ecDNA species can each range from hundreds of kilobases to over a megabase in size and can contain multiple sequence segments connected by structural rearrangements, bulk WGS (short-read or nanopore sequencing) provides a collapsed and limited picture of the true diversity of ecDNA structures (**Figure 4c**, **Extended Data Figure 9c,d**). This is because bulk DNA sequencing represents the aggregation of sequencing reads originating from multiple ecDNA species as well as chromosomal DNA. In contrast, CRISPR-CATCH enables molecular separation of multiple megabase-sized ecDNA species based on differences in size, providing fine detail of the composition of DNA segments on each ecDNA species (**Figure 4c**). Fine mapping of these ecDNA species also enable identification of connected DNA segments, providing a much more accurate picture of ecDNA structures compared to bulk sequencing (**Extended Data Figure 9c,d**). We further validated these ecDNA structures using unnormalized background signals from chromatin conformation capture (H3K27 acetylation HiChIP; covalently connected DNA segments are more likely to show increased interaction as compared to unconnected segments; **Extended Data Figure 9c,d**), optical mapping (**Figure 4d-f**, **Extended Data Figure 8b**), and DNA fluorescence in-situ hybridization (FISH) on metaphase spreads (**Extended Data Figure 9e,f**). Much of this information is lost in bulk sequencing. Even when we attempted to predict connected DNA segments in WGS data using breakpoints identified from discordant reads and sequence multiplicities (using AmpliconArchitect), most connected DNA segments failed to be detected (**Extended Data Figure 9c,d**). As sequencing reads are much shorter than the typical ecDNA (hundreds of bases in illumina sequencing, tens of kilobases in nanopore sequencing), connections between DNA segments beyond the length of a sequencing read can only be computationally inferred in bulk WGS. In contrast, CRISPR-CATCH allows separation and mapping of full-length megabase-sized ecDNAs. Together, these data show that CRISPR-CATCH provides more specific

and detailed information about the genetic landscape of ecDNAs in cancer cells compared to bulk WGS.

Overall, CRISPR-CATCH provides more detailed and targeted information about ecDNA genetic and epigenetic compositions compared to WGS of bulk genomic DNA. We have revised the Discussion to highlight the above advantages (lines 375-394). To further clarify the technical needs met by CRISPR-CATCH, we constructed a decision tree based on currently available technologies for ecDNA characterization with recommendations for how CRISPR-CATCH can be used (**Extended Data Figure 10**).

5. This is not the first empirical proof of amplicon size. Amplicon sizes have been reported by others using PFGE, including for EGFR, they have been estimated by direct visualization and even more precisely by segmentation analysis of WGS. Amplicon sizes have also been estimated by other methods including exome, RNA sequencing, and DNA methylation arrays at lower resolution.

We thank the reviewer for the comment and would like to clarify our meaning of the term “empirical proof”. There is no doubt that amplicon sizes have been estimated by computational inference in the past by WGS, RNA sequencing, southern blotting from PFGE, and other methods as listed by the reviewer. However, we use the term “empirical” as opposed to “inferential” to specifically refer to the experimental validation of inferred amplicon sizes and structures. As we discuss in the manuscript (lines 90-101), the task of validating these inferred amplicon structures and sizes can be difficult particularly when the ecDNA structure is complicated and heterogeneous such as those shown in the SNU16 stomach cancer cell line (**Figure 4, Extended Data Figure 9**). For ecDNAs with more simple structures, we have more confidence in our prediction from sequencing analysis, such as that shown previously for *EGFR* ecDNAs in GBM39 (Luebeck et al., 2020; Wu et al., 2019). However, ambiguity increases with the structural complexity and heterogeneity of ecDNAs, as is the case of the SNU16 cancer cell line shown in this manuscript (**Figure 4, Extended Data Figure 9**). We demonstrated that prediction of amplicon structures by segmentation analysis of WGS is imprecise, missing important structures identified by CRISPR-CATCH and further validated by optical mapping, DNA fluorescence in-situ hybridization (FISH) and unnormalized background signals of chromatin conformation capture (**Figure 4d-f, Extended Data Figure 8b,9c-f**). Our point is that these computational predictions can now be empirically validated or modified based on actual observed amplicon sizes on a gel and targeted sequencing using CRISPR-CATCH. As we note in the revised manuscript (lines 169-170) and as the reviewer pointed out, PFGE was originally used to show the

20size of ecDNAs containing amplified genes by southern blot (e.g. the *DHFR* gene) (van der Blik et al., 1988). Southern blotting, however, provides binary information and can only tell us whether a probe binds or not. For example, bands corresponding to 500 kb, 1 Mb and 2 Mb on a southern blot targeting *DHFR* would tell us there are amplicons containing *DHFR* that are 500 kb, 1 Mb and 2 Mb in size, but it would not tell us which amplicon structure corresponds to each of those sizes other than that they all contain *DHFR*. It gets yet more complicated when multiple sequence segments are incorporated and connected in different permutations on the circles in addition to the amplified gene itself. Instead, CRISPR-CATCH tells us which amplicon (structure determination by sequencing) is what size (visualization on the gel), which is what we meant by “empirical proof”. However, the reviewer’s point about previous visualization of amplicon sizes is well taken. To use more precise language, we have modified the wording to “first empirical pairing of ecDNA amplicon size (by molecular separation) to structure with base-pair resolution (by sequencing)” (lines 170-172).

6. Individual cancer cell lines often differ somewhat between labs and over time. They are cancer cells and continue to accumulate changes and variation. Are the authors concerned about using prior WGS from a study that not from the DNA used for CRISPR-CATCH experiments?

We thank the reviewer for the comment and understand the concern of potential changes in cancer cell lines. In the case of GBM39, we used the same cell source from an early passage as that used in our *Nature* 2019 study (Wu et al., 2019). In the case of the SNU16 cell line, the reviewer is correct that we used previously published WGS datasets on SNU16 for identifying ecDNA regions of interest. However, we note that the two datasets provide comparable regions. Furthermore, rather than referencing WGS data, we performed directed genome assembly on the CRISPR-CATCH sequencing data generated in house to identify circular ecDNA structures. The WGS data were only used for selecting initial seed regions. The seed regions found in bulk sequencing were comparable between published datasets (In other words, we used WGS data to select the same copy-number-amplified genomic regions). To further increase our confidence in our reconstructed ecDNAs, we generated optical mapping data using our cell stock to provide orthogonal evidence of DNA contigs (**Figure 4d-f, Extended Data Figure 8b**). This validated our ecDNA reconstruction, providing high confidence to amplicon structures we obtained from CRISPR-CATCH and short-read sequencing. Finally, to validate that the seed regions in the *MYC* and *FGFR2* loci are correspondingly amplified and located on ecDNAs in our cell stock, we have now included DNA FISH data on metaphase spreads of our cells using probes targeting various regions in the *MYC* and

FGFR2 loci (**Extended Data Figure 9e,f**). Together, these analyses provide us with exceptional confidence about our use of the prior WGS datasets.

7. The term purified is inappropriate. The PFGE method yields ecDNA enrichment not purification.

We thank the reviewer for the suggestion on wording and agree that “purified” should be replaced. We have modified the text to use “enrichment”, “separation” or “isolation” to describe the method.

8. The authors point out that their results with PFGE are discrepant with prior studies using Southern blotting but it is not clear why this would be the case.

We thank the reviewer for pointing out unclear language. We meant to say that our data are in agreement with previous southern blot studies. Both our current study and previous southern blot studies showed that intact circular ecDNAs do not migrate freely in PFGE. We understand that our original wording was confusing and have changed this sentence to “ecDNAs were not detectable in the resolution window, indicating that intact circular ecDNAs do not migrate freely in PFGE. This finding is in agreement with previous southern blot studies” (**lines 132-134**).

9. In figure 2, how do the authors distinguish normal single nucleotide variants from mutations without the corresponding normal DNA?

We thank the reviewer for this question. Without paired sequencing of normal DNA from the same patient, it is indeed difficult to distinguish normal single-nucleotide variants (SNVs) from somatic mutations. In the SNV analysis in this manuscript, our main goal was to demonstrate that allele-specific SNVs as well as low-frequency SNVs can be identified on either ecDNAs or chromosomes. As such, we included all SNVs that did not match the reference genome sequence as our input. However, the majority of SNVs we observed in the GBM39 neurospheres are present on either all alleles (likely homozygous germline SNVs), all ecDNA molecules, or all chromosomal DNA based on correspondence to the known level of DNA enrichment (**Extended Data Figure 4a-c**). In fact, the vast majority of chromosome SNVs were undetectable on ecDNAs, and the vast majority of ecDNA-specific SNVs were undetectable beyond the level of residual ecDNA in the enriched chromosomal DNA fraction (~30%, **Extended Data Figure 2d,4c**). This observation suggests that these are haplotype-specific SNVs originating from different parental alleles. In addition to these high-frequency SNVs, we

also observed an accumulation of somatic mutations resulting in lower-frequency variants on either ecDNA or chromosomal DNA (**Figure 2c, Extended Data Figure 4c,d**).

10. The comments on evolutionary timeline are predicated on these variants being mutations rather than being germline in origin or being related to the 10+ years this cell line has been in culture. Presumably there were either two or three copies (early clonal chromosome 7 gain that is characteristic of GBM) of the EGFR locus prior to amplification as well. Along these lines, it should be possible for the authors to determine if the original template for the ecDNA is just one of the original (maternal/paternal) chromosomal segments, or both parental alleles were amplified.

Thank you for this insightful suggestion. As we mentioned in response to **Question #3** above, we quantified chromosome copies by performing DNA fluorescence in-situ hybridization (FISH) for EGFR and chromosome 7 (centromeric probe) on metaphase chromosome spreads and found that the vast majority of GBM39 cells have six copies of chromosome 7, the chromosomal origin of the EGFR locus. (**Figure 1b, Extended Data Figure 2a**).

Second, we followed the reviewer's suggestion about using the phased SNVs in CRISPR-CATCH to identify paternal/maternal alleles on chromosomal DNA and ecDNA. We observed strong divergence of SNVs on ecDNAs compared to the chromosomal locus (**Extended Data Figure 4a-c**). Using these SNVs, we were able to infer the parental allele from which ecDNA originated (allele 1), a different allele than that whose *EGFR* locus is still present on chromosomal DNA (allele 2; **revised Figure 2, Extended Data Figure 4**). These data suggest that ecDNA arose from a single chromosomal allele. This observation also led to the discovery of an excision scar on the chromosomal allele from which ecDNA originated, one of six copies of chromosome 7 still remaining (**Figure 2d**). Quantification of variant allele frequencies in the chromosomal arm upstream and downstream of the EGFR amplified region, as well as quantification of the frequency of the excision scar, demonstrated that there is one copy of allele 1 (origin of ecDNA) and five copies of allele 2 (remaining chromosomal DNA carrying EGFR) (**Figure 2d**). Together, this analysis shows the sequence of genomic events that preceded the formation of ecDNA (**Figure 2e**). From the two original parental alleles, there was a DNA rearrangement event on allele 1 that led to the excision and circularization of the EGFR ecDNA. The gain of the EGFRvIII mutation and ecDNA amplification led to the major ecDNA allele we observed. In addition, there was a gain of 4 additional copies of allele 2 of chromosome 7. These data suggest that the

allele that served as the original template for the ecDNA no longer contains the sequence harboring *EGFR* and provide strong evidence for the “episome model”, a model of ecDNA formation in which genomic loci are excised from chromosomal DNA as an episome and circularized to form an ecDNA (**Figure 2e**), rather than duplication of sequences due to replication, translocation or other errors (Bailey et al., 2020; Carroll et al., 1988; Storlazzi et al., 2006, 2010). Using this approach on other cancer cell models may identify general features of ecDNA and chromosomal alleles, including the chromosomal allele from which ecDNA originated, and provide clues about mechanisms of ecDNA formation (e.g. episome formation, chromothripsis) as well as potential precursors of ecDNA. This analysis also demonstrates how CRISPR-CATCH profiling of ecDNAs enables discoveries of ecDNA origin and evolution and may be applied to other cancer models in the future to study the genetic composition of ecDNAs. We have incorporated the additional analysis in the revised manuscript.

11. The use of CRISPR CATCH to measure epigenomic features of ecDNA is very interesting, however it is not clear if there is much biological relevance to the differences observed. The uncertainty here is how to translate the y-axis values (relative methylation frequency (residual)” and “methylation relative frequency (mean residual)”) into approximate absolute differences. Additionally, a normalized coverage graph depicting coverage at each CpG analyzed would be useful as well.

We thank the reviewer for this comment and agree that absolute differences in methylation frequencies are important to include. For the gene promoters shown, we found them to be 10-30 percentage points lower in methylation levels on ecDNA compared to chromosomal DNA (**Extended Data Figure 5c**). Furthermore, this difference becomes yet larger when considering the difference in copy numbers between chromosomal DNA and ecDNA. For example, a 10-percent decrease in DNA methylation on ecDNAs translates to an average of 10 more unmethylated gene copies per cell as each cell contains ~100 ecDNAs on average (**Extended Data Figure 2a**) (Lange et al., 2021; Turner et al., 2017; Wu et al., 2019). We have revised the manuscript to include this result as well as normalized coverage graphs as the reviewer suggested (**Extended Data Figure 5c**).

12. Are the nanopore reads sufficiently long to read out a chromosomal *EGFR* specific alteration (SNP?) and a ecDNA specific alteration (potentially VIII deletion or SNP?) and the DNA methylation status near the promoter? If so, a control experiment to address the admixtures in the enriched DNA would be analysis of the *EGFR* locus without the prior enrichments of ecDNA and chromosomal *EGFR*. Presumably reads could be

24assigned to ecDNA or chromosomal DNA based on these differences and the methylation difference should still be apparent.

We thank the reviewer for the experimental idea. In brief, we tested brute-force long-read sequencing without CRISPR-CATCH and found that we could not properly phase the SNVs and structural variants. Without CRISPR-CATCH enrichment, obtaining sufficient coverage of ecDNA and the corresponding chromosomal DNA region using nanopore sequencing is expensive and difficult to obtain from a MinION sequencer. In the case of GBM39 cells, 30 times more sequencing reads by WGS are required to achieve the same coverage obtained by CRISPR-CATCH-isolated DNA in a MinION run. Therefore, to address the reviewer's question, we isolated high-molecular-weight genomic DNA from GBM39 cells without any enrichment and performed deep nanopore sequencing on a PromethION instrument in order to obtain sufficient coverage (27X coverage of the entire genome and 979X coverage of the *EGFR* amplicon region, close to the 1194X coverage we obtained previously after ecDNA enrichment by CRISPR-CATCH in a MinION run; **Figure R1a, Figure 3f**). As the reviewer suggested, we used chromosomal SNVs and ecDNA SNVs we identified using CRISPR-CATCH (**Figure 2c, Extended Data Figure 4a-c**) to phase nanopore sequencing reads from the unenriched genomic DNA (**Figure R1a**). We used the manufacturer's currently recommended tools for SNV identification and phasing in nanopore reads (Guppy, Longshot). Unfortunately, using these existing standard tools for SNV calling and read phasing for nanopore sequencing data, we were not able to confidently call any of the SNVs identified in the chromosomal *EGFR* locus (**Figure R1b**). While we obtained deep sequencing coverage, there are many more copies of ecDNA than chromosomal DNA containing the corresponding *EGFR* locus in GBM39 cells (~100 copies of ecDNA and 5-6 copies of chromosomal DNA; **Figure 2e, Extended Data Figure 2a**). As a result, chromosomal variant allele frequencies are much lower than those for ecDNA. Furthermore, the base calling error rate of nanopore sequencing is still much higher than Illumina sequencing due to the error-prone conversion of raw electrical signals to DNA base identities. Therefore, the skew in the distribution of variant allele frequencies and high basecalling error rate make it challenging to confidently assign haplotypes in nanopore data when it comes to SNVs with lower allele frequencies. Therefore, the experiment proposed by the reviewer showed that CRISPR-CATCH is essential to accomplish the desired goals.

Figure R1. Phasing Nanopore sequencing reads on unenriched genomic DNA from GBM39 cells using SNVs identified by CRISPR-CATCH. (a) Experimental workflow for phasing ecDNA- and chromosome-derived nanopore sequencing reads from unenriched bulk genomic DNA, based on SNVs identified by CRISPR-CATCH. **(b)** A Venn diagram of SNVs identified as chromosomal, ecDNA, or homozygous mutations using CRISPR-CATCH corresponding to data shown in Figure 2, as well as SNVs phased using nanopore sequencing on unenriched genomic DNA. Zero chromosomal mutations were phased in the nanopore sequencing data.

13. Related to the two preceding points, the enrichment for ecDNA is 75% (25% is chromosomal), and the enrichment of chromosomal is also not 100% - how is that information taken into account when making the comparison of DNA methylation in ecDNA vs chromosomal DNA at the EGFR locus?

We thank the reviewer this comment. The reviewer is correct that the separation of ecDNA and chromosomal DNA was not 100%, and this is taken into account in our interpretation of genetic variants by background subtraction. For ecDNA methylation, this suggests that the difference in methylation fractions is likely an underestimation of the actual difference between ecDNA and chromosomal DNA (the more cross-contamination between two groups of molecules, the smaller the apparent difference). Nevertheless, the fact that we were able to detect hypomethylation specifically at gene promoters on ecDNAs suggests that ecDNAs may have an altered epigenetic landscape that enable high levels of oncogene activity (Hung et al., 2021; Wu et al., 2019).

Reviewer #3:

26

Remarks to the Author:

CRISPR-CATCH is **a very innovative and interesting technology** that separates and enriches mega-sized ecDNA from chrDNA by performing a targeted cut on the DNA so that the size difference after cutting would allow the separation and enrichment of ecDNA from chrDNA.

What CRISPR-CATCH can do: separate and enrich megabase-sized ecDNA from chrDNA mixtures.

How well CRISPR-CATCH works: In this paper, the author demonstrated the technology by showing that: 1. CRISPR-CATCH could distinguish mutation and SNV on ecDNA comparing to chrDNA; 2. CRISPR-CATCH allows single-molecule DNA methylation profile on the purified ecDNA; 3. CRISPR-CATCH can reconstruct full length by performing de novo reconstruction of ecDNA.

Overall, it is a well-written and organized paper; it is easy to understand. Data and methodology are also clear. I think this paper does a great job explaining the technology, validating the technology, but I think new biological findings are lacking in this paper.

We thank the reviewer for the positive assessment of our work and the opportunity to expand upon the biological novelty of our study. The main strengths and advances of applying CRISPR-CATCH to oncogenic ecDNAs are the abilities to (1) isolate megabase-sized ecDNAs and direct profiling of their genetic and epigenetic compositions, (2) separate ecDNA and chromosomal DNA signals for direct comparisons, (3) identify diverse ecDNA structures containing large and complex structural variants. In the revised manuscript, we demonstrated how these abilities of CRISPR-CATCH can lead to new biological findings about ecDNA structure, diversity, origin, and epigenomic landscape as outlined here (major additions summarized in Overview).

- 1. Complete phasing of ecDNAs, chromosomal alleles, and subclonal mutations.** Using ecDNA- and chromosome-specific SNVs identified by CRISPR-CATCH, we demonstrated that ecDNA arose from a single chromosomal parental allele and left behind an excision scar (**revised Figure 2, Extended Data Figure 4**). These data suggest that the allele that served as the original template for the ecDNA no longer contains the sequence harboring *EGFR* and provide strong evidence for the “episome model”, a model of ecDNA formation in which a genomic locus is excised

27from chromosomal DNA as an episome and circularized to form an ecDNA (**Figure 2e**), rather than duplication of sequences due to replication, translocation or other errors (Bailey et al., 2020; Carroll et al., 1988; Storlazzi et al., 2006, 2010). We further discover subclonal mutations only on ecDNAs but not chromosomal *EGFR* alleles, supporting the *kyklonas* model of ongoing ecDNA mutagenesis (Bergstrom et al., 2022).

- 2. CRISPR-CATCH enabled epigenetic analysis led to the discovery of promoter hypomethylation on ecDNAs.** To our knowledge, this is the first ecDNA methylome analysis. This is also the first use of CRISPR-CATCH to profile the epigenetic landscape of isolated DNA molecules. Using this method, we discovered a pattern of promoter CpG hypomethylation on ecDNAs compared to chromosomal DNA originating from the same cell sample (**Figure 3**). This study may serve as the ground work for other types of targeted epigenetic analyses of ecDNAs, including the use of in-vitro DNA labeling strategies for assessment of chromatin accessibility and protein binding (Altemose et al., 2022; Shipony et al., 2020).
- 3. Discovery of ecDNA heterogeneity and functional specialization.** Heterogeneous ecDNA species are observed in many cancer samples (Deshpande et al., 2019), but the full scope and functional implications of the heterogeneity were unclear. CRISPR-CATCH enables molecular separation of multiple megabase-sized ecDNA species based on differences in size, providing fine detail of the composition of DNA segments on each ecDNA species (**revised Figure 4, Extended Data Figures 8,9**). We also implemented a new method for reconstructing ecDNA sequence paths from CRISPR-CATCH sequencing data, called **Candidate Amplicon Path EnumeratoR** (CAMPER; Methods), to perform full reconstruction of megabase-sized ecDNA circles containing multiple sequence segments. Furthermore, we discovered ecDNA species with oncogenes that are missing their enhancers and ecDNA species with select enhancer amplification, challenging the current dogma of gene expression (**Figure 4c,g,h, Extended Data Figure 9a**). These findings suggest that extrachromosomal amplification and rearrangement events may be driven by enhancer proximity to oncogenes on an ecDNA molecule as well as overall abundance of enhancer sequences in a pool of ecDNA molecules. As ecDNAs can interact with one another in trans within a hub (Hung et al., 2021), amplification of enhancer sequences in a pool of ecDNAs facilitates intermolecular enhancer-promoter interactions and further increase oncogene expression. These analyses demonstrate the utility of CRISPR-CATCH in identifying ecDNA oncogene amplicon

structures in cancer cells and how mapping these structures can be used to understand the altered enhancer landscape on ecDNAs.

In addition, I also believe that this technology has a few drawbacks/disadvantages, that are open for discussion:

1. In the introduction, the authors mentioned that “Given the prevalence of ecDNA in cancer, there is an urgent need for better characterization of unique genetic and epigenetic, features of ecDNA in order to understand how it may differ from chromosomal DNA and obtain clues about how it is formed and maintained in tumors.” I think the authors should emphasis with 1-2 sentences on why this particular technology is important, i.e. why do we need to sequence long ecDNA? What additional information does it give comparing to the shorter ecDNAs? I think these kinds of information will be helpful to highlight the importance of this technology.

We thank the reviewer for this comment and opportunity to clarify the importance of the technical gap filled by CRISPR-CATCH. The majority of ecDNAs observed in human cancers are larger than 200 kb (**Extended Data Figure 1c**). In this size range, DNA molecules are highly prone to breakage in solution and cannot be enriched using current methods such as Circle-seq, which enriches and amplifies much smaller DNA circles than those observed in human cancers (optimized for circles smaller than 10 kb; discussed in detail in response to **Question #2** below). Therefore, it is currently extremely challenging to separate any clonally selected ecDNAs from cancer cells. If we focus only on the short ecDNAs (< 10 kb), we would miss the majority of oncogenic ecDNAs in cancer.

We have revised the manuscript to clarify why it is important to profile these large ecDNAs observed in cancer (**lines 123-126**). To emphasize the importance of this advance, we also constructed a decision tree based on currently available technologies for ecDNA characterization with recommendations for how CRISPR-CATCH can be used (**Extended Data Figure 10**).

2. The authors explained very well on describing the three current technologies used for studying ecDNA and the disadvantages of each technology. The authors mentioned that “This method (circle-seq) requires intact DNA circles and is therefore highly limited by ecDNA size, as megabase-sized DNA molecules are extremely fragile in solution and prone to breakage.” Is there any data or reference supporting this? I would suggest that the authors provide some data on this, for example, perform exonuclease digestion, and then run a gel on the digested product without MDA amplification? Are all megabase-

29sized ecDNA gone? 100%? Or 90%? Or 50%? If it is not 100%, then why could not we just gel cut those megabase-sized ecDNA and run nanopore on them since nanopore sequencing could work with single molecules so you don't need to have that many molecules anyway.

We thank the reviewer for the comment and suggestion. We compared DNA treated with either CRISPR-CATCH or in-solution high molecular weight (HMW) DNA isolation, followed by pulsed field gel electrophoresis to assess DNA molecule size. We found that while the GBM39 ecDNA is shown as an intact ~1.2Mb molecule by CRISPR-CATCH, in-solution DNA extraction led to shearing of virtually all DNA into fragments smaller than 225kb (**Extended Data Figure 1a**). This DNA breakage was observed despite careful handling (slow pipetting, use of wide-bore tips, avoidance of freeze-thawing, etc.). This observation shows that the vast majority of ecDNAs are sheared simply because they are isolated in solution. These data are not particularly surprising, as manufacturers of in-solution high molecular weight DNA extraction on commercial kits themselves recommend usage for DNA molecules up to 200 kb (as an example please see one of the most commonly used kit, MagAttract HMW DNA Kit, by Qiagen: <https://www.qiagen.com/us/products/discovery-and-translational-research/dna-rna-purification/dna-purification/genomic-dna/magattract-hmw-dna-kit-48/>). To further attempt to identify any residual intact ecDNAs in solution, we performed exonuclease digestion as the reviewer suggested followed by pulsed field gel electrophoresis to assess DNA molecule size. Following exonuclease digestion, no DNA could be visualized by pulsed field gel electrophoresis (**Extended Data Figure 1b**). This further supports our conclusion that circular ecDNAs are no longer intact in the in-solution-extracted DNA sample. Therefore, using previous Circle-seq protocols which involve exonuclease digestion, oncogenic ecDNAs are digested and cannot be isolated. This observation is also not surprising, as previous studies on small extrachromosomal circular DNAs (eccDNAs) showed that the vast majority of circles isolated using Circle-seq are between 0.1 kb and 10 kb (**Figure R2**), two to three orders of magnitude smaller than clonally selected ecDNAs identified in cancer cells (Møller et al., 2018; Wang et al., 2021). Together, these data suggest that DNA circles larger than 200 kb are broken in solution and indistinguishable from DNA fragments resulting from linear chromosomal DNA. Importantly, the majority of oncogenic ecDNAs observed in human cancers are above 200 kb (**Extended Data Figure 1c**, elaborated above in response to **Question #1**). To characterize these oncogenic ecDNA molecules, a robust method for targeted isolation and profiling is needed.

Figure R2. Size distribution of extrachromosomal circular DNA isolated in solution by Circle-seq. (a) Size distribution of extrachromosomal circular DNA detected by Circle-seq (Møller et al., 2018). **(b)** Size distribution of extrachromosomal circular DNA detected by a modified Circle-seq protocol (Wang et al., 2021).

From a practical standpoint, we conclude that in-solution DNA extraction is suboptimal for molecules the size of the typical ecDNA as too much valuable ecDNA is lost. For analysis of mutations as well as structures of ecDNAs using either nanopore sequencing or short-read sequencing, we require sufficient detection of reads representing these mutations and structural rearrangements. This is particularly important when there is a heterogeneous mixture of ecDNAs such as shown in our manuscript for the SNU16 cell line. Finally, while the reviewer is correct that nanopore sequencing works with single molecules, it currently still requires a relatively high DNA input for sequencing adapter ligation or transposition (ideally above 100 ng). Using CRISPR-CATCH on agarose-embedded genomic DNA allows us to preserve as much intact ecDNA as possible. This is very important for DNA sequencing without PCR amplification, which enables direct profiling of the epigenetic landscape of ecDNAs such as CpG methylation as shown in this manuscript.

Finally, CRISPR-CATCH enables enrichment of specific ecDNA species and visualization of corresponding amplicon sizes on the agarose gel. This provides additional information compared to exonuclease digestion of genomic DNA because heterogeneous and rearranged ecDNA sequences can be precisely mapped and separated using CRISPR-CATCH. This was demonstrated in the current manuscript using the gastric cancer cell line SNU16, in which more than 10 ecDNA species are present and could be enumerated (**Figure 4, Extended Data Figures 8,9**). We further validated these individual ecDNA species identified by CRISPR-CATCH using overlapping optical mapping read contigs, background signals from chromatin conformation capture data (H3K27 acetylation HiChIP), and dual-color DNA FISH on

metaphase spreads. Given the large ecDNA sizes observed in human cancers (**Extended Data Figure 1c**) and structural heterogeneity within a cancer cell population (**Figure 4, Extended Data Figure 9**), we believe CRISPR-CATCH overcomes the significant obstacle of targeted ecDNA profiling. We have included these new data and analyses in the revised manuscript as suggested by the reviewer.

3. This technology although allows separation of megabase-size ecDNA, but it is targeted, so we would need to know where it covers beforehand so that we could design sgRNA on that particular location. So we would need to have prior knowledge on what we are looking for, which is probably one disadvantage of this technology. Could you please comment on this?

We fully agree that our method requires a prior knowledge of the amplified genomic locus/loci. We have acknowledged the limitation of requiring prior knowledge of amplified gene locus in the revised Discussion (**lines 406-409**). We note that whole-genome sequencing (WGS) or exome data are routinely generated for new cancer samples and already exist for many established cell lines and most human tumor types, such in TCGA and PCAWG. Thus the characteristic amplification landscape for most tumors are known. These bulk sequencing data can be analyzed by AmpliconArchitect (Deshpande et al., 2019) to identify amplified regions and predict potential ecDNA amplifications. Therefore, we find that it is straight forward to quickly narrow down potential ecDNA loci based on preliminary bulk DNA sequencing data. Metaphase DNA FISH is also routinely performed to validate ecDNA status on chromosomal spreads. However, as the reviewer pointed out, the major hurdle currently is in targeted isolation and profiling of ecDNA. CRISPR-CATCH fills this technical gap by enabling targeted analysis of ecDNA genetic and epigenetic composition with the resolution of short- and long-read sequencing while providing signal specificity to ecDNA. To demonstrate this technical need and how CRISPR-CATCH can meet it, we constructed a decision tree based on currently available technologies for ecDNA characterization with recommendations for how CRISPR-CATCH can be used (**Extended Data Figure 10**).

4. When applied to human samples, as we all know, the population is heterogenous. For example, in any given targeted gene, there could be cells with 4 different profiles: 1. WT chrDNA + WT ecDNA; 2. Mut chrDNA + WT ecDNA; 3. WT chrDNA + Mut ecDNA; 4. Mut chrDNA + Mut ecDNA. Any human tissue would consist of a mixture of all 4 possibilities, so in the end even if we can get the ecDNA mutation information, we would not be able to tell if they are consistent or inconsistent with the cell chrDNA because the linkage between chrDNA and ecDNA is lost, which is probably another disadvantage of

32this technology. This technology has only been applied to human cell lines, which is a homogeneous population. Could you please comment on this?

We thank the reviewer for the comment and the opportunity to elaborate on the point about cell heterogeneity. As the reviewer rightly pointed out, cancer cell populations are often heterogeneous. In the hypothetical scenario provided by the reviewer in which cells with four different profiles co-exist in a population, linkage between ecDNA and chromosomal DNA mutational statuses would have to be established on the single-cell level. This information is lost with any bulk sequencing technologies currently available. Furthermore, current single-cell sequencing technologies are also not capable of resolving these profiles because sequencing reads from any given cell supporting a mutation could have originated from either ecDNA or chromosomal DNA. Therefore, if identical mutations can be present on either ecDNA or chromosomal DNA (or both), it appears that the only way to resolve the 4 possibilities (as laid out by the reviewer) would be to simultaneously tag the cell identity as well as ecDNA/chromosomal DNA identity on each individual DNA molecule for sequencing, which is not a technology that is available at the moment. Thus, we respectfully disagree with the reviewer on the claim that the inability to link chromosomal and ecDNA mutations is a disadvantage unique to CRISPR-CATCH. Rather, it is not currently a feasible task using any existing sequencing methods.

On the other hand, if the chromosomal DNA and ecDNA mutations are different, they can first be phased and attributed to either ecDNA or chromosomal DNA using CRISPR-CATCH (as shown in **Figure 2** for GBM39 chromosomal DNA and ecDNA variants), then identified and linked on the single-cell level using single-cell sequencing. For example, if mutation A exists only on ecDNA and mutation B exists only on chromosomal DNA, then identification of both mutations in the same cells by single-cell sequencing would suggest that they co-exist. In the case of cancer samples with more than one type of ecDNA (such as the heterogeneous ecDNA structures shown in the SNU16 stomach cancer cells; **Figure 4, Extended Data Figure 9**), mutations can also be phased for each amplicon. In this respect, CRISPR-CATCH would in fact allow us to better interpret cell heterogeneity in single-cell sequencing data. The latter scenario emphasizes the utility of phasing and attributing mutations on ecDNA, a task that can now be achieved using CRISPR-CATCH (**Figure 2, Extended Data Figure 4**). These analyses further demonstrate the utility of CRISPR-CATCH on heterogeneous cancer cells.

Finally, we agree with the reviewer that expanding the capabilities of CRISPR-CATCH beyond cell lines can add to its utility in understanding ecDNA genetic and epigenetic characteristics in a heterogeneous tumor cell population. Following the reviewer's suggestion, we have now optimized methods for applying CRISPR-CATCH for clinical tumor samples from embedding tumors directly into agarose, chemical tumor dissociation, lysis, digestion and removal of fragmented tumor DNA to CRISPR-CATCH. One main difference between a tumor sample and a cancer cell line sample is the presence of fragmented DNA and general lower quality and amount of genomic DNA in a tumor sample due to tumor cell death and/or sample preparation and freezing. These DNA fragments can migrate in pulsed field gel electrophoresis and result in random sequencing background. To remove this background, we introduce electro-depletion, a sequential electrophoretic strategy to remove fragmented DNA from patient tumor samples and enable CRISPR-CATCH (**Figure 1h**). Briefly, processed tumor DNA entrapped in agarose plugs is loaded into a first agarose gel and a constant voltage is applied to deplete agarose plugs of DNA fragments. The electric field is then briefly reversed in direction to ensure that the intact DNA remains trapped in agarose plugs. These agarose plugs are then removed from the gel and subjected to CRISPR-CATCH as we described in the manuscript. This strategy effectively removes DNA fragments and traps intact genomic DNA as well as intact circular ecDNA, as evidenced by the presence of ecDNA bands from SNU16 cells after applying this method (**Extended Data Figure 3a,b**). For our clinical tumor sample, DNA bands were not visible after PFGE due to low amounts of DNA. Nonetheless, CRISPR-CATCH still successfully enriched for ecDNAs and confirmed the amplicon size shown by strong agreement between the molecular size on the gel and the length of the enriched amplified region in sequencing (**Fig. 1i, Extended Data Figure 3c-e**).

We applied optimized CRISPR-CATCH to an instructive case of metastatic melanoma. A patient with *BRAF* V600MUT melanoma was treated on BRAF and MEK inhibitors and developed metastatic lesion with acquired resistance coincident with the acquisition of ecDNA (**Extended Data Figure 3c**). CRISPR-CATCH and AmpliconArchitect confirmed the amplification of an 890-kb ecDNA encompassing *NRAS*, a gene known to confer resistance to BRAF inhibition (Nazarian et al., 2010) (**Fig. 1i, Extended Data Figure 3d**). Together, these data show that CRISPR-CATCH is fully feasible on human tumor specimens, and validate an ecDNA mechanism for acquired resistance to MAP kinase pathway inhibitors in authentic human cancer. We agree with the reviewer that the application of CRISPR-CATCH to heterogeneous cell populations such as clinical tumor specimens significantly adds to its utility and thank the reviewer for this suggestion, which greatly improved this work.

5. Just a suggestion: The authors use the word “purifying”, “purification” a lot. However, the purity after purification is not very good (60% as shown in Figure 1e), and not reported directly in the paper (i.e., when I search for the keyword purity, it is not there). I would rather use the words “enriched”, “separation”, and emphasis on how many times it is enriched (like the authors mentioned: x30 enriched, etc.).

We thank the reviewer for the comment and agree that “purification” is not the most appropriate term. We have revised the manuscript to describe this method as “enrichment”, “separation”, or “isolation” of ecDNA. We would like to point out that the level of enrichment was detailed in the text and was explicitly shown in **Figure 1e,f**, and **Extended Data Figure 2c,d** (highlighted in lines 150-152: “CRISPR-CATCH enabled a 30-fold enrichment of the targeted ecDNA (60% of all sequencing reads vs. 2% in WGS), resulting in ultrahigh (~200x normalized) sequencing coverage (**Figure 1e,f**, ecDNA in **Extended Data Figure 2b**)”).

6. Since CRISPR-CATCH could capture megabase-sized DNA, why do you need to do *de novo* assembly? Why couldn't we just use nanopore sequencing, read the full length and it is done?

We thank the reviewer for the comment. As a clarification, our method does not use *de novo* assembly; it uses a reference-guided assembly approach which identifies paths in a graph derived from alignment of CRISPR-CATCH sequencing reads to a reference genome. We do perform a *de novo* reconstruction of ecDNA circles independent of bulk sequencing, after alignment and identification of CRISPR-CATCH-isolated DNA segments, such that a circular structure can be identified by exploration of the resulting isolated genome graph. We also note that CRISPR-CATCH enrichment of ecDNA is absolutely essential for our genomic and epigenomic analyses. Long-read sequencing of unenriched total DNA led to lower genomic coverage and inability to phase ecDNA vs. chromosomal variants (**Reviewer 2, Question #11**).

Certainly, nanopore and other long-read sequencing approaches provide a greater ability to detect individual genomic breakpoints, largely in low-complexity, poorly mappable regions of the genome by providing longer flanking sequences to anchor the endpoints of the breakpoints. While useful for assembling difficult regions of the genome, accurate breakpoint detection with NGS data works well for mappable, gene-bearing regions of the genome, such as those selected on ecDNA. We developed CAMPER, a computational method to reconstruct ecDNA primary sequence (**Figure 4**).

35The accuracy of our breakpoint detection is also evidenced by the fact that our breakpoints detected by NGS are also supported in the optical mapping data (**Figure 4d-f, Extended Data Figure 8b**). We also note that achieving comparable genomic coverage with long reads is currently substantially more costly than for NGS. With standard sample preparation, the average nanopore read is in the range of 25kbp. Typical ecDNA structures, however, are 1-2 Mbp, placing them two orders of magnitude larger than the average length of nanopore reads. Only a very small fraction of nanopore reads exist in sizes larger than 200 kbp without special sample prep (e.g. Circulomics kits), and thus to get a bioinformatically meaningful amount of genomic coverage of the ecDNA with such ultra-long nanopore reads, exorbitant genomic coverage levels must be achieved. With special sample preparation, read length N50 goes to around 100kb, which is still 10-20x shorter than the average ecDNA (and worse for larger ecDNA). Lastly, because the sequenced DNA is being extracted from an agarose gel, great care would need to be taken to preserve ultra-high molecular weight DNA during the extraction process, further complicating this approach. However, the reviewer's point that ecDNA reconstruction should be amenable to long-read based sequencing is well taken, and we intend to build in compatibility for long-reads with AmpliconArchitect in the future. However, we view that task as being outside the scope of this paper. We feel its absence does not diminish the fact that this current approach combining CRISPR-CATCH and short-read sequencing works robustly and uses very mainstream technologies currently available to almost all sequencing centers.

7. This method might suffer from multiplexing issue because each ecDNA might consist of DNAs from different chromosomes, so it is possible to lose information (similar to sgRNA A+B case in Figure f) if we design multiple sgRNAs targeting multiple loci. So, this technique should only be used using one sgRNA at a time. Is this correct?

We thank the reviewer for the comment. In principle, a pool of sgRNAs could be used if they target different loci. If an ecDNA contains any of the targeted loci, it would be cleaved and enriched by CRISPR-CATCH. The reviewer is also correct that if a single circle contains multiple target loci, it may be cleaved more than once. However, the resulting fragments would still be captured by CRISPR-CATCH unless two sgRNAs cut very close to each other. sgRNA A+B shown in **Figure 1** in our manuscript were intentionally designed to cut extremely close to each other (within 20 kb) to demonstrate the point that the end-to-end junction of a circular structure explains enrichment of the *EGFR* ecDNA and depletion of the small fragment between the two cut sites. Given that ecDNAs are typically on the scale of megabases, sgRNA target sites in two different loci on the same ecDNA molecule would not be this close to each other. A fragment that is

3680 kb or larger could easily be visualized and separated by pulsed field gel electrophoresis using the same settings shown in the manuscript. Therefore, as long as all fragments are captured, information would not be lost.

We believe the power of CRISPR-CATCH mainly resides in its ability for targeted enrichment and compatibility with short-read/long-read sequencing for ecDNA sequence mapping and epigenetic profiling as demonstrated in the current manuscript. We also note that the vast majority of ecDNAs identified thus far in human cancers represent a small number of oncogene loci (e.g. *MYC*, *EGFR*, *FGFR2*, *MYCN*, *CCND1*, *ERBB2*, *KRAS*, *CDK4*) (Kim et al., 2020; Turner et al., 2017). Therefore, a collection of a dozen sgRNAs would likely be broadly applicable to many if not most ecDNAs in human cancers. This observation, combined with the fact that WGS and/or DNA FISH data are routinely collected for cancer cell samples and provide clues about oncogenes that may be amplified on ecDNAs, means that one can easily and rapidly apply CRISPR-CATCH to a new cell sample in practice without multiplexing. Currently the main obstacle in the ecDNA field is the ability to separate ecDNAs from the rest of the genome and we demonstrate in this technical report that CRISPR-CATCH provides a robust solution to this problem. We agree that multiplexing could be a useful future goal, and have added this point to Discussion (lines 409-411).

References

Abramov, S., Boytsov, A., Bykova, D., Penzar, D.D., Yevshin, I., Kolmykov, S.K., Fridman, M.V., Favorov, A.V., Vorontsov, I.E., Baulin, E., et al. (2021). Landscape of allele-specific transcription factor binding in the human genome. *Nat Commun* 12, 2751. <https://doi.org/10.1038/s41467-021-23007-0>.

Altemose, N., Maslan, A., Smith, O.K., Sundararajan, K., Brown, R.R., Mishra, R., Detweiler, A.M., Neff, N., Miga, K.H., Straight, A.F., et al. (2022). DiMeLo-seq: a long-read, single-molecule method for mapping protein–DNA interactions genome wide. *Nat Methods* 1–13. <https://doi.org/10.1038/s41592-022-01475-6>.

Bailey, C., Shoura, M.J., Mischel, P.S., and Swanton, C. (2020). Extrachromosomal DNA—relieving heredity constraints, accelerating tumour evolution. *Annals of Oncology* 31, 884–893. <https://doi.org/10.1016/j.annonc.2020.03.303>.

Bergstrom, E.N., Luebeck, J., Petljak, M., Khandekar, A., Barnes, M., Zhang, T., Steele, C.D., Pillay, N., Landi, M.T., Bafna, V., et al. (2022). Mapping clustered mutations in cancer reveals APOBEC3 mutagenesis of ecDNA. *Nature* <https://doi.org/10.1038/s41586-022-04398-6>.

van der Bliek, A.M., Lincke, C.R., and Borst, P. (1988). Circular DNA of 3T6R50 double minute chromosomes. *Nucleic Acids Research* 16, 4841–4851. <https://doi.org/10.1093/nar/16.11.4841>.

Carroll, S.M., DeRose, M.L., Gaudray, P., Moore, C.M., Needham-Vandevanter, D.R., Von Hoff, D.D., and Wahl, G.M. (1988). Double minute chromosomes can be produced from precursors derived from a chromosomal deletion. *Molecular and Cellular Biology* 8, 1525–1533. <https://doi.org/10.1128/MCB.8.4.1525>.

Deshpande, V., Luebeck, J., Nguyen, N.-P.D., Bakhtiari, M., Turner, K.M., Schwab, R., Carter, H., Mischel, P.S., and Bafna, V. (2019). Exploring the landscape of focal amplifications in cancer using AmpliconArchitect. *Nature Communications* 10, 392. <https://doi.org/10.1038/s41467-018-08200-y>.

Hung, K.L., Yost, K.E., Xie, L., Shi, Q., Helmsauer, K., Luebeck, J., Schöpflin, R., Lange, J.T., Chamorro González, R., Weiser, N.E., et al. (2021). ecDNA hubs drive

cooperative intermolecular oncogene expression. *Nature* 1–6.
<https://doi.org/10.1038/s41586-021-04116-8>.

Kim, H., Nguyen, N.-P., Turner, K., Wu, S., Gujar, A.D., Luebeck, J., Liu, J., Deshpande, V., Rajkumar, U., Namburi, S., et al. (2020). Extrachromosomal DNA is associated with oncogene amplification and poor outcome across multiple cancers. *Nature Genetics* 52, 891–897. <https://doi.org/10.1038/s41588-020-0678-2>.

Lange, J.T., Chen, C.Y., Pichugin, Y., Xie, F., Tang, J., Hung, K.L., Yost, K.E., Shi, Q., Erb, M.L., Rajkumar, U., et al. (2021). Principles of ecDNA random inheritance drive rapid genome change and therapy resistance in human cancers (*Cancer Biology*).

Luebeck, J., Coruh, C., Dehkordi, S.R., Lange, J.T., Turner, K.M., Deshpande, V., Pai, D.A., Zhang, C., Rajkumar, U., Law, J.A., et al. (2020). AmpliconReconstructor integrates NGS and optical mapping to resolve the complex structures of focal amplifications. *Nature Communications* 11, 4374. <https://doi.org/10.1038/s41467-020-18099-z>.

Møller, H.D., Mohiyuddin, M., Prada-Luengo, I., Sailani, M.R., Halling, J.F., Plomgaard, P., Maretty, L., Hansen, A.J., Snyder, M.P., Pilegaard, H., et al. (2018). Circular DNA elements of chromosomal origin are common in healthy human somatic tissue. *Nature Communications* 9, 1069. <https://doi.org/10.1038/s41467-018-03369-8>.

Morton, A.R., Dogan-Artun, N., Faber, Z.J., MacLeod, G., Bartels, C.F., Piazza, M.S., Allan, K.C., Mack, S.C., Wang, X., Gimple, R.C., et al. (2019). Functional Enhancers Shape Extrachromosomal Oncogene Amplifications. *Cell* 0. <https://doi.org/10.1016/j.cell.2019.10.039>.

Nathanson, D.A., Gini, B., Mottahedeh, J., Visnyei, K., Koga, T., Gomez, G., Eskin, A., Hwang, K., Wang, J., Masui, K., et al. (2014). Targeted Therapy Resistance Mediated by Dynamic Regulation of Extrachromosomal Mutant EGFR DNA. *Science* 343, 72–76. <https://doi.org/10.1126/science.1241328>.

Nazarian, R., Shi, H., Wang, Q., Kong, X., Koya, R.C., Lee, H., Chen, Z., Lee, M.-K., Attar, N., Sazegar, H., et al. (2010). Melanomas acquire resistance to B-RAF(V600E) inhibition by RTK or N-RAS upregulation. *Nature* 468, 973–977. <https://doi.org/10.1038/nature09626>.

Nikolaev, S., Santoni, F., Garieri, M., Makrythanasis, P., Falconnet, E., Guipponi, M., Vannier, A., Radovanovic, I., Bena, F., Forestier, F., et al. (2014). Extrachromosomal driver mutations in glioblastoma and low-grade glioma. *Nature Communications* 5, 5690. <https://doi.org/10.1038/ncomms6690>.

Shipony, Z., Marinov, G.K., Swaffer, M.P., Sinnott-Armstrong, N.A., Skotheim, J.M., Kundaje, A., and Greenleaf, W.J. (2020). Long-range single-molecule mapping of chromatin accessibility in eukaryotes. *Nature Methods* 1–9. <https://doi.org/10.1038/s41592-019-0730-2>.

Storlazzi, C.T., Fioretos, T., Surace, C., Lonoce, A., Mastrorilli, A., Strömbeck, B., D’Addabbo, P., Iacovelli, F., Minervini, C., Aventin, A., et al. (2006). MYC-containing double minutes in hematologic malignancies: evidence in favor of the episome model and exclusion of MYC as the target gene. *Human Molecular Genetics* 15, 933–942. <https://doi.org/10.1093/hmg/ddl010>.

Storlazzi, C.T., Lonoce, A., Guastadisegni, M.C., Trombetta, D., D’Addabbo, P., Daniele, G., L’Abbate, A., Macchia, G., Surace, C., Kok, K., et al. (2010). Gene amplification as double minutes or homogeneously staining regions in solid tumors: Origin and structure. *Genome Res.* 20, 1198–1206. <https://doi.org/10.1101/gr.106252.110>.

Turner, K.M., Deshpande, V., Beyter, D., Koga, T., Rusert, J., Lee, C., Li, B., Arden, K., Ren, B., Nathanson, D.A., et al. (2017). Extrachromosomal oncogene amplification drives tumour evolution and genetic heterogeneity. *Nature* 543, 122–125. <https://doi.org/10.1038/nature21356>.

Wang, Y., Wang, M., Djekidel, M.N., Chen, H., Liu, D., Alt, F.W., and Zhang, Y. (2021). eccDNAs are apoptotic products with high innate immunostimulatory activity. *Nature* 1–7. <https://doi.org/10.1038/s41586-021-04009-w>.

Wu, S., Turner, K.M., Nguyen, N., Raviram, R., Erb, M., Santini, J., Luebeck, J., Rajkumar, U., Diao, Y., Li, B., et al. (2019). Circular ecDNA promotes accessible chromatin and high oncogene expression. *Nature* 1–5. <https://doi.org/10.1038/s41586-019-1763-5>.

Decision Letter, first revision:

Our ref: NG-TR58773R1

16th Jun 2022

Dear Howard,

Thank you for submitting your revised manuscript entitled "Targeted profiling of human extrachromosomal DNA by CRISPR-CATCH" (NG-TR58773R1). It has now been seen by two of the original referees and their comments are below. Despite our multiple chase emails, reviewer #2 has not submitted a timely report. We have now decide to proceed without their comments.

The reviewers find that the paper has improved in revision, and therefore we'll be happy in principle to publish it in Nature Genetics, pending minor revisions to comply with our editorial and formatting guidelines.

Thank you again for your interest in Nature Genetics. Please do not hesitate to contact me if you have any questions.

Congratulations!

Sincerely,

Tiago

Tiago Faial, PhD
Senior Editor
Nature Genetics
<https://orcid.org/0000-0003-0864-1200>

Reviewer #1 (Remarks to the Author):

This revision is a considerable improvement over the previous version. An impressive amount of new data and findings have been added. Specifically, the new data from the metastatic melanoma patient

41sample and the phasing and SNV info strengthens the paper a lot. I'm satisfied.

Reviewer #3 (Remarks to the Author):

Overall, the work done by the authors is very impressive and the technology is very innovative and interesting and will be very useful for a lot of applications.

Previously, I had a few questions regarding the technology and, in my opinion, it was also lacking biological findings in the paper.

In this new revision paper, the authors performed additional experiments and further answered all the questions that I raised before. They further compared their technology to in-solution DNA isolation and proved that it was not previously possible to analyze megabase-size ecDNAs. They also explained why Nanopore sequencing won't work in these applications. Furthermore, they optimized and applied the protocol for analyzing patient tumor tissue, which is a significant step forward than just analyzing cell lines.

Final Decision Letter:

In reply please quote: NG-TR58773R2 Chang

22nd Aug 2022

Dear Howard,

I am delighted to say that your manuscript entitled "Targeted profiling of human extrachromosomal DNA by CRISPR-CATCH" has been accepted for publication in an upcoming issue of Nature Genetics.

42Due to the importance of these deadlines, we ask that you please let us know now whether you will be difficult to contact over the next month. If this is the case, we ask you provide us with the contact information (email, phone and fax) of someone who will be able to check the proofs on your behalf, and who will be available to address any last-minute problems.

Your paper will be published online after we receive your corrections and will appear in print in the next available issue. You can find out your date of online publication by contacting the Nature Press Office (press@nature.com) after sending your e-proof corrections. Now is the time to inform your Public Relations or Press Office about your paper, as they might be interested in promoting its publication. This will allow them time to prepare an accurate and satisfactory press release. Include your manuscript tracking number (NG-TR58773R2) and the name of the journal, which they will need when they contact our Press Office.

Please note that *Nature Genetics* is a Transformative Journal (TJ). Authors may publish their research with us through the traditional subscription access route or make their paper immediately open access through payment of an article-processing charge (APC). Authors will not be required to make a final decision about access to their article until it has been accepted. [Find out more about Transformative Journals](https://www.springernature.com/gp/open-research/transformative-journals)

Authors may need to take specific actions to achieve [compliance](https://www.springernature.com/gp/open-research/funding/policy-compliance-faqs) with funder and institutional open access mandates. If your research is supported by a funder that requires immediate open access (e.g. according to [a](https://www.springernature.com/gp/open-research/funding/policy-compliance-faqs)

<https://www.springernature.com/gp/open-research/plan-s-compliance>) Plan S principles) then you should select the gold OA route, and we will direct you to the compliant route where possible. For authors selecting the subscription publication route, the journal's standard licensing terms will need to be accepted, including <https://www.nature.com/nature-portfolio/editorial-policies/self-archiving-and-license-to-publish>. Those licensing terms will supersede any other terms that the author or any third party may assert apply to any version of the manuscript.

Please note that Nature Portfolio offers an immediate open access option only for papers that were first submitted after 1 January, 2021.

Sincerely,

Tiago

Tiago Faial, PhD
Chief Editor
Nature Genetics
<https://orcid.org/0000-0003-0864-1200>